# Target-Embedding Autoencoders for Supervised Representation Learning

**Daniel Jarrett**
Department of Mathematics
University of Cambridge, UK
`daniel.jarrett@maths.cam.ac.uk`

**Mihaela van der Schaar**
University of Cambridge, UK
University of California, Los Angeles, USA
`mv472@cam.ac.uk, mihaela@ee.ucla.edu`

## Abstract

Autoencoder-based learning has emerged as a staple for disciplining representations in unsupervised and semi-supervised settings. This paper analyzes a framework for improving generalization in a purely supervised setting, where the target space is high-dimensional. We motivate and formalize the general framework of target-embedding autoencoders (TEA) for supervised prediction, learning intermediate latent representations jointly optimized to be both predictable from features as well as predictive of targets—encoding the prior that variations in targets are driven by a compact set of underlying factors. As our theoretical contribution, we provide a guarantee of generalization for linear TEAs by demonstrating uniform stability, interpreting the benefit of the auxiliary reconstruction task as a form of regularization. As our empirical contribution, we extend validation of this approach beyond existing static classification applications to multivariate sequence forecasting, verifying their advantage on both linear and nonlinear recurrent architectures—thereby underscoring the further generality of this framework beyond feedforward instantiations.

## 1 Introduction

Representation learning deals with uncovering useful underlying structures of data, and autoencoders (Hinton & Salakhutdinov, 2006) have been a staple in a variety of problems. While much research focuses on its use in unsupervised or semi-supervised settings with such diverse objectives as sparsity (Ranzato et al., 2007), generation (Kingma & Welling, 2013), and disentanglement (Chen et al., 2018), autoencoders are also useful in *purely supervised* settings—in particular, adding an auxiliary feature-reconstruction task to supervised classification problems has been shown to empirically improve generalization (Le et al., 2018); in the linear case, the theoretically quantifiable benefit matches that of simplistic norm-based regularization (Bousquet & Elisseeff, 2002; Rosasco & Poggio, 2009).

In this paper, we consider the inverse problem setting where the *target space* $\mathcal{Y}$ is high-dimensional; for instance, consider the multi-label classification tasks of object tagging, text annotation, and image segmentation. This is in contrast to the vast majority of works designed to tackle a high-dimensional feature space $\mathcal{X}$ (where commonly $|\mathcal{X}| \gg |\mathcal{Y}|$, such as in standard classification problems). In this setting, the usual (and universal) strategy of learning to reconstruct features (Weston et al., 2012; Kingma et al., 2014; Le et al., 2018) may not be most useful: learning latent representations that encapsulate the variation within $\mathcal{X}$ does not directly address the more challenging problem of mapping back up to a higher-dimensional $\mathcal{Y}$. Instead, we argue for leveraging intermediate representations that are compact and more easily *predictable* from features, yet simultaneously guaranteed to be *predictive* of targets. In the process, we provide a unified theoretical perspective on recent applications of autoencoders to label-embedding in static, high-dimensional classification problems (Yu et al., 2014; Girdhar et al., 2016; Yeh et al., 2017). Extending into the temporal setting, we further empirically demonstrate the generality of target-embedding for recurrent, multi-variate sequence forecasting.

Our contributions are three-fold. First, we motivate and formalize the target-embedding autoencoder (TEA) framework: a general approach applicable to any underlying architecture. Second, we provide a theoretical learning guarantee in the linear case by demonstrating uniform stability; specifically, we obtain an $O(1/N)$ bound on instability by analogizing the benefit of the auxiliary reconstruction task to a form of regularization—without incurring additional bias from explicit shrinkage. Finally, we extend empirical validation of this approach beyond the domain of static classification: using the task of multivariate disease trajectory forecasting as case study, we experimentally validate the

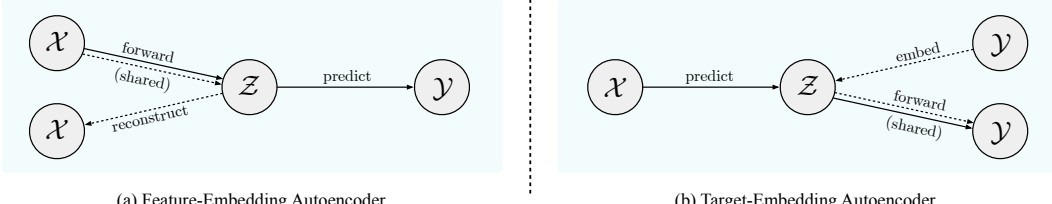

(a) Feature-Embedding Autoencoder          (b) Target-Embedding Autoencoder

Figure 1: (a) Feature-embedding and (b) Target-embedding autoencoders. Solid lines correspond to the (primary) prediction task; dashed lines to the (auxiliary) reconstruction task. Shared components are involved in both.

advantage that TEAs confer on both linear and nonlinear architectures using real-world datasets with both continuous and discrete targets. To the best of our knowledge, we are the first to formalize and quantify the theoretical benefit of autoencoder-based target-representation learning in a purely supervised setting, and to extend its application to the domain of multivariate sequence forecasting.

## 2   TARGET-EMBEDDING AUTOENCODERS

Let $\mathcal{X}$ and $\mathcal{Y}$ be finite-dimensional vector spaces, and consider the supervised learning problem of predicting targets $\mathbf{y} \in \mathcal{Y}$ from features $\mathbf{x} \in \mathcal{X}$. With a finite batch of $N$ training instances $\mathcal{D} = \{(\mathbf{x}_n, \mathbf{y}_n)\}_{n=1}^{N}$, the objective is to learn a mapping $h : \mathcal{X} \to \mathcal{Y}$ that generalizes well to new samples from the same distribution. The vast majority of existing work consider the setting—most commonly, classification—where $|\mathcal{X}| \gg |\mathcal{Y}|$; under this scenario, autoencoders are often used to first transform the input into some lower-dimensional representation $\mathbf{z} \in \mathcal{Z}$ amenable to the downstream task. Doing so involves adding an auxiliary reconstruction loss $\ell_r$ to the primary prediction loss $\ell_p$.

Formally, solutions of this form—in supervised and semi-supervised settings alike—consist of a shared forward model $\phi : \mathcal{X} \to \mathcal{Z}$, a reconstruction function $r : \mathcal{Z} \to \mathcal{X}$, and a prediction function $d : \mathcal{Z} \to \mathcal{Y}$ during training (where notation $d$ reflects the downstream nature of the prediction task). Denote $\tilde{\mathbf{x}} = r(\phi(\mathbf{x}))$ and $\hat{\mathbf{y}} = d(\phi(\mathbf{x}))$; then the complete loss function takes the following form,

$$L = \frac{1}{N}\sum_{n=1}^{N} \left[\ell_p(\hat{\mathbf{y}}_n, \mathbf{y}_n) + \ell_r(\tilde{\mathbf{x}}_n, \mathbf{x}_n)\right] \tag{1}$$

In contrast, we focus on settings where the target space $\mathcal{Y}$ is high-dimensional, and where possibly $|\mathcal{Y}| > |\mathcal{X}|$. In this case, we argue that learning to reconstruct the input is not necessarily most beneficial. In a simple classification problem, autoencoding inputs leverages the hypothesis that a reconstructive representation is also likely discriminative. In our setting, however, the more immediate problem is the high-dimensional structure of $\mathcal{Y}$; in particular, there is little guarantee that intermediate representations trained to encapsulate $\mathbf{x}$ are easily mapped back up to higher-dimensional targets.

Our goal is to make use of intermediate representations that are both *predictable* from features as well as *predictive* of targets. A target-embedding autoencoder (TEA)—versus what we shall term a feature-embedding autoencoder (FEA)—flips the model architecture around by learning an embedding of target vectors instead, which a predictor then learns a mapping into. This involves an encoder $e : \mathcal{Y} \to \mathcal{Z}$, an upstream predictor $u : \mathcal{X} \to \mathcal{Z}$, and a shared forward model $\theta : \mathcal{Z} \to \mathcal{Y}$. Denote $\tilde{\mathbf{y}} = \theta(e(\mathbf{y}))$ and $\hat{\mathbf{y}} = \theta(u(\mathbf{x}))$; the complete loss function is now of the following form,

$$L = \frac{1}{N}\sum_{n=1}^{N} \left[\ell_p(\hat{\mathbf{y}}_n, \mathbf{y}_n) + \ell_r(\tilde{\mathbf{y}}_n, \mathbf{y}_n)\right] \tag{2}$$

Abstractly, the general idea of target space reduction is not new; in particular, it has been present in various solutions in the domain of multi-label classification (see Section 4 and Appendix B for discussions of related work). Here we focus on target-embedding *autoencoders*; they leverage the assumption that variations in (high-dimensional) target space are driven by a compact and predictable set of factors. By construction, learning to reconstruct directly in output space ensures that latent representations are predictive of targets; at the same time, jointly training with the prediction loss ensures that latent representations are predictable from features. Instead of learning representations for mapping *out of* (downstream), here we learn representations for mapping *into* (upstream); the shared forward model handles the rest. See Figure 1 for high-level diagrams of TEAs versus FEAs.

**Training and Inference**. Figure 2 gives block diagrams of component functions and objectives in (a) FEAs and (b) TEAs during training (see Algorithm 1 in Appendix C for pseudocode). Training occurs in three stages. First, the autoencoder is trained (to learn representations): the parameters of $e$ and $\theta$ are learned on the reconstruction loss. Second, the prediction arm is trained to regress the learned

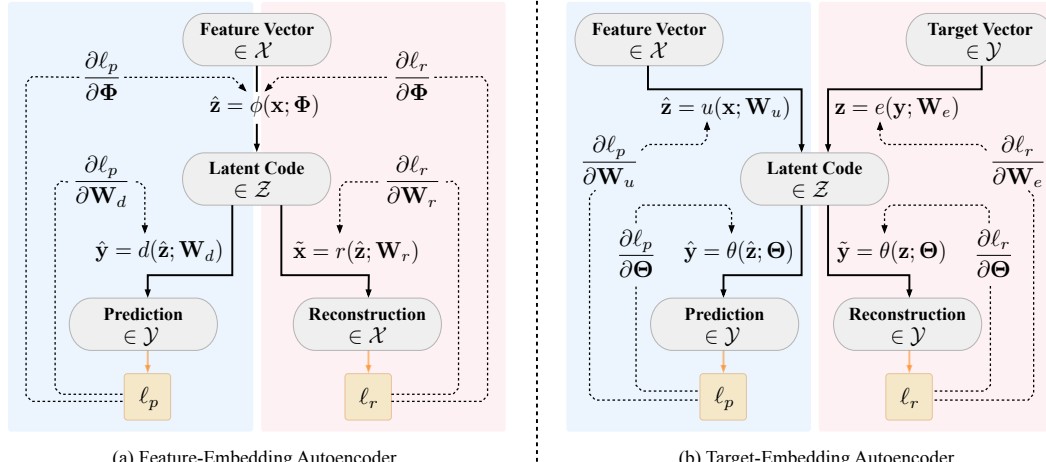

(a) Feature-Embedding Autoencoder        (b) Target-Embedding Autoencoder

Figure 2: Functions and objectives in (a) FEAs and (b) TEAs. Blue and red identify supervised and representation learning components. FEAs are parameterized by $(\boldsymbol{\Phi}, \mathbf{W}_d, \mathbf{W}_r)$ of $(\phi, d, r)$, and TEAs by $(\boldsymbol{\Theta}, \mathbf{W}_u, \mathbf{W}_e)$ of $(\theta, u, e)$. Solid lines indicate forward propagation of data; dashed lines indicate backpropagation of gradients.

embeddings (generated by the encoder): the parameters of $u$ are learned (on the latent loss) while the autoencoder is frozen. Finally, all three components are jointly trained on both prediction and reconstruction losses (Equation 2): parameters of the predictor, embedding, and shared forward model are trained simultaneously. Note that during training, the forward model receives two types of latents as input: encodings of true targets, as well as encodings predicted from features. At inference time, the target-embedding arm is dropped, leaving the learned hypothesis $h = \theta \circ u$ for prediction. Figure 4 (Appendix C) provides step-by-step block diagrams of both training and inference in greater detail.

We emphasize that TEAs—as is the case with FEAs—specify a general framework independent of the implementation details of each component. For instance, the solutions to applications in Yu et al. (2014), Yeh et al. (2017), and Girdhar et al. (2016) can be abstractly regarded as linear and nonlinear instances of this framework, with domain-specific architectures (see Section 4 and Appendix B for more detailed discussions). Linear TEAs—which we study in greater detail in Section 3—involve parameterizations $(\boldsymbol{\Theta}, \mathbf{W}_u, \mathbf{W}_e)$ of $(\theta, u, e)$ consisting of single hidden layers with linear activation.

## 3    STABILITY-BASED LEARNING GUARANTEE

Two questions are outstanding. The first is theoretical. We are motivated by the prior that variations in target space are driven by a lower-dimensional set of underlying factors. In this context, can we say something more rigorous about the benefit of TEAs? In this section, we take the first step in showing that jointly learning target representations improves generalization performance in the supervised setting. Specifically, we demonstrate that linear TEAs are characterized by uniform stability, from which theoretical guarantees are known to follow. The second question is empirical. We noted above that certain applications of label-embedding to classification can be interpreted through this framework. Does the benefit extend beyond its static, feedforward instantiations—into the temporal setting for multi-variate sequence forecasting, with both continuous and discrete targets? In Section 5, we first validate our theoretical findings with linear models and sensitivities, as well as extending our empirical analysis to the realm of recurrent, nonlinear models for both regression and classification.

Consider a linear TEA, where the upstream predictor is parameterized by $\mathbf{W}_u \in \mathbb{R}^{|\mathcal{Z}| \times |\mathcal{X}|}$, target-embedding by $\mathbf{W}_e \in \mathbb{R}^{|\mathcal{Z}| \times |\mathcal{Y}|}$, and shared forward model by $\boldsymbol{\Theta} \in \mathbb{R}^{|\mathcal{Y}| \times |\mathcal{Z}|}$, where $|\mathcal{Z}| < |\mathcal{Y}|$. The complete loss function is given by $L = \frac{1}{N} \sum_{n=1}^{N} [\ell_p(\boldsymbol{\Theta}\mathbf{W}_u\mathbf{x}_n, \mathbf{y}_n) + \ell_r(\boldsymbol{\Theta}\mathbf{W}_e\mathbf{y}_n, \mathbf{y}_n)]$ following Equation 2. Interpreting the jointly learned autoencoding component as an auxiliary task, we show that the TEA algorithm for learning the shared forward model $\boldsymbol{\Theta}$ is uniformly stable with respect to the domain of the supervised prediction task. To establish our notation, first recall the following:

**Definition 1 (Generalization Bound)** *Given a learning algorithm $\mathcal{D} \mapsto h_{\mathcal{D}}$ that returns hypothesis $h_{\mathcal{D}}$, let $R(h_{\mathcal{D}}) = \int \ell(h_{\mathcal{D}}(\mathbf{x}), \mathbf{y}) d\mu(\mathbf{x}, \mathbf{y})$ denote the risk, and $\hat{R}(h_{\mathcal{D}}) = \frac{1}{N} \sum_{n=1}^{N} \ell(h_{\mathcal{D}}(\mathbf{x}_n), \mathbf{y}_n)$ denote the empirical risk, where $\ell$ is some loss function. A generalization bound is a probabilistic bound on the defect that takes the following form: $R(h_{\mathcal{D}}) - \hat{R}(h_{\mathcal{D}}) \leq \epsilon$ with some confidence $1 - \delta$.*

**Definition 2 (Uniform Stability)** *Let $\mathcal{D}^i$ denote a modification of batch $\mathcal{D}$ where the $i$-th training instance $(\mathbf{x}_i, \mathbf{y}_i)$ is replaced by an independent and identically distributed example $(\mathbf{x}_i', \mathbf{y}_i')$. A learning algorithm is said to be $\gamma$-uniformly stable with respect to the loss function $\ell$ if $\forall \mathcal{D} \in (\mathcal{X} \times \mathcal{Y})^N, \forall i \in \{1, ..., n\}, \forall(\mathbf{x}, \mathbf{y}), (\mathbf{x}_i', \mathbf{y}_i') \in \mathcal{X} \times \mathcal{Y} : |\ell(h_\mathcal{D}(\mathbf{x}), \mathbf{y}) - \ell(h_{\mathcal{D}^i}(\mathbf{x}), \mathbf{y})| \leq \gamma$. Uniform stability holds if the minimum value of $\gamma$ converges to zero as batch size $N$ increases without limit.*

Uniform stability can be used to derive algorithm-dependent generalization bounds. In particular, Bousquet & Elisseeff (2002) first showed that the defect $\epsilon$ of a $\gamma$-uniformly stable algorithm is is less than $O((\gamma + 1/N)\sqrt{N \log(1/\delta)})$ with probability $\geq 1 - \delta$. Feldman & Vondrak (2018) recently demonstrated an improved bound of $O(\sqrt{(\gamma + 1/N) \log(1/\delta)})$. Here, we show uniform stability for linear TEAs, where $\gamma$ is $O(1/N)$—by which a tight generalization bound follows immediately. Before we begin, we introduce two additional tools: $c$-strong convexity and $\sigma$-admissibility. Note that these conditions are standard and easily satisfied—for instance, by the quadratic loss function; for more context see for example Bousquet & Elisseeff (2002), Liu et al. (2016), and Mohri et al. (2018).

**Definition 3 ($c$-Strong Convexity)** *Differentiable loss function $\ell$ is $c$-strongly convex if $\forall h, h' \in \mathcal{H} : \langle h'(\mathbf{x}) - h(\mathbf{x}), \nabla\ell(h'(\mathbf{x}), \mathbf{y}) - \nabla\ell(h(\mathbf{x}), \mathbf{y})\rangle \geq c\|h'(\mathbf{x}) - h(\mathbf{x})\|_2^2$ for some $c \in \mathbb{R}^+$, where $\nabla\ell(h(\mathbf{x}), \mathbf{y})$ denotes the gradient with respect to $h(\mathbf{x})$, and $\langle \cdot, \cdot \rangle$ denotes the dot product operation.*

**Definition 4 ($\sigma$-Admissibility)** *Loss function $\ell$ is $\sigma$-admissible with respect to the underlying hypothesis class $\mathcal{H}$ if $\forall h, h' \in \mathcal{H} : |\ell(h'(\mathbf{x}), \mathbf{y}) - \ell(h(\mathbf{x}), \mathbf{y})| \leq \sigma\|h'(\mathbf{x}) - h(\mathbf{x})\|_2$ for some $\sigma \in \mathbb{R}^+$.*

To obtain uniform stability, we make two assumptions—both analogous to prior work arguing from the benefit of learning shared models between tasks. Liu et al. (2016) deals with learning multiple tasks in general, and Le et al. (2018) deals with reconstructing inputs in what we describe as FEAs. Now in multi-task learning, the separate tasks are usually chosen due to some prior relationship between them. In the case of Assumption 1 in Liu et al. (2016) and Assumption 5 in Le et al. (2018), this is assumed to come from similarities in feature structures across tasks; hence their assumptions of cross-representability are made in feature space. (Note that this restricts primary and auxiliary features to be elements of the same space). Our setting is contrary: the inputs to primary and auxiliary tasks come from different spaces, but are trained to produce similar labels through a compact, shared latent space; hence our assumption of cross-representability will be made in this latent space instead.

**Assumption 1 (Representative Vectors)** *There exists a representative subset of target vectors $B = \{\mathbf{b}_1, ..., \mathbf{b}_M\} \subset \{\mathbf{y}_1, ..., \mathbf{y}_N\}$ such that the latent representation of any individual $(\mathbf{x}, \mathbf{y})$ can be linearly reconstructed from that of the representative subset with small error, i.e. $\mathbf{W}_p\mathbf{x} = \sum_{m=1}^M \alpha_m \mathbf{W}_e \mathbf{b}_m + \eta$ and $\mathbf{W}_e\mathbf{y} = \sum_{m=1}^M \beta_m \mathbf{W}_e \mathbf{b}_m + \eta$ for some coefficients $\alpha_m, \beta_m \in \mathbb{R}$, where $\Sigma_{m=1}^M \alpha_m^2 \leq r_\alpha^2$, $\Sigma_{m=1}^M \beta_m^2 \leq r_\beta^2$ for some $r_\alpha, r_\beta \in \mathbb{R}^+$, and $\eta$ is a small error satisfying $\|\eta\|_2 \leq \varepsilon$.*

**Remark 1** This assumption is comparatively mild, even for $\varepsilon = 0$. Note that in Liu et al. (2016) the features for separate tasks come from *different* examples in general, and the similarity of their distributions within $\mathcal{X}$ is simply assumed. Here, each pair of inputs to the prediction and reconstruction tasks comes from the *same* instance, and similarity within $\mathcal{Z}$ is explicitly enforced through the (joint) training objective. In addition, observe that the assumption will hold with zero error as long as the number of independent latent vectors is at least $|\mathcal{Z}|$. Furthermore, unlike the scenarios in Liu et al. (2016) and Le et al. (2018) we do not require that the input domains of the two tasks be identical. Therefore for ease of exposition, we assume going forward that $\varepsilon = 0$ (see Remark 6 in Appendix A).

**Remark 2** A comparison with Assumption 4 in Le et al. (2018) sheds additional light on why we expect TEAs to be beneficial where $|\mathcal{Y}| > |\mathcal{Z}|$, in contrast with the (more typical) scenario $|\mathcal{X}| > |\mathcal{Z}| \gg |\mathcal{Y}|$. Critically, the technique in Le et al. (2018) banks on the fact that the prediction arm projects the latent into a *lower-dimensional* target space. Conversely, Assumption 1 here relies on the fact that the encoding arm maps into the latent space from a *higher-dimensional* target space (rendering cross-representability therein reasonable). The distinction is crucial: we certainly do not expect any benefit from autoencoding trivially low-dimensional vectors! Note also that here the representative vectors are taken from $\mathcal{Y}$; to take them from $\mathcal{X}$ instead would be unreasonable. For any compressive autoencoder, we generally expect if some subset $\{\mathbf{b}_1, ..., \mathbf{b}_M\} \subset \{\mathbf{y}_1, ..., \mathbf{y}_N\}$ spans $\mathcal{Y}$ that $\{\mathbf{W}_e\mathbf{b}_1, ..., \mathbf{W}_e\mathbf{b}_M\}$ then also span $\mathcal{Z}$ in order to be maximally reconstructive. The same cannot be said of subsets $\{\mathbf{c}_1, ..., \mathbf{c}_M\} \subset \{\mathbf{x}_1, ..., \mathbf{x}_N\}$ that span $\mathcal{X}$—for instance, take $|\mathcal{X}| \leq |\mathcal{Z}|$.

In addition to being representative in terms of latent values, the set of representative points also needs to be representative in terms of the reconstruction error. First, let $L'$ denote the counterpart

to $L$ where the $i$-th sample $(\mathbf{x}_i, \mathbf{y}_i)$ is replaced by some new instance $(\mathbf{x}'_i, \mathbf{y}'_i) \in \mathcal{X} \times \mathcal{Y}$; that is, $L' = \frac{1}{N}[\ell_p(\Theta\mathbf{W}_u\mathbf{x}'_i, \mathbf{y}'_i) + \ell_r(\Theta\mathbf{W}_e\mathbf{y}'_i, \mathbf{y}'_i) + \sum_{n=1;n\neq i}^{N} (\ell_p(\Theta\mathbf{W}_u\mathbf{x}_n, \mathbf{y}_n) + \ell_r(\Theta\mathbf{W}_e\mathbf{y}_n, \mathbf{y}_n))]$. Then, let $\Theta_*$, $\Theta'_*$ denote the optimal parameters corresponding to the two losses $L$ and $L'$.

**Assumption 2 (Representative Errors)** *Let $L'_r$ contain the reconstruction errors of the dataset without the $i$-th sample: $L'_r = (1/N)\Sigma_{n=1;n\neq i}^{N}\ell_r(\Theta\mathbf{W}_e\mathbf{y}_n, \mathbf{y}_n)$, and let $L_r^B$ denote the reconstruction error of the representative subset: $L_r^B = (1/M)\Sigma_{m=1}^{M}\ell_r(\Theta\mathbf{W}_e\mathbf{b}_m, \mathbf{b}_m)$. Then there exists some $a > 0$ such that for any small $\kappa > 0 : L_r^B(\Theta_*) - L_r^B(\kappa\Theta'_* + (1-\kappa)\Theta_*) + L_r^B(\Theta'_*) - L_r^B(\kappa\Theta_* + (1-\kappa)\Theta'_*) \leq a\left[L'_r(\Theta_*) - L'_r(\kappa\Theta'_* + (1-\kappa)\Theta_*)\right] + a\left[L'_r(\Theta'_*) - L'_r(\kappa\Theta_* + (1-\kappa)\Theta'_*)\right]$.*

That is, the difference in reconstruction error $L_r^B$ between the two points $\Theta_*$, $\Theta'_*$ is upper bounded by some constant factor of the corresponding difference in reconstruction error $L'_r$ at the two points. Importantly, note that this does *not* require that the values of the errors $L'_r$ and $L_r^B$ themselves be similar, only that their differences be similar. This assumption is identical to Assumption 6 in Le et al. (2018), and plays an identical role: we make use of $L_r^B$—which is only dependent on $M$—to allow the bound to decay with $N$; this is in contrast with the generic multi-task analysis of Liu et al. (2016), which—if applied directly to TEAs (as with FEAs)—would give a bound that does not decay with $N$.

**Theorem 1 (Uniform Stability)** *Let $\ell_p$ and $\ell_r$ be $\sigma_p$-admissible and $\sigma_r$-admissible loss functions, and let $\ell_r$ be $c$-strongly convex. Then under Assumptions 1 and 2, the following inequality holds,*

$$|\ell_p(\Theta'_*\mathbf{W}_u\mathbf{x}, \mathbf{y}) - \ell_p(\Theta_*\mathbf{W}_u\mathbf{x}, \mathbf{y})| \leq \frac{2(\sigma_p^2 r_\alpha^2 + \sigma_p\sigma_r r_\alpha r_\beta)aM}{cN}$$

*Proof.* Appendix A.

**Corollary 1 (Generalization Bound)** *Consider the same conditions as in Theorem 1; that is, let $\ell_p$ and $\ell_r$ be $\sigma_p$-admissible and $\sigma_r$-admissible losses, and let $\ell_r$ be $c$-strongly convex. Then under Assumptions 1 and 2, the defect $\epsilon$ is less than $O(\sqrt{(1/N)\log(1/\delta)})$ with probability at least $1 - \delta$.*

*Proof.* Follows immediately from Theorem 1 (above) and either of the following (similar results hold for both): Theorem 1.2 (Feldman & Vondrak, 2018), and Theorem 12 (Bousquet & Elisseeff, 2002).

In supervised learning, it is often easy to make an argument—on an intuitive level—for the "regularizing" effect of additional loss terms. In contrast, this analysis allows us to unambiguously identify and quantify the benefit of the embedding component as a regularizer (see Remark 4 in Appendix A).

**Remark 3** In the linear label space reduction framework of Yu et al. (2014), uniform convergence is also shown to hold via norm-based regularization. Specifically for uniform stability, a similar bound can also be achieved by adding a strongly convex term to the objective, such as Tikhonov—and $\ell_2$—regularization (Bousquet & Elisseeff, 2002; Rosasco & Poggio, 2009; Shalev-Shwartz et al., 2010). Here, however, the joint reconstruction task leverages a different kind of bias—precisely, the assumption that there exist compact and predictable representations of targets. Therefore the significance of this analysis is that we achieve an equivalent result *independent of* explicit regularization.

## 4 RELATED WORK

Our work straddles three threads of research: (1) supervised representation learning with autoencoders, (2) label space reduction for multi-label classification, and (3) stability-based learning guarantees. Appendix B provides a much expanded treatment, and presents summary tables for additional context.

**Supervised representation learning.** While a great deal of research is devoted to uncovering useful underlying structures of data through autoencoders—with various properties such as sparsity (Ranzato et al., 2007) and disentanglement (Chen et al., 2018), among many others (Tschannen et al., 2018)—the goal of better representations is often for the benefit of downstream tasks. Semi-supervised autoencoders *jointly optimized* on partially-labeled data can obtain compact representations that improve prediction (Weston et al., 2012; Kingma et al., 2014; Ghifary et al., 2016). Furthermore, auxiliary reconstruction is also useful in a *purely supervised* setting: rather than focusing on how specific architectures better structure unlabeled data, Le et al. (2018) show the simple benefit of feature-reconstruction on supervised classification—a special case of what we describe as FEAs.

In contrast, we focus on *target*-representation learning in the supervised setting, and analyze its benefit under the prior that high-dimensional targets are driven by a compact and predictable set of factors.

We take inspiration from the empirical study of Girdhar et al. (2016), where latent representations of 3D objects are jointly trained to be predictable from 2D images. Their setup can be viewed as a specific instance of TEAs with (nonlinear) convolutional components, with a minor variation in training: in the joint stage, predictors continue to regress the learned embeddings, and gradients only backpropagate from latent space (instead of target space). Unlike the symmetry of our losses (which we require for our analysis above), their common decoder is only shared *indirectly* (and predictions made indirectly). As it turns out, this does not appear to matter for performance (see Section 5). In Mostajabi et al. (2018), a two-stage procedure is used for semantic segmentation—loosely comparable to the first two stages in TEAs; in contrast to our emphasis on joint training, they study the benefit of a *frozen* embedding branch in parallel with direct prediction. More broadly related to target-embedding, Dalca et al. (2018) build anatomical priors for biomedical segmentation in *unsupervised* settings.

**Multi-label classification**. The general idea of target space dimension reduction has been explored for multi-label classification problems (commonly, annotation based on bags of features). These first derive a reduced label space, then *subsequently* associate inputs to it; methods include compressed sensing (Hsu et al., 2009), principal components (Tai & Lin, 2010), maximum-margin coding (Zhang & Schneider, 2012), and landmarking (Balasubramanian & Lebanon, 2012). Closer to our theme of *joint learning*, Chen & Lin (2012) first propose simultaneously minimizing encoding and prediction errors via an SVD formulation. Using generic empirical risk minimization, Yu et al. (2014) formulate the problem as a linear model with a low-rank constraint. While this captures an intuition (similar to ours) of restricted latent factors, their error bounds require norm-based regularization (unlike ours).

Recently, Yeh et al. (2017) generalize the label-embedding approach to *autoencoders*. This flexibly accommodates custom losses to exploit correlations, as well as deep learning for nonlinearities. Our work is related to this line of research, although we operate at a higher level of abstraction, with a significant difference in focus. Their problem is multi-label classification, and their starting point is binary relevance (i.e. label by label). During reduction, they worry about specific losses that capture dependencies within and among spaces. In contrast, we worry about autoencoding *at all*—that is, we focus on the effect of joint reconstruction on learning the prediction model. Problems can be of any form: classification or regression, and our starting point is direct prediction (i.e. no reconstruction).

**Stability and learning guarantees**. Generalizability via hypothesis stability is first studied in Rogers & Wagner (1978) and Devroye & Wagner (1979); unlike arguments based on the complexity of the search space (Vapnik & Chervonenkis, 1971; Pollard, 1984; Koltchinskii, 2001), these account for how the algorithm depends on the data. Bousquet & Elisseeff (2002) first formalize the notion of *uniform stability* sufficient for learnability, and Feldman & Vondrak (2018) use ideas related to differential privacy (Bassily et al., 2016) for further improvement. Separately, while there is a wealth of research on dimensionality reduction and autoencoders (Singh et al., 2009; Mohri et al., 2015; Gottlieb et al., 2016; Epstein & Meir, 2019), they either operate in the semi-supervised setting, or focus on the benefit of feature representations (not targets) and also do not consider joint learning.

The benefit of *jointly* learning multiple tasks through a common operator (Caruana, 1997) is explored with VC-based (Baxter, 2000) and Rademacher complexity-based (Maurer, 2006; Maurer et al., 2016) analyses. Recently, Liu et al. (2016) show that the algorithm for learning the shared model in a multi-task setting is uniformly stable. While our argument is based on theirs, we are not interested in a generic bound for all tasks; closer to Le et al. (2018), we focus on the primary prediction task, and leverage the auxiliary reconstruction task for stability. Similarly, we arrive at an $O(1/N)$ on instability without an explicit regularization term as in Bousquet & Elisseeff (2002). Unlike them, however, the fundamental distinction of our setting is that $\mathcal{Y}$ is high-dimensional (but where the underlying factors are assumed compact); in this sense our focus is the mirror opposite of theirs.

## 5 EXPERIMENTS AND DISCUSSION

So far, we have formalized a general target-autoencoding framework for supervised learning, and quantified the benefit via uniform stability. Our overall goal in this section is to explore this benefit in a simple controlled setting, such that we can identify and isolate its utility on the prediction task, and investigate any sensitivities of interest. By way of preface, we emphasize two observations from above: **(1)** In the *static, multi-label classification* setting, the gain from label-embedding has been studied, including the autoencoder approach of Yeh et al. (2017)—which can be viewed as an instantiation of TEAs with sophisticated refinements. **(2)** The benefit of target-autoencoding is also

Table 1: Dataset statistics and input/output dimensions used in experiments

| Dataset | Num. patients | Samp. freq. | Target type | Static dim. | Temp. dim. (history) | Temp. dim. (forecast) | Window (history) | Window (forecast) | Effective $|\mathcal{X}|$ | Effective $|\mathcal{Y}|$ |
|---|---|---|---|---|---|---|---|---|---|---|
| UKCF | 10,000 | 1 yr. | Binary | 11 | 43 | 34 | 3 | 4 | 140 | 136 |
| ADNI | 1,700 | 6 m. | Continuous | 11 | 26 | 24 | 4 | 8 | 115 | 192 |
| MIMIC | 22,000 | 4 hr. | Mixed | 26 | 361 | 361 | 5 | 5 | 1,831 | 1,805 |

The effective input dimension $|\mathcal{X}|$ is computed as the dimension of static data plus the product of the width of the historical window (of temporal information) with its dimension; the effective target dimension $|\mathcal{Y}|$ is similarly computed as the product of the width of the forecast window (of temporal information) with its dimension.

demonstrated using *nonlinear, convolutional architectures* in Girdhar et al. (2016)—which is also an instantiation of TEAs, also noting significant gains. Therefore a natural question of interest is:

- Does the utility of target-embedding extend to (nonlinear) recurrent models with sequential data for general, high-dimensional targets (i.e. regression and/or classification)?

**Disease Trajectories**. In this section, we take the first step in answering this question—as our empirical contribution, we extend validation of target-embedding autoencoders to the domain of *multivariate sequence forecasting*, exploring its utility on linear *and* nonlinear sequence-to-sequence architectures. What makes a good testbed? In particular, the progression of diseases (and their markers) is high-dimensional in presentation; at the same time, their evolution is often driven by latent biological dynamics (Szczesniak et al., 2017; Pascoal et al., 2017; Alaa & van der Schaar, 2019). With the increasing importance of early diagnosis and timely intervention in healthcare, the ability to forecast individual disease trajectories ($\mathcal{Y}$) in the presence of limited windows of information ($\mathcal{X}$) has become increasingly desirable (Donohue et al., 2014; Pham et al., 2017; Bhagwat et al., 2018).

**Datasets**. We use three datasets in our experiments. The first consists of a cohort of patients enrolled in the UK Cystic Fibrosis registry (**UKCF**), which records follow-up trajectories for over 10,000 patients. We are interested in forecasting future trajectories for the 11 possible infections and 23 possible comorbidities (all binary variables) recorded at each follow-up, using past trajectories and basic demographics as input. The second consists of patients in the Alzheimer's Disease Neuroimaging Initiative study (**ADNI**), which tracks disease progression for over 1,700 patients. We are interested in forecasting the evolution of the 8 primary biomarkers and 16 cognitive tests (all continuous variables) measured at each visit, using past measurements and basic demographics as input. The third consists of a cohort of patients in intensive care units from the Medical Information Mart for Intensive Care (**MIMIC**), which records physiological data streams for over 22,000 patients. Likewise, we are interested in forecasting future trajectories for the 361 most frequently measured variables such as vital signs and lab tests (both binary and continuous variables), again using past measurements and basic demographics as input. See Appendix D for more information on datasets.

**Experimental Setup**. In each instance, the prediction input is a precedent window of up to width $w_x$, and the prediction target is the succedent window of width $w_y$. For UKCF $(w_x, w_y) = (3, 4)$ at 1-year resolution, for ADNI $(4, 8)$ at 6-month resolution, and for MIMIC $(5, 5)$ at 4-hour (resampled) resolution. All models are implemented in Tensorflow. Linear models consist of a single hidden layer with no nonlinearity; for the nonlinear case, we implement an RNN model for each component using GRUs. See Appendix D for additional detail on model implementation and configuration. For evaluation, we measure the mean squared error (MSE) for continuous targets (averaged across variables), and the area under the precision-recall curve (PRC) and area under the receiver operating characteristic curve (ROC) for binary targets (averaged across variables). We use cross-validation on the training set for hyperparameter tuning, selecting the setting that gives the lowest validation loss averaged across folds. For each model and dataset, we report the average and standard error of each performance metric across 10 different experiment runs, each with a different random train-test split.

Note that forecasting high-dimensional disease trajectories is challenging, and input information is *deliberately* limited (as is often the case in medical practice); the desired targets are similar or higher-dimensional than the inputs (see Table 1). This obviously results in an inherently difficult prediction problem—but which makes for a good candidate setting to test the utility of target-representation learning. RNN autoencoders have previously been proposed for learning representations of *inputs* (i.e. FEAs instantiated with RNNs) to improve classification (Dai & Le, 2015), prediction (Lyu et al., 2018), generation (Srivastava et al., 2015), and clustering (Baytas et al., 2017); similarly, their mission is not in excessively optimizing specific architectural novelties to match state-of-the-art, but rather in exploring the benefit of the autoencoding framework. Here, we learn representations of *targets*.

Table 2: Summary results for TEA and comparators on linear model with UKCF (Bold indicates best)

|  | Base | REG | FEA | TEA | F/TEA |
|---|---|---|---|---|---|
| PRC(I) | $0.322 \pm 0.099$* | $0.347 \pm 0.085$* | $0.351 \pm 0.079$* | $\mathbf{0.450 \pm 0.035}$ | $0.414 \pm 0.028$* |
| PRC(C) | $0.416 \pm 0.100$* | $0.433 \pm 0.083$* | $0.455 \pm 0.087$* | $\mathbf{0.559 \pm 0.060}$ | $0.520 \pm 0.052$ |
| ROC(I) | $0.689 \pm 0.089$* | $0.710 \pm 0.072$* | $0.720 \pm 0.073$ | $\mathbf{0.767 \pm 0.026}$ | $0.766 \pm 0.023$ |
| ROC(C) | $0.679 \pm 0.091$* | $0.700 \pm 0.075$* | $0.713 \pm 0.075$ | $\mathbf{0.767 \pm 0.042}$ | $0.755 \pm 0.037$ |

The two-sample $t$-test for difference in means is conducted on the results. An asterisk indicates statistically significant difference in means ($p$-value $< 0.05$) relative to the TEA result. PRC and ROC metrics are reported separately for variables representing infections (I) and comorbidities (C). See Tables 9–10 for extended results.

Table 3: Summary results for TEA and comparators on nonlinear (RNN) model (Bold indicates best)

|  | UKCF (Binary Targets) | | ADNI (Continuous Targets) | | MIMIC (Mixed Targets) | |
|---|---|---|---|---|---|---|
|  | PRC(I) | PRC(C) | MSE(B) | MSE(C) | PRC | MSE |
| Base | $0.411 \pm 0.035$* | $0.497 \pm 0.057$* | $0.105 \pm 0.018$* | $0.361 \pm 0.064$ | $0.142 \pm 0.028$* | $0.153 \pm 0.011$ |
| REG | $0.415 \pm 0.030$* | $0.518 \pm 0.052$* | $0.096 \pm 0.014$* | $0.360 \pm 0.066$ | $0.143 \pm 0.019$* | $0.152 \pm 0.010$ |
| FEA | $0.410 \pm 0.033$* | $0.521 \pm 0.054$* | $0.092 \pm 0.012$* | $0.356 \pm 0.068$ | $0.144 \pm 0.030$* | $0.152 \pm 0.012$ |
| TEA | $\mathbf{0.483 \pm 0.045}$ | $\mathbf{0.583 \pm 0.072}$ | $\mathbf{0.063 \pm 0.010}$ | $\mathbf{0.330 \pm 0.066}$ | $\mathbf{0.239 \pm 0.039}$ | $\mathbf{0.150 \pm 0.012}$ |
| F/TEA | $0.457 \pm 0.037$ | $0.576 \pm 0.071$ | $0.073 \pm 0.010$* | $0.338 \pm 0.067$ | $0.166 \pm 0.023$* | $0.154 \pm 0.011$ |

The two-sample $t$-test for difference in means is conducted on the results. An asterisk indicates statistically significant difference in means ($p$-value $< 0.05$) relative to the TEA result. For UKCF, only PRC metrics for infections (I) and comorbidities (C) are shown due to space limitation; for ADNI, MSE metrics are reported separately for targets representing biomarkers (B) and cognitive tests (C). See Tables 13–14, 17–18, and 21–22.

## 5.1 MAIN RESULTS

**Overall Benefit**. First, we examine the overall utility of TEAs. To verify the *linear* case first, Table 2 summarizes the performance of TEA and alternate setups on UKCF. The temporal axis is flattened to simulate ordinary static multi-label classification, and the base case is direct prediction (Base)—that is, absent any auxiliary representation learning or regularization. Next, we allow for $\ell_2$-regularization over direct prediction (REG), as well as over all other methods. FEAs differ only by the added feature-reconstruction, and TEAs only by the target-reconstruction; as an additional sensitivity, we also implement a combined approach (F/TEA). More generally, we also wish to examine the benefit of TEA for the *nonlinear* case: Table 3 summarizes analogous results where component functions are implemented with GRU networks; results are shown for all datasets. Ceteris paribus, we observe that target-representation learning has a notable positive impact on performance. Interestingly, learning representations of inputs does not yield significant benefit, and the hybrid approach (F/TEA) is worse than TEA; this suggests that forcing intermediate representation to encode both features and targets may be overly constraining. (Note that for the linear model, the instances are restricted to those for which the full input window is available; as a consequence, the results for linear and nonlinear cases are not directly comparable). Figures 4 (Appendix C) and 5 (Appendix D) give training diagrams for all comparators. Additional experiment results (by model, timestep, metric) are in Appendix E.1-2.

**Source of Gain**. There are two (related) interpretations of TEAs. First, we studied the *regularization* view in Section 3; this concerns the benefit of joint training using both prediction and reconstruction losses. Ceteris paribus, we expect performance to improve purely by dint of the jointly trained TEA objective. Second, the *reduction* view says that TEAs decompose the (difficult) prediction problem into two (smaller) tasks: the autoencoder learns a compact representation $\mathbf{z}$ of $\mathbf{y}$, and the predictor learns to map $\mathbf{x}$ to $\mathbf{z}$. This makes the potential benefit of staged training (Section 2 and Appendix C) intuitively clear, and suggests an alternative—that of simply training the autoencoder and predictor arms in two stages—à la Mostajabi et al. (2018). As a general framework, TEAs is a combination of both ideas: all three components are jointly trained in a third stage—à la Girdhar et al. (2016). We now account for the improvement in performance due to these two sources of benefit; Table 4 does so for the linear case (on UKCF), and Table 5 for the more general nonlinear case (on all datasets). The "No Joint" setting isolates the benefit from staged training only. This is analogous to basic unsupervised pretraining (though using targets), and corresponds to omitting the final joint training stage in Algorithm 1. The "No Staged" setting isolates the benefit from joint training only (without pretraining the autoencoder or predictor), and corresponds to omitting the first two training stages in Algorithm 1. The "Neither" setting is equivalent to vanilla prediction (REG) without leveraging either of the advantages. We observe that while both sources of benefit are individually important, neither setting performs quite as well as when both are combined. See Appendix E.1-2 for extended results.

Table 4: Summary source of gain and TEA variants on linear model with UKCF (Bold indicates best)

|  | Neither | No Joint | No Staged | TEA | TEA(L) | TEA(LP) |
|---|---|---|---|---|---|---|
| PRC(I) | $0.347 \pm 0.085$ | $0.402 \pm 0.026$ | $0.431 \pm 0.031$ | $0.450 \pm 0.035$ | $0.435 \pm 0.031$ | $\textbf{0.454} \pm \textbf{0.036}$ |
| PRC(C) | $0.433 \pm 0.083$ | $0.507 \pm 0.040$ | $0.543 \pm 0.054$ | $0.559 \pm 0.060$ | $0.544 \pm 0.053$ | $\textbf{0.560} \pm \textbf{0.061}$ |
| ROC(I) | $0.710 \pm 0.072$ | $0.747 \pm 0.022$ | $0.764 \pm 0.022$ | $0.767 \pm 0.026$ | $0.759 \pm 0.025$ | $\textbf{0.768} \pm \textbf{0.028}$ |
| ROC(C) | $0.700 \pm 0.075$ | $0.744 \pm 0.038$ | $0.766 \pm 0.038$ | $\textbf{0.767} \pm \textbf{0.042}$ | $0.760 \pm 0.042$ | $\textbf{0.767} \pm \textbf{0.042}$ |

"No Joint" omits final joint training, and "No Staged" skips (pre)-training stages. PRC and ROC metrics are reported separately for targets representing infections (I) & comorbidities (C). See Tables 11–12 for extended results.

Table 5: Summary source of gain and TEA variants on nonlinear (RNN) model (Bold indicates best)

|  | UKCF (Binary Targets) | | ADNI (Continuous Targets) | | MIMIC (Mixed Targets) | |
|---|---|---|---|---|---|---|
|  | PRC(I) | PRC(C) | MSE(B) | MSE(C) | PRC | MSE |
| Neither | $0.415 \pm 0.030$ | $0.518 \pm 0.052$ | $0.096 \pm 0.014$ | $0.360 \pm 0.066$ | $0.143 \pm 0.019$ | $0.152 \pm 0.010$ |
| No Joint | $0.455 \pm 0.039$ | $0.574 \pm 0.069$ | $0.092 \pm 0.014$ | $0.353 \pm 0.070$ | $0.183 \pm 0.038$ | $0.151 \pm 0.011$ |
| No Staged | $0.424 \pm 0.031$ | $0.543 \pm 0.061$ | $0.106 \pm 0.022$ | $0.363 \pm 0.067$ | $0.167 \pm 0.022$ | $0.150 \pm 0.012$ |
| TEA | $\textbf{0.483} \pm \textbf{0.045}$ | $\textbf{0.583} \pm \textbf{0.072}$ | $0.063 \pm 0.010$ | $0.330 \pm 0.066$ | $0.239 \pm 0.039$ | $0.150 \pm 0.012$ |
| TEA(L) | $\textbf{0.483} \pm \textbf{0.047}$ | $0.581 \pm 0.074$ | $\textbf{0.058} \pm \textbf{0.012}$ | $0.330 \pm 0.076$ | $\textbf{0.249} \pm \textbf{0.049}$ | $\textbf{0.149} \pm \textbf{0.012}$ |
| TEA(LP) | $0.480 \pm 0.044$ | $\textbf{0.583} \pm \textbf{0.072}$ | $0.064 \pm 0.012$ | $\textbf{0.329} \pm \textbf{0.068}$ | $0.229 \pm 0.039$ | $0.151 \pm 0.011$ |

"No Joint" omits final joint training, and "No Staged" skips (pre)-training stages. For UKCF, only PRC metrics for infections (I) and comorbidities (C) are shown due to space limitation; for ADNI, MSE metrics are reported separately for targets representing biomarkers (B) and cognitive tests (C). See Tables 15–16, 19–20, and 23–24.

**Variations**. Having established the utility of target-embedding, we can ask whether variations on the same theme perform similarly. In particular, the embeddings in the empirical studies of Girdhar et al. (2016) and Yeh et al. (2017) are jointly learned via the reconstruction loss $\ell_r$ and latent loss $\ell_z$—that is, the prediction arm continues to regress learned embeddings during the joint training stage (Figure 4(d), in Appendix D). The principle is similar, although (as noted in Section 4) the primary task is therefore learned *indirectly*; this is in contrast to the vanilla TEA setup, where the primary task is learned *directly* via the prediction loss $\ell_p$. Tables 4 and 5 also compare the performance of vanilla TEAs with this indirect variant (TEA(L)), as well as a hybrid variant (TEA(LP)) for which both latent and prediction losses are trained jointly with the reconstruction loss (Figure 4(e)). Perhaps as expected, we observe that performance across all three variants are more or less identical, affirming the general benefit of target-representation learning. Again, see Appendix E.1-2 for extended results.

## 5.2 SENSITIVITIES

**Regularization**. Of course, target-representation learning is not a replacement for other regularization strategies; it is an additional tool that can be used in parallel where appropriate. Figure 3(a) shows the performance of TEA and REG with various coefficients $\nu$ on $\ell_2$-regularization. By itself, introducing $\ell_2$-regularization does improve performance up to a certain point, beyond which the additional shrinkage bias incurred begins to be counterproductive; this is not surprising. Interestingly, introducing target-representation learning appears to leverage an *orthogonal* bias: it consistently improves prediction performance regardless of level of shrinkage. This is a practical result of the theoretical observation in Remark 3: while prior works obtain stability through explicit $\ell_2$-regularization, the benefit from target-embedding relies on a different bias entirely, which allows us to combine them. While increasing the strength of either form of regularization reduces variability in results (see also below), excessive bias of either alone degrades performance. See Appendix E.3 for full results.

**Strength of Prior**. Target-embedding attempts to leverage the assumption that there exist compact and predictable representations of targets. Realistically (e.g. due to measurement noise), of course, this will not hold perfectly. In our experiments, we set the ratio of prediction and reconstruction losses to be $1 : 1$ for TEA (as well as FEA and F/TEA); that is, the "strength-of-prior" coefficient $\lambda$ on $\ell_r$ is 0.5. In order to isolate the effect of $\lambda$ during joint training, we observe the performance of TEAs with joint training only (i.e. removing the confounding effect of staged training). For large values of $\lambda$, we expect the reconstruction task to dominate in priority, which is (under an imperfect prior) not beneficial for the ultimate prediction task—in general, a hidden representation that is most reconstructive is *not* necessarily also what is most predictable). For small values of $\lambda$, the setup begins to resemble direct prediction. Figure 3(b) verifies our intuition. Note that in the extreme case of $\lambda = 1$, predictions are no better than random (see ROC$\sim 0.5$). See Appendix E.3 for full results.

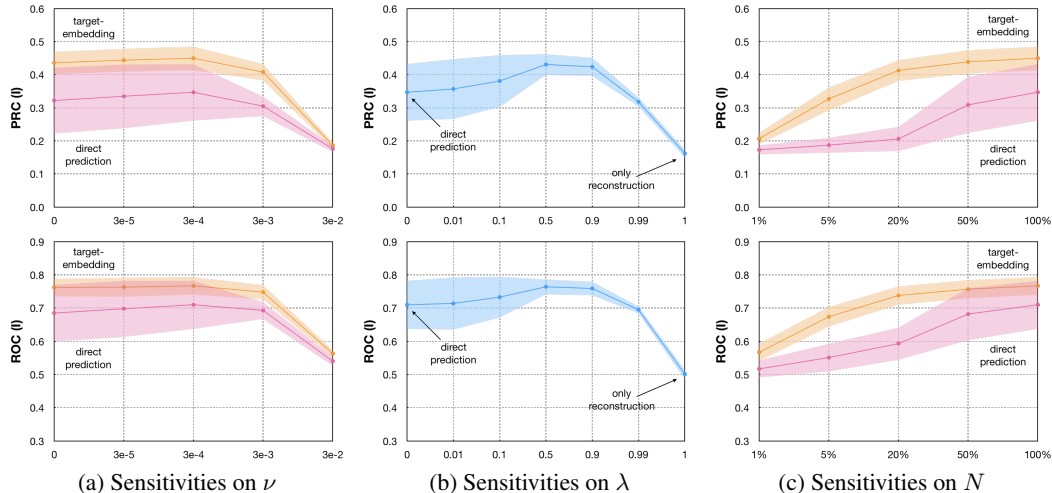

(a) Sensitivities on $\nu$     (b) Sensitivities on $\lambda$     (c) Sensitivities on $N$

Figure 3: Sensitivities on $\ell_2$-regularization coefficient $\nu$, strength-of-prior coefficient $\lambda$, and training size $N$ for direct prediction (REG) and with target-embedding (TEA) on linear model with UKCF. For sensitivities on $\lambda$, we perform joint training only, so that we isolate the effect of the joint reconstruction task (i.e. removing the confounding effect of staged training). Standard errors are indicated with shaded regions. For full results, see Tables 25–28 for sensitivities on $\nu$, Tables 29–30 for sensitivities on $\lambda$, and Tables 31–34 for sensitivites on $N$.

**Sample Complexity**. Figure 3(c) shows the performance of TEA and REG under various levels of data scarcity. The benefit conferred by TEAs is significant especially when the amount of training data $N$ is *limited*. Importantly, note that we are operating strictly within the context of supervised learning: unlike in semi-supervised settings, here we are not just restricting access to *paired* data; we are restricting access to data *per se*. (Though beyond the scope of this paper, we expect that extending TEAs to semi-supervised learning with additional unpaired data would yield larger gains). Here, without the luxury of learning from unpaired data, we highlight the comparative advantage purely from the addition of target-representation learning. Again, see Appendix E.3 for full results.

## 5.3 DISCUSSION

By way of conclusion, we emphasize the importance of our central assumption: that there exist compact and predictable representations of the (high-dimensional) targets. This is critical: target-embedding is not useful where this is not true. Now obviously, learning representations of targets is unnecessary if the output dimension is trivially small (e.g. if the target is a single classification label), or if the problem itself is trivially easy (e.g. if direct prediction is already perfect). Also obvious is the situation where representations cannot possibly be compact (e.g. if all output dimensions are independent of each other), in which case any model with a compressive (bottleneck) representation as an intermediate target may make little sense to begin with. Perhaps *less* obvious is that we cannot assume that the goals of prediction and reconstruction are always aligned. Just as in learning feature-embeddings (for downstream classification), what is most reconstructive may not necessarily encode what is most discriminative; so too in learning target-embeddings (for upstream prediction), what is most reconstructive may not necessarily encode what is most predictable. In the case of disease trajectories, it is *medical knowledge* that permits this assumption with some confidence. Appendix E.4 gives an extreme (synthetic) counterexample where this prior is outright false—i.e. prediction and reconstruction are directly at odds. While certainly contrived, it serves as a caveat about assumptions.

Using the deliberately challenging setting of disease trajectory forecasting with limited information, we have illustrated the nontrivial utility of target-representation learning in a controlled setting with baseline models. While we appreciate that component models in the wild may be more tailored, this setting better allows us to identify and isolate the potential utility of target-autoencoding *per se*. In addition to verifying our intuitions for the linear case, we have extended empirical validation of target-autoencoding to (nonlinear) sequence-to-sequence recurrent architectures; along the way, we explored the sources of gain from joint and staged training, as well as various sensitivities of interest. Where the prior holds, target-embedding autoencoders are potentially applicable to any high-dimensional prediction task beyond static classification and imaging applications, and exploring its utility for other specific domain-architectures may be a practical direction for future research.

ACKNOWLEDGMENTS

This work was supported by Alzheimer's Research UK (ARUK), the US Office of Naval Research (ONR), and the National Science Foundation (NSF): grant numbers ECCS1462245, ECCS1533983, and ECCS1407712. We thank the UK Cystic Fibrosis Trust, the Alzheimer's Disease Neuroimaging Initiative, and the MIT Lab for Computational Physiology respectively for making the UKCF, ADNI, and MIMIC datasets available for research. We thank the reviewers for their helpful comments.

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

# A PROOF OF THEOREM 1

**Definition 5 (Bregman Distance)** *Let $\ell$ be some differentiable, strictly convex loss. For any $h, h' \in \mathcal{H}$, the Bregman distance associated with $\ell$ is given by $D_\ell(h'\|h) = \ell(h') - \ell(h) - \langle h' - h, \nabla\ell(h)\rangle$. Additivity and non-negativity are easy to see; that is, $D_\ell(h'\|h) \geq 0$ for any $h, h' \in \mathcal{H}$, and if $\ell = \ell_1 + \ell_2$ and both $\ell_1$ and $\ell_2$ are convex functions, then $D_\ell(h'\|h) = D_{\ell_1}(h'\|h) + D_{\ell_2}(h'\|h)$.*

**Theorem 1 (Uniform Stability)** *Let $\ell_p$ and $\ell_r$ be $\sigma_p$-admissible and $\sigma_r$-admissible loss functions, and let $\ell_r$ be $c$-strongly convex. Then under Assumptions 1 and 2, the following inequality holds,*

$$|\ell_p(\boldsymbol{\Theta}'_*\mathbf{W}_u\mathbf{x}, \mathbf{y}) - \ell_p(\boldsymbol{\Theta}_*\mathbf{W}_u\mathbf{x}, \mathbf{y})| \leq \frac{2(\sigma_p^2 r_\alpha^2 + \sigma_p\sigma_r r_\alpha r_\beta)aM}{cN}$$

*Proof.* The overall strategy consists of three steps in sequence. First, we want to bound the delta in prediction loss under $\boldsymbol{\Theta}_*$ and $\boldsymbol{\Theta}'_*$ using the set of representative vectors $B$. Second, we bound the resulting expression in terms of the Bregman divergence of the complete loss functions $L$ and $L'$ under $\boldsymbol{\Theta}_*$ and $\boldsymbol{\Theta}'_*$. Third, we express the divergence back in terms of the original expression itself (consisting of representative vectors), which allows us to solve for a bound on that expression. Finally, combining the results from the three steps completes the proof. We begin with the left-hand term,

$$|\ell_p(\boldsymbol{\Theta}'_*\mathbf{W}_u\mathbf{x}, \mathbf{y}) - \ell_p(\boldsymbol{\Theta}_*\mathbf{W}_u\mathbf{x}, \mathbf{y})| \leq \sigma_p\|(\boldsymbol{\Theta}'_* - \boldsymbol{\Theta}_*)\mathbf{W}_u\mathbf{x}\|_2$$

$$= \sigma_p\left\|\sum_{m=1}^{M} \alpha_m(\boldsymbol{\Theta}'_* - \boldsymbol{\Theta}_*)\mathbf{W}_e\mathbf{b}_m\right\|_2 \leq \sigma_p r_\alpha \sqrt{\sum_{m=1}^{M} \|(\boldsymbol{\Theta}'_* - \boldsymbol{\Theta}_*)\mathbf{W}_e\mathbf{b}_m\|_2^2} \tag{3}$$

where the first inequality follows from the fact that $\ell_p$ is $\sigma_p$-admissible, the equality follows from Assumption 1 for some coefficients $\alpha_m \in \mathbb{R}$, and the second inequality follows from the Cauchy-Schwarz inequality. As our second step, the goal is to upper-bound the term under the square-root,

$$\sum_{m=1}^{M} \|(\boldsymbol{\Theta}'_* - \boldsymbol{\Theta}_*)\mathbf{W}_e\mathbf{b}_m\|_2^2$$

$$\leq \frac{1}{c}\sum_{m=1}^{M} \langle(\boldsymbol{\Theta}'_* - \boldsymbol{\Theta}_*)\mathbf{W}_e\mathbf{b}_m, \nabla\ell_r(\boldsymbol{\Theta}'_*\mathbf{W}_e\mathbf{b}_m, \mathbf{b}_m) - \nabla\ell_r(\boldsymbol{\Theta}_*\mathbf{W}_e\mathbf{b}_m, \mathbf{b}_m)\rangle$$

$$= \frac{1}{c}\sum_{m=1}^{M} \left\langle\boldsymbol{\Theta}'_* - \boldsymbol{\Theta}_*, \nabla\ell_r(\boldsymbol{\Theta}'_*\mathbf{W}_e\mathbf{b}_m, \mathbf{b}_m)\mathbf{b}_m^\top\mathbf{W}_e^\top - \nabla\ell_r(\boldsymbol{\Theta}_*\mathbf{W}_e\mathbf{b}_m, \mathbf{b}_m)\mathbf{b}_m^\top\mathbf{W}_e^\top\right\rangle$$

$$= \frac{M}{c}\left[D_{L_r^B}(\boldsymbol{\Theta}'_*\|\boldsymbol{\Theta}_*) + D_{L_r^B}(\boldsymbol{\Theta}_*\|\boldsymbol{\Theta}'_*)\right] \tag{4}$$

where the first inequality follows from the fact that $\ell_r$ is $c$-strongly convex, and the final equality follows from the definition of Bregman divergence (the standalone loss terms cancel). We want an expression in terms of the loss functions $L$ and $L'$, which will subsequently allow us to obtain a bound expressed back in terms of the set of representative vectors. Focusing on the term in the brackets,

$$D_{L_r^B}(\boldsymbol{\Theta}'_*\|\boldsymbol{\Theta}_*) + D_{L_r^B}(\boldsymbol{\Theta}_*\|\boldsymbol{\Theta}'_*)$$

$$= -\langle\boldsymbol{\Theta}'_* - \boldsymbol{\Theta}_*, \nabla L_r^B(\boldsymbol{\Theta}_*)\rangle - \langle\boldsymbol{\Theta}_* - \boldsymbol{\Theta}'_*, \nabla L_r^B(\boldsymbol{\Theta}'_*)\rangle$$

$$= \lim_{\kappa\to 0^+} \frac{1}{\kappa}\left[L_r^B(\boldsymbol{\Theta}_*) - L_r^B(\kappa\boldsymbol{\Theta}'_* + (1-\kappa)\boldsymbol{\Theta}_*) + L_r^B(\boldsymbol{\Theta}'_*) - L_r^B(\kappa\boldsymbol{\Theta}_* + (1-\kappa)\boldsymbol{\Theta}'_*)\right]$$

$$\leq \lim_{\kappa\to 0^+} \frac{a}{\kappa}\left[L'_r(\boldsymbol{\Theta}_*) - L'_r(\kappa\boldsymbol{\Theta}'_* + (1-\kappa)\boldsymbol{\Theta}_*) + L'_r(\boldsymbol{\Theta}'_*) - L'_r(\kappa\boldsymbol{\Theta}_* + (1-\kappa)\boldsymbol{\Theta}'_*)\right]$$

$$= a\left[-\langle\boldsymbol{\Theta}'_* - \boldsymbol{\Theta}_*, \nabla L'_r(\boldsymbol{\Theta}_*)\rangle - \langle\boldsymbol{\Theta}_* - \boldsymbol{\Theta}'_*, \nabla L'_r(\boldsymbol{\Theta}'_*)\rangle\right]$$

$$= a\left[D_{L'_r}(\boldsymbol{\Theta}'_*\|\boldsymbol{\Theta}_*) + D_{L'_r}(\boldsymbol{\Theta}_*\|\boldsymbol{\Theta}'_*)\right]$$

$$\leq a\left[D_L(\boldsymbol{\Theta}'_*\|\boldsymbol{\Theta}_*) + D_{L'}(\boldsymbol{\Theta}_*\|\boldsymbol{\Theta}'_*)\right] \tag{5}$$

where the first and last equalities follows from the definition of Bregman divergence, and the second equality from the definition of directional derivatives. The first inequality follows from Assumption 2; for the second inequality, note that $L'_r$ consists of a strict subset of the set of strictly convex losses that $L$ consists of, and similarly for $L'$. Therefore by additivity and non-negativity of the Bregman distance, we have that $D_{L'_r}(\mathbf{\Theta}'_*\|\mathbf{\Theta}_*) \leq D_L(\mathbf{\Theta}'_*\|\mathbf{\Theta}_*)$ and $D_{L'_r}(\mathbf{\Theta}_*\|\mathbf{\Theta}'_*) \leq D_{L'}(\mathbf{\Theta}_*\|\mathbf{\Theta}'_*)$. Our third and final step is to go back and bound this term again using the set of representative vectors,

$$D_L(\mathbf{\Theta}'_*\|\mathbf{\Theta}_*) + D_{L'}(\mathbf{\Theta}_*\|\mathbf{\Theta}'_*)$$

$$= L(\mathbf{\Theta}'_*) - L(\mathbf{\Theta}_*) + L'(\mathbf{\Theta}_*) - L'(\mathbf{\Theta}'_*)$$

$$= [L(\mathbf{\Theta}'_*) - L'(\mathbf{\Theta}'_*)] - [L(\mathbf{\Theta}_*) - L'(\mathbf{\Theta}_*)]$$

$$= \frac{1}{N}[\ell_p(\mathbf{\Theta}'_*\mathbf{W}_u\mathbf{x}_i, \mathbf{y}_i) + \ell_r(\mathbf{\Theta}'_*\mathbf{W}_e\mathbf{y}_i, \mathbf{y}_i)] - \frac{1}{N}[\ell_p(\mathbf{\Theta}'_*\mathbf{W}_u\mathbf{x}'_i, \mathbf{y}'_i) + \ell_r(\mathbf{\Theta}'_*\mathbf{W}_e\mathbf{y}'_i, \mathbf{y}'_i)]$$

$$- \frac{1}{N}[\ell_p(\mathbf{\Theta}_*\mathbf{W}_u\mathbf{x}_i, \mathbf{y}_i) + \ell_r(\mathbf{\Theta}_*\mathbf{W}_e\mathbf{y}_i, \mathbf{y}_i)] + \frac{1}{N}[\ell_p(\mathbf{\Theta}_*\mathbf{W}_u\mathbf{x}'_i, \mathbf{y}'_i) + \ell_r(\mathbf{\Theta}_*\mathbf{W}_e\mathbf{y}'_i, \mathbf{y}'_i)]$$

$$\leq \frac{\sigma_p}{N}(\|(\mathbf{\Theta}'_* - \mathbf{\Theta}_*)\mathbf{W}_u\mathbf{x}_i\|_2 + \|(\mathbf{\Theta}'_* - \mathbf{\Theta}_*)\mathbf{W}_u\mathbf{x}'_i\|_2)$$

$$+ \frac{\sigma_r}{N}(\|(\mathbf{\Theta}'_* - \mathbf{\Theta}_*)\mathbf{W}_e\mathbf{y}_i\|_2 + \|(\mathbf{\Theta}'_* - \mathbf{\Theta}_*)\mathbf{W}_e\mathbf{y}'_i\|_2)$$

$$\leq \frac{2(\sigma_p r_\alpha + \sigma_r r_\beta)}{N}\sqrt{\sum_{m=1}^M \|(\mathbf{\Theta}'_* - \mathbf{\Theta}_*)\mathbf{W}_e\mathbf{b}_m\|_2^2} \tag{6}$$

where for first equality note that by construction the gradients of the losses $L$ and $L'$ are zero at the respective optimal models $\mathbf{\Theta}_*$ and $\mathbf{\Theta}'_*$. The first inequality follows from the fact that $\ell_p$ and $\ell_r$ are $\sigma_p$-admissible and $\sigma_r$-admissible respectively, and the second inequality follows from Assumption 1 and the Cauchy-Schwarz inequality. Now, combining Equations 4, 5, and 6 allows us to write

$$\sqrt{\sum_{m=1}^M \|(\mathbf{\Theta}'_* - \mathbf{\Theta}_*)\mathbf{W}_e\mathbf{b}_m\|_2^2} \leq \frac{2(\sigma_p r_\alpha + \sigma_r r_\beta)aM}{cN} \tag{7}$$

which—by substitution into Equation 3—completes the proof.

**Remark 4** Formulating the autoencoding component as an auxiliary task allows us to unambiguously interpret its benefit as a regularizer. Specifically, the complete loss can be summarized and rewritten as $L(\mathbf{\Theta}) = L_p(\mathbf{\Theta}) + R_1(\mathbf{\Theta}) + R_2(\mathbf{\Theta})$; that is, the TEA objective is a combination of the primary prediction loss $L_p(\mathbf{\Theta}) = \frac{1}{N}\sum_{n=1}^N \ell_p(\mathbf{\Theta}\mathbf{W}_u\mathbf{x}_n, \mathbf{y}_n)$ plus additional regularization, where $R_1(\mathbf{\Theta}) = \frac{1}{N}\sum_{m=1}^M \ell_r(\mathbf{\Theta}\mathbf{W}_e\mathbf{b}_m, \mathbf{b}_m) = \frac{M}{N}L_r^B(\mathbf{\Theta})$ and $R_2(\mathbf{\Theta}) = \frac{1}{N}\sum_{n=1}^N \ell_r(\mathbf{\Theta}\mathbf{W}_e\mathbf{y}_n, \mathbf{y}_n) - \frac{M}{N}L_r^B(\mathbf{\Theta})$. In particular, the proof of Theorem 1 relies on $R_1(\mathbf{\Theta})$ to upper-bound instability (Appendix A). This precisely identifies the regularizer in question, while Theorem 1 quantifies its generalization benefit.

**Remark 5** Technicality: Moving from uniform stability (Theorem 1) to generalization bounds (Corollary 1) requires that the loss function not take on arbitrarily large values (see e.g. Bousquet & Elisseeff (2002)). In practice, the label space itself is often bounded (see e.g. Assumption 1 in Le et al. (2018)); then the problem effectively reduces to that with a bounded loss function. For example, consider a regression setting using the quadratic loss function, where the target data lie within $[-U, U]$; then the loss function is bounded to be within $[0, 4U^2]$. See Castro & Nowak (2018).

**Remark 6** Technicality: Earlier we assumed $\varepsilon = 0$ for ease of exposition; carrying around the extra $O(\varepsilon)$ term (from Equations 3 and 6) is not particularly illuminating. Generalizing to $\varepsilon \neq 0$ can be done in a similar manner to Le et al. (2018), with the additional assumptions of bounded spaces and that $\varepsilon$ decreases as $1/N$ (which they note is reasonable, since the more samples in the data, the more likely the cross-representativity assumption will hold with low error). Again, note that $\varepsilon = 0$ holds as long as the number of independent latent vectors is at least $|\mathcal{Z}|$). Similarly, Liu et al. (2016) consider $\varepsilon = 0$, noting in any case that they can increase $N$ to obtain a small $\|\eta\|_2$; see their analysis for detail.

## B    EXPANDED RELATED WORK

In this paper, we motivate and analyze a general autoencoder-based target-representation learning technique in the supervised setting, quantifying the generalization benefit via an argument from uniform stability, as well as verifying its practical utility. As such, our work lies at the intersection of three threads of research: (1) supervised representation learning using autoencoders, (2) label space reduction for multi-label classification, as well as (3) algorithmic stability-based learning guarantees.

### B.1    Supervised Representation Learning using Autoencoders

Table 6: Autoencoder-Based Supervised Representation Learning

| Work | Contribution | Setting | Type | Embedding |
|---|---|---|---|---|
| Weston et al. (2012) | Jointly optimized classific-ation and embedding | Semi-supervised learning | Deterministic | Features |
| Kingma et al. (2014) | Variational inference for generative modeling | Semi-supervised learning | Probabilistic | Features |
| Narayanaswamy et al. (2017) | Disentangled latent representations | Semi-supervised learning | Probabilistic | Features |
| Zhuang et al. (2015) | Input- and output-encoding for transfer learning | Semi-supervised learning | Deterministic | Features |
| Bousmalis et al. (2016) | Paired autoencoders for transfer learning | Semi-supervised learning | Deterministic | Features |
| Ghifary et al. (2016) | Jointly optimized feature-embedding | Semi-supervised learning | Deterministic | Features |
| Le et al. (2018) | Jointly optimized feature-embedding | Supervised learning | Deterministic | Features |
| Dalca et al. (2018) | Generative model using learned target priors | Unsupervised, Unpaired | Probabilistic | Targets |
| Girdhar et al. (2016) | Jointly optimized target-embedding (Indirect) | Supervised learning | Deterministic | Targets |
| **(Ours)** | Jointly optimized target-embedding (Direct) | Supervised learning | Deterministic | Targets |

*Autoencoder-based representation learning* (Hinton & Salakhutdinov, 2006) has long played important roles in unsupervised and semi-supervised settings. Various inductive biases have been proposed to promote representations that are sparse (Ranzato et al., 2007), discrete (van den Oord et al., 2017), factorized (Chen et al., 2018), or hierarchical (Zhao et al., 2017), among others. For a more thorough overview of various methods, we refer the reader to Bengio et al. (2013) and Tschannen et al. (2018).

The goal of better representations is often for the benefit of downstream tasks. Semi-supervised autoencoders—trained on partially-labeled data—can be *jointly optimized* to obtain compact representations that improve generalization on supervised tasks (Ranzato & Szummer, 2008; Weston et al., 2012). This naturally extends to representations that are generative (Kingma et al., 2014), disentangled (Narayanaswamy et al., 2017), or hierarchical (Rasmus et al., 2015). In addition, semi-supervised autoencoders enable transfer learning across different domains via jointly-trained reconstruction-classification networks Ghifary et al. (2016), private-shared partitioned representations (Bousmalis et al., 2016), or by augmenting models with label-encoding layers (Zhuang et al., 2015).

Although less studied, more closely related to our work is the use of autoencoders in a *purely supervised* setting: rather than focusing on how specific architectural novelties may better structure unlabeled data, Le et al. (2018) instead study the generalization benefit by the simple addition of reconstruction to the supervised classification task—a special case of what we describe as FEAs. Now, all aforementioned studies operate on the basis of autoencoding *features* (for an explicit or implicit downstream prediction task). In this paper, we instead focus on autoencoder-based *target*-representation learning (using TEAs) in the supervised setting, and—importantly—analyze the theoretical and empirical benefits of the approach. Unlike in simple classification (for which FEAs make sense), we are motivated by problems with high-dimensional output spaces, but where we operate under the assumption of a more compact and predictable set of underlying factors.

We take inspiration from the empirical investigation of Girdhar et al. (2016), where latent representations of 3D objects (targets) are jointly trained to be predictable from 2D images (features);

Table 7: Label Space Dimension Reduction via Label Embedding

| Work | Contribution | Learning | Problem |
|---|---|---|---|
| Hsu et al. (2009) | Coding with compressed sensing (random projections) | Separate | Multi-label classification |
| Tai & Lin (2010) | Coding with principal label space transformation (PC-based projections) | Separate | Multi-label classification |
| Zhang & Schneider (2011) | Coding with maximum margin (between prediction distances) | Separate | Multi-label classification |
| Balasubramanian & Lebanon (2012) | Landmark selection of labels via group-sparse learning | Separate | Multi-label classification |
| Bi & Kwok (2013) | Landmark selection of labels via randomized sampling | Separate | Multi-label classification |
| Chen & Lin (2012) | Feature-aware principal label space transformation for embedding | Joint | Multi-label classification |
| Yu et al. (2014) | Generic empirical risk minimization formulation and bounds | Joint | Multi-label classification |
| Yeh et al. (2017) | Autoencoder embedding with canonical correlation analysis | Joint | Multi-label classification |
| Mostajabi et al. (2018) | Autoencoder component as (Implicit) regularization for learning predictor | Separate | General; Semantic segmentation |
| **(Ours)** | Autoencoder component as (Explicit) regularization for learning predictor | Joint | General; Sequence forecasting |

Oktay et al. (2017) deploy a similar approach for medical image segmentation. On the other hand, in both cases the supervised task is truncated: the predictors are trained to regress the unsupervised embeddings (instead of ground-truth targets), and gradients only backpropagate from the latent space (instead of the target space). This means that their common decoder function is only shared *indirectly* (and predictions made indirectly), versus the symmetric and simultaneously optimized forward model proposed for TEAs—an important distinction that our analysis relies on to obtain uniform stability. In Mostajabi et al. (2018), a two-stage procedure is used for semantic segmentation—loosely comparable to the first two stages in TEAs; in contrast to our emphasis on joint training, they study the benefit of a *frozen* embedding branch in parallel with direct prediction. More broadly related to target-embedding, Dalca et al. (2018) build anatomical priors for biomedical segmentation in *unsupervised* settings.

### B.2 Label Space Reduction for Multi-Label Classification

*Label space dimension reduction* comprises techniques that focus specifically on multi-label classification. Early approaches to multi-label classification employ simplistic transformations such as label power-sets (Boutell et al., 2004), binary relevance (Tsoumakas et al., 2009), and label rankings (Fürnkranz et al., 2008); these are computationally inefficient, and do not capture interdependencies between labels. In contrast, label-embedding methods first derive a latent label space with reduced dimensionality, and subsequently associate inputs to that latent space instead. Encodings have been obtained via random projections (Hsu et al., 2009), principal components-based projections (Tai & Lin, 2010), canonical correlation analysis (Zhang & Schneider, 2011), as well as maximum-margin coding (Zhang & Schneider, 2012). A parallel thread of research has focused on selecting representative and reconstructive subsets of labels through group-sparse learning (Balasubramanian & Lebanon, 2012) and randomized sampling (Bi & Kwok, 2013). Various extensions of label-embedding techniques abound, such as using bloom filters (Cisse et al., 2013), nearest-neighbors (Bhatia et al., 2015), handling missing data (Wu et al., 2014), as well as using binary compression (Zhou et al., 2017).

Closer our theme of *joint* learning by utilizing both features and targets, Chen & Lin (2012) first proposed simultaneously minimizing the encoding error (from labels) and prediction error (from features) through an SVD formulation. Towards more *flexible* learning, Lin et al. (2014) did away with explicitly specified encoding functions, proposing to learn code matrices directly—making no assumptions whatsoever. Unifying several prior methods, Yu et al. (2014) cast label-embedding within the generic *empirical risk minimization* framework—as learning a linear model with a low-rank constraint; this perspective captures the generic intuition of a restricted number of latent factors, and admits generalization bounds based on norm-based regularization. Recently, Yeh et al. (2017) generalized the label-embedding approach to *autoencoders*; this formulation flexibly allows the

Table 8: Generalizability, Multiple Tasks, and Algorithmic Stability

| Work | Contribution | Setting | Focus | Learning |
|---|---|---|---|---|
| Bousquet & Elisseeff (2002) | Uniform stability for generalization | Supervised, general | Single task | - |
| Feldman & Vondrak (2018) | Improve on Bousquet & Elisseeff's bound | Supervised, general | Single task | - |
| Baxter (2000) | VC-dimension for generalization | Multi-task learning | All tasks | Jointly |
| Maurer (2006) | Rademacher complexity for generalization | Multi-task learning | All tasks | Jointly |
| Liu et al. (2016) | Uniform stability for generalization | Multi-task learning | All tasks | Jointly |
| Mohri et al. (2015) | Rademacher complexity for generalization | Feature-embedding | Primary task | Separately |
| Epstein & Meir (2019) | Non-contractiveness and semi-supervision | Feature-embedding | Primary task | Separately |
| Le et al. (2018) | Uniform stability for generalization | Feature-embedding | Primary task | Jointly |
| **(Ours)** | Uniform stability for generalization | Target-embedding | Primary task | Jointly |

addition of specific losses to exploit correlations—a tactic also used in Wang et al. (2018) with multi-dimensional scaling. Furthermore, nonlinearities can be handled by deep learning in component functions, unlike earlier approaches limited to kernel methods (Lin et al., 2014; Li & Guo, 2015).

Our work is related to this general autoencoder approach to label-embedding, although there are significant differences in focus. In particular, we operate at a higher level of abstraction. Label-embedding techniques worry about label *reduction*, and about specific loss functions that aim to preserve dependencies within and among spaces; their problem is one of multi-label classification, and their baseline is binary relevance. In contrast, we worry about *autoencoding* at all—that is, we focus on the regularizing effect of the reconstruction loss on learning the prediction model; our baseline is direct prediction, and the output can be of any form (classification or regression). In light of the sizable performance improvement of the autoencoder-based model of Yeh et al. (2017) over comparators using direct prediction, our work can be regarded as a more generalized analysis of the contribution of the autoencoding component. Moreover, unlike the uniform convergence-based analysis in Yu et al. (2014), our bound does not rely on explicit norm-based regularization—instead, we interpret the embedding task itself as an intrinsic form of regularization to derive our stability-based guarantee.

Finally, also worth mentioning is the field of *extreme* multi-label classification (Bhatia et al., 2015), for which probabilistic methods such as Rai et al. (2015) and Kapoor et al. (2012) present sophisticated approaches to extremely high-dimensional classification problems—with advantages in performance and use cases. In light of the medical relevance of our experimental setting, we point out the application of Yan et al. (2010) to medical coding. See Bhatia et al. (2019) for a more detailed overview.

### B.3  Algorithmic Stability and Learning Guarantees

Generalizability is central to machine learning, and its analysis via hypothesis stability is first studied in Rogers & Wagner (1978) and Devroye & Wagner (1979). Unlike arguments based on the complexity of the search space (Vapnik & Chervonenkis, 1971; Pollard, 1984; Koltchinskii, 2001), stability-based approaches account for how the model produced by the algorithm depends on the data. Based on concentration inequalities (McDiarmid, 1989), improved bounds are developed in Lugosi & Pawlak (1994) by estimating posterior error probabilities. The landmark work of Bousquet & Elisseeff (2002) first formalizes the notion of *uniform stability* sufficient for learnability, obtaining relatively strong bounds for several regularization algorithms, and Feldman & Vondrak (2018) recently use ideas related to differential privacy (Bassily et al., 2016) for further improved bounds without additional assumptions. For further context, see Mukherjee et al. (2006) and Shalev-Shwartz et al. (2010).

For semi-supervised representation learning, Rigollet (2007) first introduces the notion of cluster excess-risk and convergence, formalizing the clustering criterion for unlabeled features to be useful (Seeger, 2000). Based on the clustering assumption, Singh et al. (2009) develops a finite sample analysis to quantify the performance improvement from unlabeled features. Focusing on autoencoders,

Epstein & Meir (2019) adapt recent margin, norm, and compression-based results for deep networks (Bartlett et al., 2017; Neyshabur et al., 2017; Arora et al., 2018), and relate generalization of feature reconstructions to the benefit of additional unlabeled features for the primary classification task.

In the context of *supervised* problems, Mohri et al. (2015) and Gottlieb et al. (2016) analyze the generalization properties of dimensionality reduction techniques for features with respect to a downstream task; however, rather than joint training, the primary task is optimized subsequently over the learned representations. Taking a joint, multi-task approach (Caruana, 1997), Baxter (2000) first leverages the inductive bias of a common optimal hypothesis class to obtain a VC-based generalization bound. Maurer (2006) and Maurer et al. (2016) argue from Rademacher complexity to illustrate the benefit of the common operator; however, they only consider the task-averaged benefit, whereas we want to focus specifically on the primary task. There has been some work on generalization for each task (Ben-David & Schuller, 2003), but limited to binary classification—contrary to our setting.

Arguing from stability, our approach is related to Liu et al. (2016) in showing that the algorithm for learning the shared model in a multi-task setting is uniformly stable. Our analysis also resembles Le et al. (2018) in the more specific setting where the bound for the primary prediction task is obtained with assistance from the auxiliary reconstruction loss; unlike Liu et al. (2016), we are not interested in a generic bound for all tasks. Again, however, the fundamental (and motivating) difference of our work stems from the (inverted) problem setting and resulting framework. In the vast majority of works, the primary task is one of classification (or more generally $|\mathcal{X}| \gg |\mathcal{Y}|$), where *feature*-embeddings make sense to learn. Instead, we attend to the setting in which $\mathcal{Y}$ is high-dimensional (but where the underlying factors are assumed to be compact). In this setting, we argue (theoretically and empirically) that *target*-embeddings make more sense to learn in an auxiliary reconstruction task.

## C    EXPANDED ALGORITHM DETAIL

In the following, Algorithm 1 gives pseudocode for TEA training. Figure 4 gives detailed block diagrams of component functions and objectives corresponding to each training stage (and variant).

---

**Algorithm 1** Pseudocode for TEA Training

---

**Input:** $\mathcal{D} = \{(\mathbf{x}_n, \mathbf{y}_n)\}_{n=1}^N$ , learning rate $\psi$, minibatch size $N_s$
**Output:** Parameters $\boldsymbol{\Theta}, \mathbf{W}_u, \mathbf{W}_e$ of components $\theta, u, e$

1: **Initialize:** $\boldsymbol{\Theta}, \mathbf{W}_u, \mathbf{W}_e$
2: **while** not converged **do**                         ▷ **Stage 1:** Learn Target-Embedding
3:      Sample $\{(\mathbf{x}_n, \mathbf{y}_n)\}_{n=1}^{N_s} \overset{\text{i.i.d.}}{\sim} \mathcal{D}$
4:      **for** $n \in \{1, ..., N_s\}$ **do**
5:          $\mathbf{z}_n \leftarrow e(\mathbf{y}_n; \mathbf{W}_e)$                         ▷ Encode
6:          $\tilde{\mathbf{y}}_n \leftarrow \theta(\mathbf{z}_n; \boldsymbol{\Theta})$                         ▷ Decode
7:      **end for**
8:      $L_r \leftarrow \frac{1}{N_s} \sum_{n=1}^{N_s} \ell_r(\tilde{\mathbf{y}}_n, \mathbf{y}_n)$
9:      $\mathbf{W}_e \leftarrow \mathbf{W}_e - \psi \nabla_{\mathbf{W}_e} L_r$
10:      $\boldsymbol{\Theta} \leftarrow \boldsymbol{\Theta} - \psi \nabla_{\boldsymbol{\Theta}} L_r$
11: **end while**
12: **while** not converged **do**                         ▷ **Stage 2:** Regress Embeddings
13:      Sample $\{(\mathbf{x}_n, \mathbf{y}_n)\}_{n=1}^{N_s} \overset{\text{i.i.d.}}{\sim} \mathcal{D}$
14:      **for** $n \in \{1, ..., N_s\}$ **do**
15:          $\hat{\mathbf{z}}_n \leftarrow u(\mathbf{x}_n; \mathbf{W}_u)$                         ▷ Predict
16:          $\mathbf{z}_n \leftarrow e(\mathbf{y}_n; \mathbf{W}_e)$                         ▷ Encode
17:      **end for**
18:      $L_z \leftarrow \frac{1}{N_s} \sum_{n=1}^{N_s} \ell_z(\hat{\mathbf{z}}_n, \mathbf{z}_n)$
19:      $\mathbf{W}_u \leftarrow \mathbf{W}_u - \psi \nabla_{\mathbf{W}_u} L_z$
20: **end while**
21: **while** not converged **do**                         ▷ **Stage 3:** Joint Training
22:      Sample $\{(\mathbf{x}_n, \mathbf{y}_n)\}_{n=1}^{N_s} \overset{\text{i.i.d.}}{\sim} \mathcal{D}$

23:     **for** $n \in \{1, ..., N_s\}$ **do**
24:         $\hat{\mathbf{z}}_n \leftarrow u(\mathbf{x}_n; \mathbf{W}_u)$         $\triangleright$ Predict
25:         $\mathbf{z}_n \leftarrow e(\mathbf{y}_n; \mathbf{W}_e)$         $\triangleright$ Encode
26:         $\hat{\mathbf{y}}_n \leftarrow \theta(\hat{\mathbf{z}}_n; \boldsymbol{\Theta})$         $\triangleright$ Decode
27:         $\tilde{\mathbf{y}}_n \leftarrow \theta(\mathbf{z}_n; \boldsymbol{\Theta})$         $\triangleright$ Decode
28:     **end for**
29:     $L_p \leftarrow \frac{1}{N_s} \sum_{n=1}^{N_s} \ell_p(\hat{\mathbf{y}}_n, \mathbf{x}_n)$
30:     $L_r \leftarrow \frac{1}{N_s} \sum_{n=1}^{N_s} \ell_r(\tilde{\mathbf{y}}_n, \mathbf{y}_n)$
31:     $\mathbf{W}_u \leftarrow \mathbf{W}_u - \psi \nabla_{\mathbf{W}_u} L_p$
32:     $\mathbf{W}_e \leftarrow \mathbf{W}_e - \psi \nabla_{\mathbf{W}_e} L_r$
33:     $\boldsymbol{\Theta} \leftarrow \boldsymbol{\Theta} - \psi \nabla_{\boldsymbol{\Theta}} [L_p + L_r]$
34: **end while**
35: **return** $\boldsymbol{\Theta}, \mathbf{W}_u, \mathbf{W}_e$

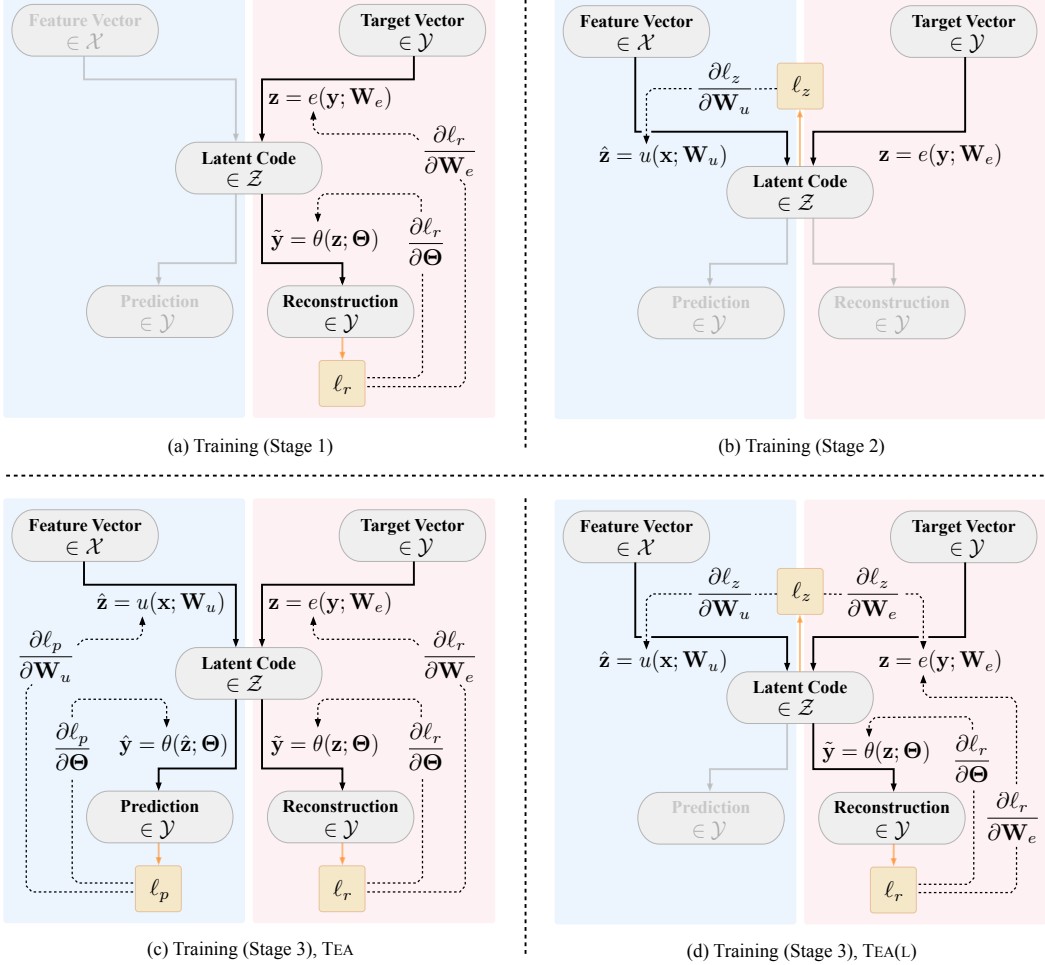

(a) Training (Stage 1)

(b) Training (Stage 2)

(c) Training (Stage 3), TEA

(d) Training (Stage 3), TEA(L)

Figure 4: TEAs consist of a shared forward model $\theta$, upstream predictor $u$, and target-embedding function $e$, parameterized by $(\boldsymbol{\Theta}, \mathbf{W}_u, \mathbf{W}_e)$. Blue and red respectively identify the supervised and representation learning components in each arrangement. Solid lines indicate forward propagation of data; dashed lines indicate backpropagation of gradients. (a) First, the autoencoding components $e, \theta$ are trained to learn target representations. (b) Next, using the inputs, the prediction arm $u$ is trained to regress the learned embeddings generated by the encoder. (c) Finally, all three components are jointly trained on both prediction and reconstruction losses. (d) In the indirect variant (Girdhar et al., 2016),

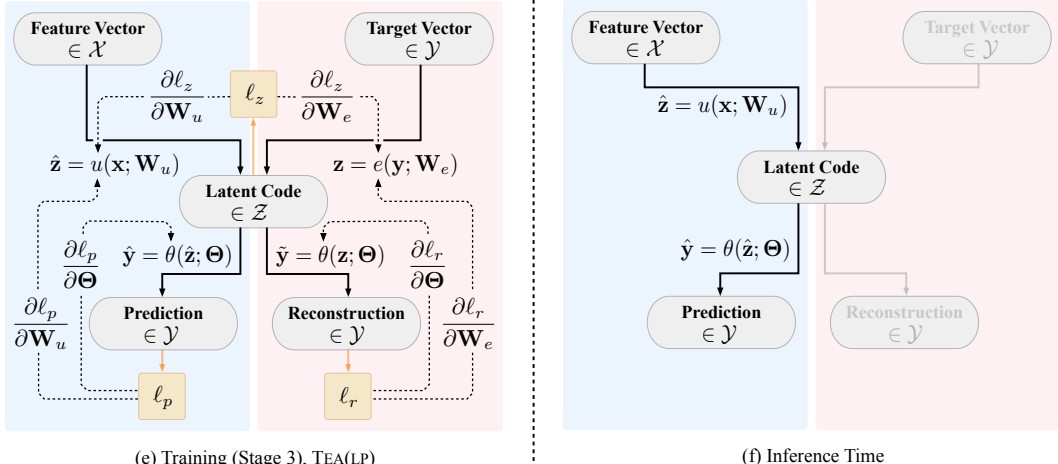

(e) Training (Stage 3), TEA(LP)

(f) Inference Time

Figure 4: (*continued*) the predictor continues to regress the learned embeddings, and the latent loss backpropagates through both $u$ and $e$ (TEA(L)). (e) The TEA(LP) variant combines the previous two: both the latent loss and prediction loss are trained jointly together with the reconstruction loss. (f) At inference time, the target-embedding arm is dropped, leaving the hypothesis $h = \theta \circ u$ for prediction.

## D ADDITIONAL EXPERIMENT DETAIL

The UK Cystic Fibrosis registry records follow-up trajectories for over 10,000 patients over the period from 2008 and 2015 with a total of over 60,000 hospital visits. Each patient is associated with 90 variables over time, which includes data on treatments and diagnoses for 23 possible comorbidities (e.g. ABPA, diabetes, hypertension, pancreatitis), 11 possible infections (e.g. aspergillus, burkholderia cepacia, klebsiella pneumoniae), as well as static demographic information (e.g. gender, genetics, smoking status). Using both static and temporal information in a precedent window, we forecast the future trajectories for the diagnoses of infections and comorbidities (all binary variables) recorded at each follow-up. The Alzheimer's Disease Neuroimaging Initiative study tracks disease progression for over 1,700 patients over the period from 2004 to 2016 with a total of over 10,000 (bi-annual)

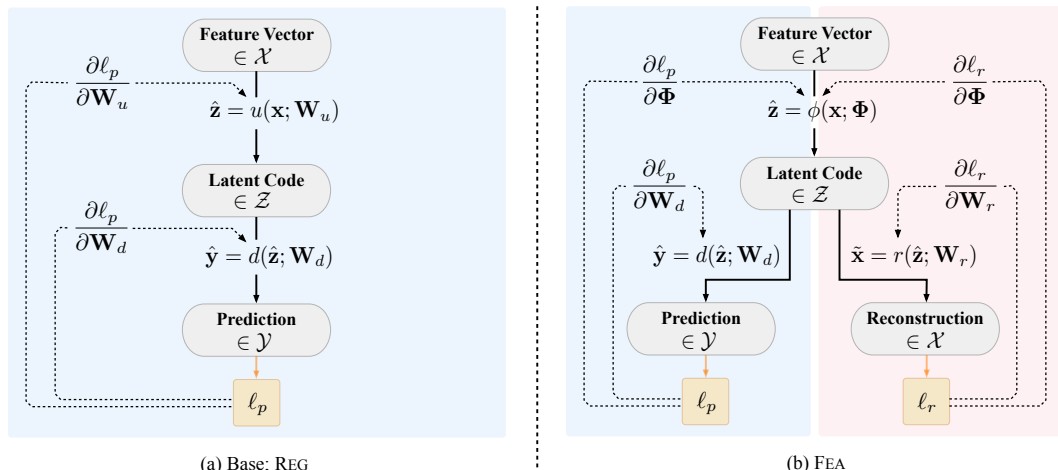

(a) Base; REG

(b) FEA

Figure 5: Component functions and training objectives for comparators in experiments. Blue and red respectively identify the supervised and representation learning components in each arrangement. Solid lines indicate forward propagation of data; dashed lines indicate backpropagation of gradients. (a) The baseline is direct prediction (with (REG) and without (Base) $\ell_2$-regularization), which simply corresponds to removing the autoencoder; here we explicitly identify some intermediate hidden layer to preserve visual correspondence with the autoencoder models, but note that the "latent code" is strictly speaking a misnomer—as nothing is being encoded here. (b) FEAs consist of a shared forward

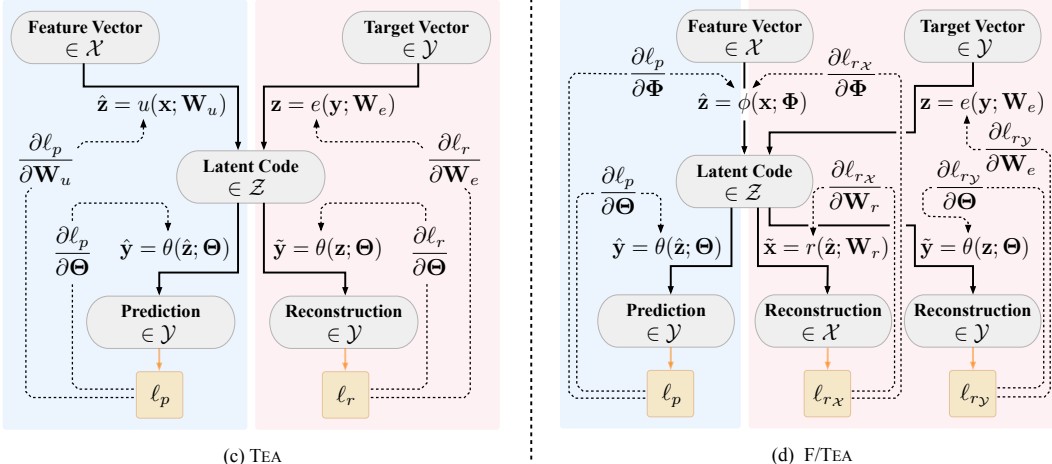

(c) TEA        (d) F/TEA

Figure 5: (*continued*) model $\phi$, downstream predictor $d$, and feature reconstructor $r$, parameterized by $(\Phi, \mathbf{W}_d, \mathbf{W}_r)$. (c) TEAs consist of a shared forward model $\theta$, upstream predictor $u$, and target-embedding function $e$, parameterized by $(\Theta, \mathbf{W}_u, \mathbf{W}_e)$. (d) As an additional sensitivity, F/TEAs combine the previous two, forcing intermediate representations to encode *both* features and targets.

clinical visits. We focus on the 8 primary quantitative biomarkers (e.g. entorhinal cortex, fusiform gyrus, hippocampus), 16 cognitive tests (e.g. ADAS11, CDR sum of boxes, mini mental state exam), as well as static demographic information (e.g. apolipoprotein E4, education level, ethnicity); we omit the remaining variables, for which the rate of missingness is over 50%. Using a precedent window, we forecast the future the evolution of the primary quantitative biomarkers and cognitive test results (all continuous variables) measured at each visit. The Medical Information Mart for Intensive Care records physiological data streams patients admitted to intensive care units after 2008. We use over 22,000 patients with a total of over 500,000 measurements (resampled at 4 hour intervals). We focus on the most frequently measured vital signs and lab tests (e.g. heart rate, oxygen saturation, respiratory rate) recorded over time (with categorical variables binarized, this gives a total of 361 variables), as well as static demographic information (e.g. admission type, gender, location, marital status); we omit the remaining variables, for which the rate of missingness is over 50%. Using a precedent window, we forecast the subsequent window of those variables. For each dataset, sequences are randomized at the patient level in order to obtain splits for training (and validation) and testing.

We implement all models using Tensorflow. For the linear model, each component (encoder, decoder, and predictor) consists of a single layer, no bias term, and linear activation; static (demographic) and temporal data are concatenated and flattened for both features and targets. For the nonlinear case, we implement each component as an RNN using GRUs with the number of hidden layers $\zeta \in \{1, 2\}$ where the number of hidden units is equal to the temporal feature dimension, and tanh is used for activation; the dimension of the latent space is (therefore) equal to the hidden state dimension. (For even larger hidden capacities, the increased number of parameters rapidly degrades performance). Static (demographic) information is incorporated as a mapping into the initial state for recurrent cells. Training is performed using the ADAM optimizer with a learning rate of $\psi \in \{3e-5, 3e-4, 3e-3, 3e-2\}$. Models are trained until convergence up to a maximum of 10,000 iterations with a minibatch size of $N_s \in \{32, 64, 128\}$; the empirical loss is computed on the validation set every 50 iterations of training, and convergence is determined on the basis of that error. Checkpointing is implemented every 50 iterations, and the best model parameters are restored (upon convergence) for use on the testing set. For all models except "Base", we allow the opportunity to select among the $\ell_2$-regularization coefficients $\nu \in \{0, 3e-5, 3e-4, 3e-3, 3e-2\}$. We set the strength-of-prior coefficient $\lambda = 0.5$ for FEA, F/TEA, as well as all variants of TEA (however, we do provide sensitivities on $\lambda$ for TEA in our experiments). For hyperparameter tuning ($\zeta, \psi, \nu, N_s$), we use cross-validation on the training set using 20 iterations of random search, selecting the setting that gives the lowest validation loss averaged across folds. For fair comparison (so as to isolate the effect of supervised representation learning over and above direct prediction), we apply the same setting chosen for REG for FEA, F/TEA, and all variants of TEA; therefore the only difference is the presence

or absence of each autoencoding component. For each model and dataset, the experiment is repeated for a total of 10 times (each with a different random split of data into training and held-out testing sets); all results are reported as means and standard errors of each performance metric across runs.

# E  ADDITIONAL EXPERIMENT RESULTS

## E.1  Results for Linear Models

Table 9: Extended results for TEA and comparators on linear model with UKCF (Bold indicates best)

| | $\tau$ | Base | REG | FEA | TEA | F/TEA |
|---|---|---|---|---|---|---|
| PRC(I) | 1 | $0.340 \pm 0.118$* | $0.370 \pm 0.101$* | $0.374 \pm 0.091$* | $\mathbf{0.497 \pm 0.016}$ | $0.454 \pm 0.010$* |
| | 2 | $0.325 \pm 0.099$* | $0.350 \pm 0.085$* | $0.353 \pm 0.077$* | $\mathbf{0.459 \pm 0.017}$ | $0.419 \pm 0.012$* |
| | 3 | $0.314 \pm 0.088$* | $0.337 \pm 0.076$* | $0.342 \pm 0.071$* | $\mathbf{0.432 \pm 0.015}$ | $0.399 \pm 0.013$* |
| | 4 | $0.307 \pm 0.082$* | $0.328 \pm 0.071$* | $0.333 \pm 0.067$* | $\mathbf{0.413 \pm 0.010}$ | $0.386 \pm 0.009$* |
| PRC(C) | 1 | $0.445 \pm 0.131$* | $0.467 \pm 0.106$* | $0.499 \pm 0.110$* | $\mathbf{0.653 \pm 0.009}$ | $0.599 \pm 0.019$* |
| | 2 | $0.418 \pm 0.099$* | $0.435 \pm 0.081$* | $0.457 \pm 0.083$* | $\mathbf{0.566 \pm 0.010}$ | $0.524 \pm 0.017$* |
| | 3 | $0.403 \pm 0.082$* | $0.418 \pm 0.067$* | $0.436 \pm 0.069$* | $\mathbf{0.521 \pm 0.008}$ | $0.487 \pm 0.015$* |
| | 4 | $0.398 \pm 0.073$* | $0.412 \pm 0.060$* | $0.426 \pm 0.061$* | $\mathbf{0.498 \pm 0.009}$ | $0.471 \pm 0.013$* |
| ROC(I) | 1 | $0.713 \pm 0.104$* | $0.737 \pm 0.081$* | $0.750 \pm 0.078$ | $\mathbf{0.806 \pm 0.007}$ | $0.801 \pm 0.007$ |
| | 2 | $0.688 \pm 0.089$* | $0.709 \pm 0.070$* | $0.718 \pm 0.068$* | $\mathbf{0.771 \pm 0.007}$ | $0.765 \pm 0.009$ |
| | 3 | $0.677 \pm 0.080$* | $0.697 \pm 0.065$* | $0.706 \pm 0.066$ | $\mathbf{0.750 \pm 0.007}$ | $0.749 \pm 0.009$ |
| | 4 | $0.677 \pm 0.076$* | $0.696 \pm 0.063$ | $0.707 \pm 0.068$ | $0.740 \pm 0.008$ | $\mathbf{0.748 \pm 0.006}$* |
| ROC(C) | 1 | $0.713 \pm 0.110$* | $0.741 \pm 0.088$* | $0.756 \pm 0.084$* | $\mathbf{0.829 \pm 0.006}$ | $0.810 \pm 0.008$* |
| | 2 | $0.686 \pm 0.091$* | $0.707 \pm 0.072$* | $0.719 \pm 0.071$* | $\mathbf{0.775 \pm 0.008}$ | $0.762 \pm 0.008$* |
| | 3 | $0.668 \pm 0.077$* | $0.686 \pm 0.061$* | $0.699 \pm 0.063$ | $\mathbf{0.743 \pm 0.005}$ | $0.735 \pm 0.007$* |
| | 4 | $0.651 \pm 0.070$* | $0.667 \pm 0.056$* | $0.679 \pm 0.057$* | $\mathbf{0.720 \pm 0.006}$ | $0.712 \pm 0.009$* |

PRC and ROC evaluations are reported separately for targets representing infections (I) and comorbidities (C). The two-sample $t$-test for a difference in means is conducted on the results. An asterisk next to the comparator result is used to indicate a statistically significant difference in means ($p$-value $< 0.05$) relative to the TEA result.

Table 10: Summary results for TEA and comparators on linear model with UKCF (Bold indicates best)

| | Base | REG | FEA | TEA | F/TEA |
|---|---|---|---|---|---|
| PRC(I) | $0.322 \pm 0.099$* | $0.347 \pm 0.085$* | $0.351 \pm 0.079$* | $\mathbf{0.450 \pm 0.035}$ | $0.414 \pm 0.028$* |
| PRC(C) | $0.416 \pm 0.100$* | $0.433 \pm 0.083$* | $0.455 \pm 0.087$* | $\mathbf{0.559 \pm 0.060}$ | $0.520 \pm 0.052$ |
| ROC(I) | $0.689 \pm 0.089$* | $0.710 \pm 0.072$* | $0.720 \pm 0.073$ | $\mathbf{0.767 \pm 0.026}$ | $0.766 \pm 0.023$ |
| ROC(C) | $0.679 \pm 0.091$* | $0.700 \pm 0.075$* | $0.713 \pm 0.075$ | $\mathbf{0.767 \pm 0.042}$ | $0.755 \pm 0.037$ |

PRC and ROC evaluations are reported separately for targets representing infections (I) and comorbidities (C). The two-sample $t$-test for a difference in means is conducted on the results. An asterisk next to the comparator result is used to indicate a statistically significant difference in means ($p$-value $< 0.05$) relative to the TEA result. Results are grouped over the temporal axis; note that the variance between splits is an artifact of this grouping.

Table 11: Extended source of gain and variants on linear model with UKCF (Bold indicates best)

| | $\tau$ | No Joint | No Staged | TEA | TEA(L) | TEA(LP) |
|---|---|---|---|---|---|---|
| PRC(I) | 1 | $0.434 \pm 0.017$ | $0.472 \pm 0.016$ | $0.497 \pm 0.016$ | $0.474 \pm 0.018$ | $\mathbf{0.502 \pm 0.015}$ |
| | 2 | $0.408 \pm 0.016$ | $0.439 \pm 0.016$ | $0.459 \pm 0.017$ | $0.443 \pm 0.018$ | $\mathbf{0.463 \pm 0.016}$ |
| | 3 | $0.390 \pm 0.015$ | $0.415 \pm 0.015$ | $0.432 \pm 0.015$ | $0.420 \pm 0.014$ | $\mathbf{0.435 \pm 0.013}$ |
| | 4 | $0.377 \pm 0.014$ | $0.399 \pm 0.011$ | $0.413 \pm 0.010$ | $0.402 \pm 0.010$ | $\mathbf{0.415 \pm 0.009}$ |
| PRC(C) | 1 | $0.563 \pm 0.025$ | $0.620 \pm 0.030$ | $0.653 \pm 0.009$ | $0.626 \pm 0.012$ | $\mathbf{0.655 \pm 0.008}$ |
| | 2 | $0.511 \pm 0.018$ | $0.550 \pm 0.022$ | $\mathbf{0.566 \pm 0.010}$ | $0.550 \pm 0.008$ | $\mathbf{0.566 \pm 0.010}$ |
| | 3 | $0.484 \pm 0.014$ | $0.512 \pm 0.017$ | $\mathbf{0.521 \pm 0.008}$ | $0.511 \pm 0.006$ | $\mathbf{0.521 \pm 0.008}$ |
| | 4 | $0.469 \pm 0.014$ | $0.491 \pm 0.015$ | $0.498 \pm 0.009$ | $0.490 \pm 0.009$ | $\mathbf{0.499 \pm 0.009}$ |
| ROC(I) | 1 | $0.779 \pm 0.008$ | $0.799 \pm 0.007$ | $0.806 \pm 0.007$ | $0.797 \pm 0.007$ | $\mathbf{0.809 \pm 0.005}$ |
| | 2 | $0.750 \pm 0.008$ | $0.765 \pm 0.009$ | $0.771 \pm 0.007$ | $0.763 \pm 0.008$ | $\mathbf{0.772 \pm 0.007}$ |

| | | | | | | |
|---|---|---|---|---|---|---|
| | 3 | $0.733 \pm 0.008$ | $\mathbf{0.750 \pm 0.008}$ | $\mathbf{0.750 \pm 0.007}$ | $0.744 \pm 0.007$ | $\mathbf{0.750 \pm 0.008}$ |
| | 4 | $0.725 \pm 0.007$ | $\mathbf{0.744 \pm 0.005}$ | $0.740 \pm 0.008$ | $0.734 \pm 0.006$ | $0.740 \pm 0.007$ |
| Roc(c) | 1 | $0.799 \pm 0.012$ | $0.821 \pm 0.011$ | $0.829 \pm 0.006$ | $0.823 \pm 0.005$ | $\mathbf{0.830 \pm 0.005}$ |
| | 2 | $0.751 \pm 0.012$ | $0.774 \pm 0.009$ | $\mathbf{0.775 \pm 0.008}$ | $0.769 \pm 0.006$ | $\mathbf{0.775 \pm 0.006}$ |
| | 3 | $0.723 \pm 0.010$ | $\mathbf{0.746 \pm 0.007}$ | $0.743 \pm 0.005$ | $0.737 \pm 0.005$ | $0.742 \pm 0.005$ |
| | 4 | $0.702 \pm 0.011$ | $\mathbf{0.722 \pm 0.007}$ | $0.720 \pm 0.006$ | $0.713 \pm 0.008$ | $0.720 \pm 0.006$ |

PRC and ROC evaluations are reported separately for targets representing infections (I) and comorbidities (C). The "No Joint" setting isolates the benefit from staged training only (analogous to basic unsupervised pretraining, though using targets); the "No Staged" setting isolates the benefit from joint training only (without pretraining).

Table 12: Summary source of gain and variants on linear model with UKCF (Bold indicates best)

| | No Joint | No Staged | TEA | TEA(L) | TEA(LP) |
|---|---|---|---|---|---|
| PRC(I) | $0.402 \pm 0.026$ | $0.431 \pm 0.031$ | $0.450 \pm 0.035$ | $0.435 \pm 0.031$ | $\mathbf{0.454 \pm 0.036}$ |
| PRC(C) | $0.507 \pm 0.040$ | $0.543 \pm 0.054$ | $0.559 \pm 0.060$ | $0.544 \pm 0.053$ | $\mathbf{0.560 \pm 0.061}$ |
| ROC(I) | $0.747 \pm 0.022$ | $0.764 \pm 0.022$ | $0.767 \pm 0.026$ | $0.759 \pm 0.025$ | $\mathbf{0.768 \pm 0.028}$ |
| ROC(C) | $0.744 \pm 0.038$ | $0.766 \pm 0.038$ | $\mathbf{0.767 \pm 0.042}$ | $0.760 \pm 0.042$ | $\mathbf{0.767 \pm 0.042}$ |

PRC and ROC evaluations are reported separately for targets representing infections (I) and comorbidities (C). The "No Joint" setting isolates the benefit from staged training only (analogous to basic unsupervised pretraining, though using targets); the "No Staged" setting isolates the benefit from joint training only (without pretraining). Results are grouped over the temporal axis; note that the variance between splits is an artifact of this grouping.

## E.2 Results for Recurrent Models

Table 13: Extended results for TEA and comparators on RNN model with UKCF (Bold indicates best)

| | $\tau$ | Base | REG | FEA | TEA | F/TEA |
|---|---|---|---|---|---|---|
| PRC(I) | 1 | $0.451 \pm 0.027^*$ | $0.456 \pm 0.016^*$ | $0.448 \pm 0.026^*$ | $\mathbf{0.549 \pm 0.014}$ | $0.509 \pm 0.014^*$ |
| | 2 | $0.417 \pm 0.024^*$ | $0.420 \pm 0.013^*$ | $0.416 \pm 0.022^*$ | $\mathbf{0.490 \pm 0.012}$ | $0.463 \pm 0.011^*$ |
| | 3 | $0.395 \pm 0.019^*$ | $0.400 \pm 0.013^*$ | $0.395 \pm 0.020^*$ | $\mathbf{0.457 \pm 0.010}$ | $0.437 \pm 0.013^*$ |
| | 4 | $0.380 \pm 0.017^*$ | $0.385 \pm 0.010^*$ | $0.380 \pm 0.017^*$ | $\mathbf{0.434 \pm 0.007}$ | $0.417 \pm 0.013^*$ |
| PRC(C) | 1 | $0.561 \pm 0.056^*$ | $0.592 \pm 0.029^*$ | $0.598 \pm 0.029^*$ | $\mathbf{0.695 \pm 0.010}$ | $0.685 \pm 0.015$ |
| | 2 | $0.504 \pm 0.039^*$ | $0.523 \pm 0.022^*$ | $0.527 \pm 0.021^*$ | $\mathbf{0.591 \pm 0.014}$ | $0.584 \pm 0.018$ |
| | 3 | $0.471 \pm 0.028^*$ | $0.488 \pm 0.018^*$ | $0.489 \pm 0.017^*$ | $\mathbf{0.537 \pm 0.007}$ | $0.530 \pm 0.017$ |
| | 4 | $0.453 \pm 0.023^*$ | $0.469 \pm 0.017^*$ | $0.470 \pm 0.016^*$ | $\mathbf{0.510 \pm 0.007}$ | $0.504 \pm 0.015$ |
| ROC(I) | 1 | $0.788 \pm 0.018^*$ | $0.791 \pm 0.009^*$ | $0.794 \pm 0.014^*$ | $\mathbf{0.827 \pm 0.007}$ | $0.818 \pm 0.006^*$ |
| | 2 | $0.753 \pm 0.015^*$ | $0.757 \pm 0.011^*$ | $0.758 \pm 0.017^*$ | $\mathbf{0.783 \pm 0.008}$ | $0.778 \pm 0.009$ |
| | 3 | $0.736 \pm 0.013^*$ | $0.741 \pm 0.012^*$ | $0.740 \pm 0.016^*$ | $\mathbf{0.760 \pm 0.007}$ | $0.757 \pm 0.010$ |
| | 4 | $0.725 \pm 0.012^*$ | $0.731 \pm 0.011^*$ | $0.727 \pm 0.014^*$ | $\mathbf{0.748 \pm 0.008}$ | $0.744 \pm 0.010$ |
| ROC(C) | 1 | $0.794 \pm 0.022^*$ | $0.809 \pm 0.015^*$ | $0.808 \pm 0.012^*$ | $\mathbf{0.838 \pm 0.007}$ | $0.834 \pm 0.007$ |
| | 2 | $0.750 \pm 0.017^*$ | $0.761 \pm 0.010^*$ | $0.761 \pm 0.009^*$ | $\mathbf{0.782 \pm 0.007}$ | $0.781 \pm 0.008$ |
| | 3 | $0.723 \pm 0.013^*$ | $0.733 \pm 0.007^*$ | $0.735 \pm 0.010^*$ | $\mathbf{0.752 \pm 0.006}$ | $0.751 \pm 0.009$ |
| | 4 | $0.699 \pm 0.009^*$ | $0.709 \pm 0.009^*$ | $0.711 \pm 0.010^*$ | $\mathbf{0.726 \pm 0.006}$ | $0.724 \pm 0.008$ |

PRC and ROC evaluations are reported separately for targets representing infections (I) and comorbidities (C). The two-sample $t$-test for a difference in means is conducted on the results. An asterisk next to the comparator result is used to indicate a statistically significant difference in means ($p$-value $< 0.05$) relative to the TEA result.

Table 14: Summary results for TEA and comparators on RNN model with UKCF (Bold indicates best)

| | Base | REG | FEA | TEA | F/TEA |
|---|---|---|---|---|---|
| PRC(I) | $0.411 \pm 0.035^*$ | $0.415 \pm 0.030^*$ | $0.410 \pm 0.033^*$ | $\mathbf{0.483 \pm 0.045}$ | $0.457 \pm 0.037$ |
| PRC(C) | $0.497 \pm 0.057^*$ | $0.518 \pm 0.052^*$ | $0.521 \pm 0.054^*$ | $\mathbf{0.583 \pm 0.072}$ | $0.576 \pm 0.071$ |
| ROC(I) | $0.750 \pm 0.028^*$ | $0.755 \pm 0.025$ | $0.755 \pm 0.029$ | $\mathbf{0.779 \pm 0.031}$ | $0.774 \pm 0.030$ |
| ROC(C) | $0.742 \pm 0.038$ | $0.753 \pm 0.039$ | $0.754 \pm 0.037$ | $\mathbf{0.774 \pm 0.042}$ | $0.772 \pm 0.042$ |

PRC and ROC evaluations are reported separately for targets representing infections (I) and comorbidities (C). The two-sample $t$-test for a difference in means is conducted on the results. An asterisk next to the comparator result is used to indicate a statistically significant difference in means ($p$-value $< 0.05$) relative to the TEA result. Results are grouped over the temporal axis; note that the variance between splits is an artifact of this grouping.

Table 15: Extended source of gain and variants on RNN model with UKCF (Bold indicates best)

|  | $\tau$ | No Joint | No Staged | TEA | TEA(L) | TEA(LP) |
|---|---|---|---|---|---|---|
| PRC(I) | 1 | $0.511 \pm 0.014$ | $0.468 \pm 0.015$ | $0.549 \pm 0.014$ | $\mathbf{0.553 \pm 0.009}$ | $0.545 \pm 0.011$ |
|  | 2 | $0.461 \pm 0.013$ | $0.429 \pm 0.011$ | $0.490 \pm 0.012$ | $\mathbf{0.492 \pm 0.013}$ | $0.487 \pm 0.012$ |
|  | 3 | $0.434 \pm 0.016$ | $0.407 \pm 0.011$ | $\mathbf{0.457 \pm 0.010}$ | $0.457 \pm 0.013$ | $0.455 \pm 0.014$ |
|  | 4 | $0.414 \pm 0.009$ | $0.392 \pm 0.011$ | $\mathbf{0.434 \pm 0.007}$ | $0.432 \pm 0.008$ | $0.433 \pm 0.009$ |
| PRC(C) | 1 | $0.682 \pm 0.011$ | $0.633 \pm 0.032$ | $0.695 \pm 0.010$ | $\mathbf{0.697 \pm 0.010}$ | $0.695 \pm 0.012$ |
|  | 2 | $0.581 \pm 0.012$ | $0.549 \pm 0.025$ | $0.591 \pm 0.014$ | $0.589 \pm 0.013$ | $\mathbf{0.592 \pm 0.011}$ |
|  | 3 | $0.530 \pm 0.010$ | $0.506 \pm 0.018$ | $0.537 \pm 0.007$ | $0.534 \pm 0.009$ | $\mathbf{0.538 \pm 0.009}$ |
|  | 4 | $0.504 \pm 0.009$ | $0.484 \pm 0.015$ | $\mathbf{0.510 \pm 0.007}$ | $0.505 \pm 0.008$ | $0.509 \pm 0.010$ |
| ROC(I) | 1 | $0.816 \pm 0.004$ | $0.795 \pm 0.008$ | $\mathbf{0.827 \pm 0.007}$ | $0.825 \pm 0.005$ | $0.822 \pm 0.010$ |
|  | 2 | $0.774 \pm 0.010$ | $0.759 \pm 0.007$ | $\mathbf{0.783 \pm 0.008}$ | $0.782 \pm 0.008$ | $0.778 \pm 0.007$ |
|  | 3 | $0.749 \pm 0.010$ | $0.744 \pm 0.008$ | $\mathbf{0.760 \pm 0.007}$ | $0.758 \pm 0.006$ | $0.758 \pm 0.006$ |
|  | 4 | $0.732 \pm 0.008$ | $0.735 \pm 0.008$ | $\mathbf{0.748 \pm 0.008}$ | $0.739 \pm 0.007$ | $0.745 \pm 0.004$ |
| ROC(C) | 1 | $0.830 \pm 0.008$ | $0.816 \pm 0.011$ | $0.838 \pm 0.007$ | $\mathbf{0.839 \pm 0.008}$ | $\mathbf{0.839 \pm 0.007}$ |
|  | 2 | $0.775 \pm 0.010$ | $0.767 \pm 0.009$ | $0.782 \pm 0.007$ | $\mathbf{0.784 \pm 0.010}$ | $0.782 \pm 0.005$ |
|  | 3 | $0.743 \pm 0.009$ | $0.735 \pm 0.007$ | $\mathbf{0.752 \pm 0.006}$ | $0.751 \pm 0.010$ | $0.751 \pm 0.007$ |
|  | 4 | $0.721 \pm 0.006$ | $0.712 \pm 0.007$ | $\mathbf{0.726 \pm 0.006}$ | $0.724 \pm 0.007$ | $\mathbf{0.726 \pm 0.006}$ |

PRC and ROC evaluations are reported separately for targets representing infections (I) and comorbidities (C). The "No Joint" setting isolates the benefit from staged training only (analogous to basic unsupervised pretraining, though using targets); the "No Staged" setting isolates the benefit from joint training only (without pretraining).

Table 16: Summary source of gain and variants on RNN model with UKCF (Bold indicates best)

|  | No Joint | No Staged | TEA | TEA(L) | TEA(LP) |
|---|---|---|---|---|---|
| PRC(I) | $0.455 \pm 0.039$ | $0.424 \pm 0.031$ | $\mathbf{0.483 \pm 0.045}$ | $\mathbf{0.483 \pm 0.047}$ | $0.480 \pm 0.044$ |
| PRC(C) | $0.574 \pm 0.069$ | $0.543 \pm 0.061$ | $\mathbf{0.583 \pm 0.072}$ | $0.581 \pm 0.074$ | $\mathbf{0.583 \pm 0.072}$ |
| ROC(I) | $0.768 \pm 0.033$ | $0.758 \pm 0.024$ | $\mathbf{0.779 \pm 0.031}$ | $0.776 \pm 0.033$ | $0.776 \pm 0.030$ |
| ROC(C) | $0.767 \pm 0.042$ | $0.758 \pm 0.040$ | $\mathbf{0.774 \pm 0.042}$ | $\mathbf{0.774 \pm 0.044}$ | $\mathbf{0.774 \pm 0.043}$ |

PRC and ROC evaluations are reported separately for targets representing infections (I) and comorbidities (C). The "No Joint" setting isolates the benefit from staged training only (analogous to basic unsupervised pretraining, though using targets); the "No Staged" setting isolates the benefit from joint training only (without pretraining). Results are grouped over the temporal axis; note that the variance between splits is an artifact of this grouping.

Table 17: Extended results for TEA and comparators on RNN model with ADNI (Bold indicates best)

|  | $\tau$ | Base | REG | FEA | TEA | F/TEA |
|---|---|---|---|---|---|---|
| MSE(B) | 1 | $0.095 \pm 0.014$* | $0.088 \pm 0.010$* | $0.082 \pm 0.007$* | $\mathbf{0.057 \pm 0.007}$ | $0.065 \pm 0.008$* |
|  | 2 | $0.097 \pm 0.015$* | $0.089 \pm 0.010$* | $0.084 \pm 0.008$* | $\mathbf{0.057 \pm 0.007}$ | $0.066 \pm 0.007$* |
|  | 3 | $0.100 \pm 0.015$* | $0.092 \pm 0.010$* | $0.087 \pm 0.008$* | $\mathbf{0.059 \pm 0.007}$ | $0.068 \pm 0.007$* |
|  | 4 | $0.104 \pm 0.016$* | $0.095 \pm 0.011$* | $0.091 \pm 0.008$* | $\mathbf{0.061 \pm 0.007}$ | $0.071 \pm 0.008$* |
|  | 5 | $0.105 \pm 0.017$* | $0.097 \pm 0.012$* | $0.093 \pm 0.008$* | $\mathbf{0.062 \pm 0.008}$ | $0.073 \pm 0.008$* |
|  | 6 | $0.109 \pm 0.017$* | $0.100 \pm 0.013$* | $0.097 \pm 0.009$* | $\mathbf{0.065 \pm 0.008}$ | $0.076 \pm 0.009$* |
|  | 7 | $0.112 \pm 0.019$* | $0.103 \pm 0.014$* | $0.101 \pm 0.011$* | $\mathbf{0.068 \pm 0.009}$ | $0.080 \pm 0.009$* |
|  | 8 | $0.115 \pm 0.021$* | $0.106 \pm 0.016$* | $0.105 \pm 0.013$* | $\mathbf{0.072 \pm 0.011}$ | $0.083 \pm 0.010$* |
| MSE(C) | 1 | $0.275 \pm 0.013$* | $0.270 \pm 0.013$* | $0.265 \pm 0.011$* | $\mathbf{0.239 \pm 0.015}$ | $0.243 \pm 0.013$ |
|  | 2 | $0.300 \pm 0.015$* | $0.295 \pm 0.013$* | $0.290 \pm 0.012$* | $\mathbf{0.265 \pm 0.014}$ | $0.273 \pm 0.014$ |
|  | 3 | $0.323 \pm 0.018$* | $0.320 \pm 0.015$* | $0.314 \pm 0.013$* | $\mathbf{0.287 \pm 0.014}$ | $0.297 \pm 0.015$ |
|  | 4 | $0.358 \pm 0.019$* | $0.354 \pm 0.018$* | $0.352 \pm 0.017$* | $\mathbf{0.322 \pm 0.015}$ | $0.333 \pm 0.018$ |
|  | 5 | $0.371 \pm 0.024$* | $0.370 \pm 0.023$* | $0.367 \pm 0.023$* | $\mathbf{0.341 \pm 0.019}$ | $0.350 \pm 0.021$ |
|  | 6 | $0.393 \pm 0.033$ | $0.393 \pm 0.032$ | $0.391 \pm 0.034$ | $\mathbf{0.366 \pm 0.026}$ | $0.374 \pm 0.028$ |
|  | 7 | $0.417 \pm 0.043$ | $0.419 \pm 0.040$ | $0.417 \pm 0.044$ | $\mathbf{0.394 \pm 0.035}$ | $0.399 \pm 0.038$ |

| | 8 | $0.453 \pm 0.058$ | $0.455 \pm 0.054$ | $0.454 \pm 0.062$ | $\mathbf{0.430 \pm 0.048}$ | $0.435 \pm 0.057$ |

MSE evaluations reported separately for targets representing quantitative biomarkers (B) and cognitive tests (C). The two-sample $t$-test for a difference in means is conducted on the results. An asterisk next to the comparator result is used to indicate a statistically significant difference in means ($p$-value $< 0.05$) relative to the TEA result.

Table 18: Summary results for TEA and comparators on RNN model with ADNI (Bold indicates best)

| | Base | REG | FEA | TEA | F/TEA |
|---|---|---|---|---|---|
| MSE(B) | $0.105 \pm 0.018^*$ | $0.096 \pm 0.014^*$ | $0.092 \pm 0.012^*$ | $\mathbf{0.063 \pm 0.010}$ | $0.073 \pm 0.010^*$ |
| MSE(C) | $0.361 \pm 0.064$ | $0.360 \pm 0.066$ | $0.356 \pm 0.068$ | $\mathbf{0.330 \pm 0.066}$ | $0.338 \pm 0.067$ |

MSE evaluations reported separately for targets representing quantitative biomarkers (B) and cognitive tests (C). The two-sample $t$-test for a difference in means is conducted on the results. An asterisk next to the comparator result is used to indicate a statistically significant difference in means ($p$-value $< 0.05$) relative to the TEA result. Results are grouped over the temporal axis; note that the variance between splits is an artifact of this grouping.

Table 19: Extended source of gain and variants on RNN model with ADNI (Bold indicates best)

| | $\tau$ | No Joint | No Staged | TEA | TEA(L) | TEA(LP) |
|---|---|---|---|---|---|---|
| MSE(B) | 1 | $0.081 \pm 0.011$ | $0.098 \pm 0.018$ | $0.057 \pm 0.007$ | $\mathbf{0.049 \pm 0.009}$ | $0.057 \pm 0.009$ |
| | 2 | $0.084 \pm 0.011$ | $0.098 \pm 0.018$ | $0.057 \pm 0.007$ | $\mathbf{0.051 \pm 0.010}$ | $0.058 \pm 0.008$ |
| | 3 | $0.087 \pm 0.011$ | $0.101 \pm 0.019$ | $0.059 \pm 0.007$ | $\mathbf{0.054 \pm 0.011}$ | $0.059 \pm 0.008$ |
| | 4 | $0.090 \pm 0.011$ | $0.105 \pm 0.019$ | $0.061 \pm 0.007$ | $\mathbf{0.056 \pm 0.011}$ | $0.062 \pm 0.008$ |
| | 5 | $0.092 \pm 0.011$ | $0.106 \pm 0.021$ | $0.062 \pm 0.008$ | $\mathbf{0.059 \pm 0.011}$ | $0.064 \pm 0.009$ |
| | 6 | $0.097 \pm 0.012$ | $0.110 \pm 0.023$ | $0.065 \pm 0.008$ | $\mathbf{0.063 \pm 0.011}$ | $0.068 \pm 0.010$ |
| | 7 | $0.100 \pm 0.013$ | $0.113 \pm 0.025$ | $0.068 \pm 0.009$ | $\mathbf{0.066 \pm 0.010}$ | $0.072 \pm 0.011$ |
| | 8 | $0.104 \pm 0.014$ | $0.117 \pm 0.027$ | $0.072 \pm 0.011$ | $\mathbf{0.070 \pm 0.011}$ | $0.076 \pm 0.013$ |
| MSE(C) | 1 | $0.258 \pm 0.016$ | $0.274 \pm 0.017$ | $0.239 \pm 0.015$ | $\mathbf{0.231 \pm 0.020}$ | $0.241 \pm 0.016$ |
| | 2 | $0.285 \pm 0.016$ | $0.297 \pm 0.017$ | $0.265 \pm 0.014$ | $\mathbf{0.258 \pm 0.021}$ | $0.266 \pm 0.018$ |
| | 3 | $0.311 \pm 0.017$ | $0.321 \pm 0.018$ | $0.287 \pm 0.014$ | $\mathbf{0.282 \pm 0.021}$ | $0.287 \pm 0.018$ |
| | 4 | $0.346 \pm 0.019$ | $0.356 \pm 0.020$ | $0.322 \pm 0.015$ | $\mathbf{0.319 \pm 0.021}$ | $0.321 \pm 0.022$ |
| | 5 | $0.363 \pm 0.024$ | $0.373 \pm 0.024$ | $0.341 \pm 0.019$ | $\mathbf{0.337 \pm 0.024}$ | $0.338 \pm 0.026$ |
| | 6 | $0.389 \pm 0.033$ | $0.397 \pm 0.031$ | $0.366 \pm 0.026$ | $0.366 \pm 0.030$ | $\mathbf{0.362 \pm 0.033}$ |
| | 7 | $0.416 \pm 0.041$ | $0.424 \pm 0.040$ | $0.394 \pm 0.035$ | $0.401 \pm 0.043$ | $\mathbf{0.390 \pm 0.044}$ |
| | 8 | $0.454 \pm 0.059$ | $0.462 \pm 0.053$ | $0.430 \pm 0.048$ | $0.447 \pm 0.064$ | $\mathbf{0.427 \pm 0.063}$ |

MSE evaluations reported separately for targets representing quantitative biomarkers (B) and cognitive tests (C). The "No Joint" setting isolates the benefit from staged training only (analogous to basic unsupervised pretraining, though using targets); the "No Staged" setting isolates the benefit from joint training only (without pretraining).

Table 20: Summary source of gain and variants on RNN model with ADNI (Bold indicates best)

| | No Joint | No Staged | TEA | TEA(L) | TEA(LP) |
|---|---|---|---|---|---|
| MSE(B) | $0.092 \pm 0.014$ | $0.106 \pm 0.022$ | $0.063 \pm 0.010$ | $\mathbf{0.058 \pm 0.012}$ | $0.064 \pm 0.012$ |
| MSE(C) | $0.353 \pm 0.070$ | $0.363 \pm 0.067$ | $0.330 \pm 0.066$ | $0.330 \pm 0.076$ | $\mathbf{0.329 \pm 0.068}$ |

MSE evaluations reported separately for targets representing quantitative biomarkers (B) and cognitive tests (C). The "No Joint" setting isolates the benefit from staged training only (analogous to basic unsupervised pretraining, though using targets); the "No Staged" setting isolates the benefit from joint training only (without pretraining). Results are grouped over the temporal axis; note that the variance between splits is an artifact of this grouping.

Table 21: Extended results for TEA and comparators on RNN model with MIMIC (Bold indicates best)

| | $\tau$ | Base | REG | FEA | TEA | F/TEA |
|---|---|---|---|---|---|---|
| PRC | 1 | $0.159 \pm 0.034^*$ | $0.159 \pm 0.022^*$ | $0.162 \pm 0.036^*$ | $\mathbf{0.293 \pm 0.031}$ | $0.193 \pm 0.023^*$ |
| | 2 | $0.148 \pm 0.028^*$ | $0.149 \pm 0.018^*$ | $0.150 \pm 0.030^*$ | $\mathbf{0.254 \pm 0.025}$ | $0.174 \pm 0.018^*$ |
| | 3 | $0.139 \pm 0.024^*$ | $0.141 \pm 0.015^*$ | $0.142 \pm 0.027^*$ | $\mathbf{0.230 \pm 0.021}$ | $0.162 \pm 0.015^*$ |
| | 4 | $0.133 \pm 0.021^*$ | $0.135 \pm 0.012^*$ | $0.135 \pm 0.024^*$ | $\mathbf{0.214 \pm 0.018}$ | $0.153 \pm 0.012^*$ |
| | 5 | $0.129 \pm 0.019^*$ | $0.130 \pm 0.011^*$ | $0.130 \pm 0.022^*$ | $\mathbf{0.203 \pm 0.015}$ | $0.147 \pm 0.011^*$ |
| ROC | 1 | $0.699 \pm 0.049^*$ | $0.704 \pm 0.028^*$ | $0.709 \pm 0.060^*$ | $\mathbf{0.801 \pm 0.018}$ | $0.745 \pm 0.021^*$ |

| | τ | Base | REG | FEA | TEA | F/TEA |
|---|---|---|---|---|---|---|
| | 2 | 0.701 ± 0.044* | 0.707 ± 0.025* | 0.705 ± 0.050* | **0.778 ± 0.015** | 0.740 ± 0.018* |
| | 3 | 0.690 ± 0.041* | 0.696 ± 0.024* | 0.693 ± 0.046* | **0.758 ± 0.016** | 0.726 ± 0.019* |
| | 4 | 0.681 ± 0.038* | 0.688 ± 0.023* | 0.684 ± 0.042* | **0.745 ± 0.015** | 0.715 ± 0.019* |
| | 5 | 0.679 ± 0.037* | 0.685 ± 0.023* | 0.680 ± 0.043* | **0.736 ± 0.012** | 0.713 ± 0.019* |
| MSE | 1 | 0.141 ± 0.007 | 0.140 ± 0.006 | 0.138 ± 0.010 | **0.137 ± 0.008** | 0.139 ± 0.007 |
| | 2 | 0.159 ± 0.010 | 0.159 ± 0.007 | 0.160 ± 0.008 | **0.154 ± 0.009** | 0.162 ± 0.007* |
| | 3 | 0.156 ± 0.009 | **0.155 ± 0.007** | 0.156 ± 0.008 | 0.158 ± 0.008 | 0.158 ± 0.008 |
| | 4 | 0.154 ± 0.008 | **0.153 ± 0.008** | **0.153 ± 0.008** | **0.153 ± 0.009** | 0.155 ± 0.009 |
| | 5 | 0.154 ± 0.010 | 0.152 ± 0.010 | 0.152 ± 0.010 | **0.150 ± 0.011** | 0.155 ± 0.010 |

The two-sample $t$-test for a difference in means is conducted on the results. An asterisk next to the comparator result is used to indicate a statistically significant difference in means ($p$-value $< 0.05$) relative to the TEA result.

Table 22: Summary results for TEA and comparators on RNN model with MIMIC (Bold indicates best)

| | Base | REG | FEA | TEA | F/TEA |
|---|---|---|---|---|---|
| PRC | 0.142 ± 0.028* | 0.143 ± 0.019* | 0.144 ± 0.030* | **0.239 ± 0.039** | 0.166 ± 0.023* |
| ROC | 0.690 ± 0.043* | 0.696 ± 0.026* | 0.694 ± 0.050* | **0.763 ± 0.028** | 0.728 ± 0.023* |
| MSE | 0.153 ± 0.011 | 0.152 ± 0.010 | 0.152 ± 0.012 | **0.150 ± 0.012** | 0.154 ± 0.011 |

The two-sample $t$-test for a difference in means is conducted on the results. An asterisk next to the comparator result is used to indicate a statistically significant difference in means ($p$-value $< 0.05$) relative to the TEA result. Results are grouped over the temporal axis; note that the variance between splits is an artifact of this grouping.

Table 23: Extended source of gain and variants on RNN model with MIMIC (Bold indicates best)

| | τ | No Joint | No Staged | TEA | TEA(L) | TEA(LP) |
|---|---|---|---|---|---|---|
| PRC | 1 | 0.216 ± 0.044 | 0.194 ± 0.020 | 0.293 ± 0.031 | **0.310 ± 0.047** | 0.280 ± 0.033 |
| | 2 | 0.194 ± 0.035 | 0.175 ± 0.017 | 0.254 ± 0.025 | **0.265 ± 0.035** | 0.242 ± 0.026 |
| | 3 | 0.178 ± 0.030 | 0.163 ± 0.013 | 0.230 ± 0.021 | **0.239 ± 0.030** | 0.221 ± 0.022 |
| | 4 | 0.168 ± 0.027 | 0.154 ± 0.011 | 0.214 ± 0.018 | **0.222 ± 0.026** | 0.206 ± 0.020 |
| | 5 | 0.160 ± 0.024 | 0.148 ± 0.010 | 0.203 ± 0.015 | **0.210 ± 0.023** | 0.195 ± 0.018 |
| ROC | 1 | 0.756 ± 0.042 | 0.742 ± 0.022 | 0.801 ± 0.018 | **0.807 ± 0.025** | 0.791 ± 0.018 |
| | 2 | 0.741 ± 0.031 | 0.738 ± 0.021 | 0.778 ± 0.015 | **0.783 ± 0.017** | 0.773 ± 0.012 |
| | 3 | 0.726 ± 0.031 | 0.724 ± 0.019 | 0.758 ± 0.016 | **0.761 ± 0.019** | 0.756 ± 0.013 |
| | 4 | 0.715 ± 0.030 | 0.715 ± 0.019 | 0.745 ± 0.015 | **0.747 ± 0.017** | 0.742 ± 0.011 |
| | 5 | 0.710 ± 0.031 | 0.711 ± 0.018 | 0.736 ± 0.012 | **0.741 ± 0.019** | 0.736 ± 0.014 |
| MSE | 1 | 0.138 ± 0.008 | 0.137 ± 0.007 | 0.137 ± 0.008 | **0.136 ± 0.006** | 0.137 ± 0.008 |
| | 2 | 0.158 ± 0.007 | 0.156 ± 0.012 | 0.154 ± 0.009 | **0.153 ± 0.007** | 0.154 ± 0.007 |
| | 3 | **0.155 ± 0.009** | 0.156 ± 0.010 | 0.158 ± 0.008 | 0.156 ± 0.010 | 0.158 ± 0.008 |
| | 4 | 0.152 ± 0.008 | 0.153 ± 0.009 | 0.153 ± 0.009 | **0.151 ± 0.011** | 0.154 ± 0.008 |
| | 5 | 0.151 ± 0.009 | 0.150 ± 0.009 | 0.150 ± 0.011 | **0.147 ± 0.011** | 0.150 ± 0.008 |

The "No Joint" setting isolates the benefit from staged training only (analogous to basic unsupervised pretraining, though using targets); the "No Staged" setting isolates the benefit from joint training only (without pretraining).

Table 24: Summary source of gain and variants on RNN model with MIMIC (Bold indicates best)

| | No Joint | No Staged | TEA | TEA(L) | TEA(LP) |
|---|---|---|---|---|---|
| PRC | 0.183 ± 0.038 | 0.167 ± 0.022 | 0.239 ± 0.039 | **0.249 ± 0.049** | 0.229 ± 0.039 |
| ROC | 0.730 ± 0.038 | 0.726 ± 0.023 | 0.763 ± 0.028 | **0.768 ± 0.031** | 0.759 ± 0.025 |
| MSE | 0.151 ± 0.011 | 0.150 ± 0.012 | 0.150 ± 0.012 | **0.149 ± 0.012** | 0.151 ± 0.011 |

The "No Joint" setting isolates the benefit from staged training only (analogous to basic unsupervised pretraining, though using targets); the "No Staged" setting isolates the benefit from joint training only (without pretraining). Results are grouped over the temporal axis; note that the variance between splits is an artifact of this grouping.

### E.3 Results for Sensitivities

Table 25: Extended $\nu$-Sensitivities for REG on linear model with UKCF (Bold indicates best)

| | $\tau$ | $\nu = 0$ | $\nu = 3e{-}5$ | $\nu = 3e{-}4$ | $\nu = 3e{-}3$ | $\nu = 3e{-}2$ |
|---|---|---|---|---|---|---|
| PRC(I) | 1 | $0.340 \pm 0.118$ | $0.355 \pm 0.114$ | $\mathbf{0.370 \pm 0.101}$ | $0.320 \pm 0.034$ | $0.163 \pm 0.003$ |
| | 2 | $0.325 \pm 0.099$ | $0.338 \pm 0.096$ | $\mathbf{0.350 \pm 0.085}$ | $0.309 \pm 0.026$ | $0.176 \pm 0.004$ |
| | 3 | $0.314 \pm 0.088$ | $0.327 \pm 0.086$ | $\mathbf{0.337 \pm 0.076}$ | $0.300 \pm 0.024$ | $0.182 \pm 0.005$ |
| | 4 | $0.307 \pm 0.082$ | $0.318 \pm 0.080$ | $\mathbf{0.328 \pm 0.071}$ | $0.293 \pm 0.023$ | $0.184 \pm 0.004$ |
| PRC(C) | 1 | $0.445 \pm 0.131$ | $0.460 \pm 0.125$ | $\mathbf{0.467 \pm 0.106}$ | $0.426 \pm 0.034$ | $0.240 \pm 0.004$ |
| | 2 | $0.418 \pm 0.099$ | $0.428 \pm 0.095$ | $\mathbf{0.435 \pm 0.081}$ | $0.409 \pm 0.025$ | $0.260 \pm 0.005$ |
| | 3 | $0.403 \pm 0.082$ | $0.412 \pm 0.078$ | $\mathbf{0.418 \pm 0.067}$ | $0.399 \pm 0.021$ | $0.272 \pm 0.006$ |
| | 4 | $0.398 \pm 0.073$ | $0.406 \pm 0.070$ | $\mathbf{0.412 \pm 0.060}$ | $0.397 \pm 0.019$ | $0.281 \pm 0.007$ |
| ROC(I) | 1 | $0.713 \pm 0.104$ | $0.724 \pm 0.100$ | $\mathbf{0.737 \pm 0.081}$ | $0.715 \pm 0.022$ | $0.527 \pm 0.006$ |
| | 2 | $0.688 \pm 0.089$ | $0.697 \pm 0.086$ | $\mathbf{0.709 \pm 0.070}$ | $0.693 \pm 0.022$ | $0.532 \pm 0.008$ |
| | 3 | $0.677 \pm 0.080$ | $0.686 \pm 0.078$ | $\mathbf{0.697 \pm 0.065}$ | $0.683 \pm 0.021$ | $0.543 \pm 0.007$ |
| | 4 | $0.677 \pm 0.076$ | $0.686 \pm 0.074$ | $\mathbf{0.696 \pm 0.063}$ | $0.681 \pm 0.019$ | $0.555 \pm 0.007$ |
| ROC(C) | 1 | $0.713 \pm 0.110$ | $0.727 \pm 0.105$ | $\mathbf{0.741 \pm 0.088}$ | $0.716 \pm 0.025$ | $0.512 \pm 0.010$ |
| | 2 | $0.686 \pm 0.091$ | $0.696 \pm 0.086$ | $\mathbf{0.707 \pm 0.072}$ | $0.686 \pm 0.022$ | $0.523 \pm 0.008$ |
| | 3 | $0.668 \pm 0.077$ | $0.678 \pm 0.074$ | $\mathbf{0.686 \pm 0.061}$ | $0.668 \pm 0.021$ | $0.530 \pm 0.013$ |
| | 4 | $0.651 \pm 0.070$ | $0.660 \pm 0.066$ | $\mathbf{0.667 \pm 0.056}$ | $0.652 \pm 0.018$ | $0.527 \pm 0.015$ |

The $\nu$ coefficient controls the strength of $\ell_2$-regularization applied on top of the original loss function minimized. PRC and ROC evaluations are reported separately for targets representing infections (I) and comorbidities (C).

Table 26: Summary $\nu$-Sensitivities for REG on linear model with UKCF (Bold indicates best)

| | $\nu = 0$ | $\nu = 3e{-}5$ | $\nu = 3e{-}4$ | $\nu = 3e{-}3$ | $\nu = 3e{-}2$ |
|---|---|---|---|---|---|
| PRC(I) | $0.322 \pm 0.099$ | $0.335 \pm 0.096$ | $\mathbf{0.347 \pm 0.085}$ | $0.305 \pm 0.029$ | $0.176 \pm 0.009$ |
| PRC(C) | $0.416 \pm 0.100$ | $0.426 \pm 0.097$ | $\mathbf{0.433 \pm 0.083}$ | $0.408 \pm 0.028$ | $0.263 \pm 0.016$ |
| ROC(I) | $0.689 \pm 0.089$ | $0.698 \pm 0.087$ | $\mathbf{0.710 \pm 0.072}$ | $0.693 \pm 0.025$ | $0.540 \pm 0.013$ |
| ROC(C) | $0.679 \pm 0.091$ | $0.690 \pm 0.087$ | $\mathbf{0.700 \pm 0.075}$ | $0.681 \pm 0.032$ | $0.523 \pm 0.013$ |

The $\nu$ coefficient controls the strength of $\ell_2$-regularization applied on top of the original loss function minimized. PRC and ROC evaluations are reported separately for targets representing infections (I) and comorbidities (C). Results are grouped over the temporal axis; note that the variance between splits is an artifact of this grouping.

Table 27: Extended $\nu$-Sensitivities for TEA on linear model with UKCF (Bold indicates best)

| | $\tau$ | $\nu = 0$ | $\nu = 3e{-}5$ | $\nu = 3e{-}4$ | $\nu = 3e{-}3$ | $\nu = 3e{-}2$ |
|---|---|---|---|---|---|---|
| PRC(I) | 1 | $0.484 \pm 0.020$ | $0.489 \pm 0.019$ | $\mathbf{0.497 \pm 0.016}$ | $0.442 \pm 0.005$ | $0.174 \pm 0.004$ |
| | 2 | $0.450 \pm 0.016$ | $0.453 \pm 0.016$ | $\mathbf{0.459 \pm 0.017}$ | $0.414 \pm 0.007$ | $0.186 \pm 0.005$ |
| | 3 | $0.424 \pm 0.014$ | $0.426 \pm 0.013$ | $\mathbf{0.432 \pm 0.015}$ | $0.394 \pm 0.009$ | $0.192 \pm 0.005$ |
| | 4 | $0.405 \pm 0.012$ | $0.407 \pm 0.010$ | $\mathbf{0.413 \pm 0.010}$ | $0.381 \pm 0.008$ | $0.193 \pm 0.004$ |
| PRC(C) | 1 | $0.641 \pm 0.021$ | $0.644 \pm 0.019$ | $\mathbf{0.653 \pm 0.009}$ | $0.612 \pm 0.009$ | $0.276 \pm 0.007$ |
| | 2 | $0.561 \pm 0.013$ | $0.562 \pm 0.011$ | $\mathbf{0.566 \pm 0.010}$ | $0.544 \pm 0.007$ | $0.293 \pm 0.008$ |
| | 3 | $0.519 \pm 0.008$ | $0.519 \pm 0.007$ | $\mathbf{0.521 \pm 0.008}$ | $0.508 \pm 0.005$ | $0.302 \pm 0.008$ |
| | 4 | $0.495 \pm 0.008$ | $0.496 \pm 0.007$ | $\mathbf{0.498 \pm 0.009}$ | $0.489 \pm 0.007$ | $0.309 \pm 0.009$ |
| ROC(I) | 1 | $0.800 \pm 0.015$ | $0.803 \pm 0.015$ | $\mathbf{0.806 \pm 0.007}$ | $0.779 \pm 0.007$ | $0.555 \pm 0.005$ |
| | 2 | $0.765 \pm 0.009$ | $0.767 \pm 0.011$ | $\mathbf{0.771 \pm 0.007}$ | $0.751 \pm 0.007$ | $0.557 \pm 0.007$ |
| | 3 | $0.746 \pm 0.008$ | $0.747 \pm 0.008$ | $\mathbf{0.750 \pm 0.007}$ | $0.735 \pm 0.007$ | $0.565 \pm 0.006$ |
| | 4 | $0.736 \pm 0.006$ | $0.737 \pm 0.007$ | $\mathbf{0.740 \pm 0.008}$ | $0.727 \pm 0.005$ | $0.575 \pm 0.006$ |
| ROC(C) | 1 | $0.825 \pm 0.011$ | $0.826 \pm 0.010$ | $\mathbf{0.829 \pm 0.006}$ | $0.819 \pm 0.006$ | $0.560 \pm 0.011$ |
| | 2 | $0.772 \pm 0.010$ | $0.774 \pm 0.009$ | $\mathbf{0.775 \pm 0.008}$ | $0.771 \pm 0.006$ | $0.564 \pm 0.009$ |
| | 3 | $0.742 \pm 0.004$ | $\mathbf{0.744 \pm 0.004}$ | $0.743 \pm 0.005$ | $0.741 \pm 0.004$ | $0.566 \pm 0.012$ |
| | 4 | $0.718 \pm 0.006$ | $\mathbf{0.720 \pm 0.006}$ | $0.720 \pm 0.006$ | $0.717 \pm 0.008$ | $0.561 \pm 0.015$ |

The $\nu$ coefficient controls the strength of $\ell_2$-regularization applied on top of the original loss function minimized. PRC and ROC evaluations are reported separately for targets representing infections (I) and comorbidities (C).

Table 28: Summary $\nu$-Sensitivities for TEA on linear model with UKCF (Bold indicates best)

| | $\nu = 0$ | $\nu = 3\mathrm{e}{-5}$ | $\nu = 3\mathrm{e}{-4}$ | $\nu = 3\mathrm{e}{-3}$ | $\nu = 3\mathrm{e}{-2}$ |
|---|---|---|---|---|---|
| PRC(I) | $0.441 \pm 0.033$ | $0.444 \pm 0.034$ | $\mathbf{0.450 \pm 0.035}$ | $0.408 \pm 0.024$ | $0.186 \pm 0.009$ |
| PRC(C) | $0.554 \pm 0.057$ | $0.555 \pm 0.058$ | $\mathbf{0.559 \pm 0.060}$ | $0.538 \pm 0.047$ | $0.295 \pm 0.015$ |
| ROC(I) | $0.762 \pm 0.026$ | $0.763 \pm 0.028$ | $\mathbf{0.767 \pm 0.026}$ | $0.748 \pm 0.021$ | $0.563 \pm 0.010$ |
| ROC(C) | $0.764 \pm 0.041$ | $0.766 \pm 0.041$ | $\mathbf{0.767 \pm 0.042}$ | $0.762 \pm 0.039$ | $0.563 \pm 0.012$ |

The $\nu$ coefficient controls the strength of $\ell_2$-regularization applied on top of the original loss function minimized. PRC and ROC evaluations are reported separately for targets representing infections (I) and comorbidities (C). Results are grouped over the temporal axis; note that the variance between splits is an artifact of this grouping.

Table 29: Extended $\lambda$-Sensitivities for TEA on linear model with UKCF (Bold indicates best)

| | $\tau$ | $\lambda = 0$ | $\lambda = 0.01$ | $\lambda = 0.1$ | $\lambda = 0.5$ | $\lambda = 0.9$ | $\lambda = 0.99$ | $\lambda = 1$ |
|---|---|---|---|---|---|---|---|---|
| PRC(I) | 1 | $0.370 \pm 0.101$ | $0.383 \pm 0.106$ | $0.412 \pm 0.091$ | $\mathbf{0.472 \pm 0.016}$ | $0.461 \pm 0.012$ | $0.327 \pm 0.011$ | $0.150 \pm 0.008$ |
| | 2 | $0.350 \pm 0.085$ | $0.361 \pm 0.090$ | $0.386 \pm 0.076$ | $\mathbf{0.439 \pm 0.016}$ | $0.431 \pm 0.012$ | $0.323 \pm 0.011$ | $0.162 \pm 0.008$ |
| | 3 | $0.337 \pm 0.076$ | $0.347 \pm 0.081$ | $0.368 \pm 0.068$ | $\mathbf{0.415 \pm 0.015}$ | $0.410 \pm 0.009$ | $0.316 \pm 0.012$ | $0.168 \pm 0.007$ |
| | 4 | $0.328 \pm 0.071$ | $0.337 \pm 0.075$ | $0.357 \pm 0.064$ | $\mathbf{0.399 \pm 0.011}$ | $0.395 \pm 0.008$ | $0.307 \pm 0.011$ | $0.170 \pm 0.007$ |
| PRC(C) | 1 | $0.467 \pm 0.106$ | $0.481 \pm 0.110$ | $0.528 \pm 0.104$ | $\mathbf{0.620 \pm 0.030}$ | $\mathbf{0.620 \pm 0.016}$ | $0.433 \pm 0.012$ | $0.236 \pm 0.012$ |
| | 2 | $0.435 \pm 0.081$ | $0.445 \pm 0.084$ | $0.481 \pm 0.079$ | $0.550 \pm 0.022$ | $\mathbf{0.553 \pm 0.013}$ | $0.427 \pm 0.010$ | $0.249 \pm 0.010$ |
| | 3 | $0.418 \pm 0.067$ | $0.427 \pm 0.070$ | $0.456 \pm 0.064$ | $0.512 \pm 0.017$ | $\mathbf{0.516 \pm 0.009}$ | $0.421 \pm 0.009$ | $0.259 \pm 0.011$ |
| | 4 | $0.412 \pm 0.060$ | $0.420 \pm 0.062$ | $0.445 \pm 0.057$ | $0.491 \pm 0.015$ | $\mathbf{0.494 \pm 0.009}$ | $0.415 \pm 0.009$ | $0.266 \pm 0.011$ |
| ROC(I) | 1 | $0.737 \pm 0.081$ | $0.742 \pm 0.089$ | $0.764 \pm 0.067$ | $\mathbf{0.799 \pm 0.007}$ | $0.791 \pm 0.004$ | $0.708 \pm 0.008$ | $0.499 \pm 0.013$ |
| | 2 | $0.709 \pm 0.070$ | $0.713 \pm 0.077$ | $0.733 \pm 0.058$ | $\mathbf{0.765 \pm 0.009}$ | $0.760 \pm 0.008$ | $0.694 \pm 0.007$ | $0.502 \pm 0.014$ |
| | 3 | $0.697 \pm 0.065$ | $0.701 \pm 0.071$ | $0.719 \pm 0.054$ | $\mathbf{0.750 \pm 0.008}$ | $0.746 \pm 0.008$ | $0.690 \pm 0.006$ | $0.500 \pm 0.014$ |
| | 4 | $0.696 \pm 0.063$ | $0.699 \pm 0.068$ | $0.715 \pm 0.053$ | $\mathbf{0.744 \pm 0.005}$ | $0.741 \pm 0.006$ | $0.690 \pm 0.007$ | $0.501 \pm 0.015$ |
| ROC(C) | 1 | $0.741 \pm 0.088$ | $0.747 \pm 0.092$ | $0.775 \pm 0.075$ | $\mathbf{0.821 \pm 0.011}$ | $0.819 \pm 0.006$ | $0.725 \pm 0.014$ | $0.493 \pm 0.034$ |
| | 2 | $0.707 \pm 0.072$ | $0.712 \pm 0.076$ | $0.735 \pm 0.061$ | $\mathbf{0.774 \pm 0.009}$ | $\mathbf{0.774 \pm 0.007}$ | $0.704 \pm 0.012$ | $0.496 \pm 0.027$ |
| | 3 | $0.686 \pm 0.061$ | $0.691 \pm 0.067$ | $0.711 \pm 0.052$ | $\mathbf{0.746 \pm 0.007}$ | $0.745 \pm 0.003$ | $0.690 \pm 0.011$ | $0.497 \pm 0.028$ |
| | 4 | $0.667 \pm 0.056$ | $0.672 \pm 0.059$ | $0.689 \pm 0.048$ | $\mathbf{0.722 \pm 0.007}$ | $0.721 \pm 0.005$ | $0.675 \pm 0.014$ | $0.497 \pm 0.025$ |

The $\lambda$ coefficient controls the strength of prior—i.e. the tradeoff between the prediction and reconstruction loss. PRC and ROC evaluations are reported separately for targets representing infections (I) and comorbidities (C).

Table 30: Summary $\lambda$-Sensitivities for TEA on linear model with UKCF (Bold indicates best)

| | $\lambda = 0$ | $\lambda = 0.01$ | $\lambda = 0.1$ | $\lambda = 0.5$ | $\lambda = 0.9$ | $\lambda = 0.99$ | $\lambda = 1$ |
|---|---|---|---|---|---|---|---|
| PRC(I) | $0.347 \pm 0.085$ | $0.357 \pm 0.090$ | $0.381 \pm 0.078$ | $\mathbf{0.431 \pm 0.031}$ | $0.424 \pm 0.027$ | $0.318 \pm 0.013$ | $0.162 \pm 0.011$ |
| PRC(C) | $0.433 \pm 0.083$ | $0.443 \pm 0.087$ | $0.477 \pm 0.084$ | $0.543 \pm 0.054$ | $\mathbf{0.546 \pm 0.050}$ | $0.424 \pm 0.012$ | $0.252 \pm 0.016$ |
| ROC(I) | $0.710 \pm 0.072$ | $0.714 \pm 0.078$ | $0.733 \pm 0.061$ | $\mathbf{0.764 \pm 0.022}$ | $0.759 \pm 0.020$ | $0.695 \pm 0.010$ | $0.501 \pm 0.014$ |
| ROC(C) | $0.700 \pm 0.075$ | $0.705 \pm 0.080$ | $0.727 \pm 0.068$ | $\mathbf{0.766 \pm 0.038}$ | $0.765 \pm 0.037$ | $0.698 \pm 0.022$ | $0.496 \pm 0.029$ |

The $\lambda$ coefficient controls the strength of prior—i.e. the tradeoff between the prediction and reconstruction loss. PRC and ROC evaluations are reported separately for targets representing infections (I) and comorbidities (C). Results are grouped over the temporal axis; note that the variance between splits is an artifact of this grouping.

Table 31: Extended $N$-Sensitivities for REG on linear model with UKCF (Bold indicates best)

| | $\tau$ | $N \times 1\%$ | $N \times 5\%$ | $N \times 20\%$ | $N \times 50\%$ | $N \times 100\%$ |
|---|---|---|---|---|---|---|
| PRC(I) | 1 | $0.160 \pm 0.010$ | $0.176 \pm 0.024$ | $0.199 \pm 0.044$ | $0.325 \pm 0.114$ | $\mathbf{0.370 \pm 0.101}$ |
| | 2 | $0.172 \pm 0.009$ | $0.187 \pm 0.021$ | $0.207 \pm 0.037$ | $0.312 \pm 0.095$ | $\mathbf{0.350 \pm 0.085}$ |
| | 3 | $0.180 \pm 0.011$ | $0.192 \pm 0.020$ | $0.209 \pm 0.032$ | $0.303 \pm 0.085$ | $\mathbf{0.337 \pm 0.076}$ |
| | 4 | $0.182 \pm 0.009$ | $0.193 \pm 0.020$ | $0.208 \pm 0.028$ | $0.295 \pm 0.078$ | $\mathbf{0.328 \pm 0.071}$ |
| PRC(C) | 1 | $0.246 \pm 0.011$ | $0.268 \pm 0.034$ | $0.293 \pm 0.046$ | $0.421 \pm 0.119$ | $\mathbf{0.467 \pm 0.106}$ |
| | 2 | $0.263 \pm 0.011$ | $0.283 \pm 0.027$ | $0.304 \pm 0.036$ | $0.401 \pm 0.091$ | $\mathbf{0.435 \pm 0.081}$ |
| | 3 | $0.275 \pm 0.013$ | $0.292 \pm 0.028$ | $0.313 \pm 0.032$ | $0.390 \pm 0.076$ | $\mathbf{0.418 \pm 0.067}$ |
| | 4 | $0.286 \pm 0.013$ | $0.302 \pm 0.025$ | $0.319 \pm 0.029$ | $0.384 \pm 0.066$ | $\mathbf{0.412 \pm 0.060}$ |
| ROC(I) | 1 | $0.512 \pm 0.024$ | $0.553 \pm 0.046$ | $0.598 \pm 0.057$ | $0.705 \pm 0.090$ | $\mathbf{0.737 \pm 0.081}$ |
| | 2 | $0.516 \pm 0.025$ | $0.557 \pm 0.040$ | $0.594 \pm 0.049$ | $0.681 \pm 0.077$ | $\mathbf{0.709 \pm 0.070}$ |

| | $\tau$ | $N \times 1\%$ | $N \times 5\%$ | $N \times 20\%$ | $N \times 50\%$ | $N \times 100\%$ |
|---|---|---|---|---|---|---|
| | 3 | $0.521 \pm 0.026$ | $0.549 \pm 0.037$ | $0.590 \pm 0.045$ | $0.671 \pm 0.071$ | $\mathbf{0.697 \pm 0.065}$ |
| | 4 | $0.519 \pm 0.024$ | $0.546 \pm 0.037$ | $0.590 \pm 0.039$ | $0.669 \pm 0.068$ | $\mathbf{0.696 \pm 0.063}$ |
| ROC(C) | 1 | $0.507 \pm 0.026$ | $0.539 \pm 0.049$ | $0.591 \pm 0.054$ | $0.702 \pm 0.101$ | $\mathbf{0.741 \pm 0.088}$ |
| | 2 | $0.520 \pm 0.024$ | $0.546 \pm 0.040$ | $0.587 \pm 0.041$ | $0.675 \pm 0.082$ | $\mathbf{0.707 \pm 0.072}$ |
| | 3 | $0.526 \pm 0.023$ | $0.549 \pm 0.038$ | $0.586 \pm 0.036$ | $0.659 \pm 0.070$ | $\mathbf{0.686 \pm 0.061}$ |
| | 4 | $0.528 \pm 0.027$ | $0.550 \pm 0.039$ | $0.581 \pm 0.034$ | $0.642 \pm 0.062$ | $\mathbf{0.667 \pm 0.056}$ |

The proportion of data $N$ used is randomly restricted, showing performance under various levels of data scarcity. PRC and ROC evaluations are reported separately for targets representing infections (I) and comorbidities (C).

Table 32: Summary $N$-Sensitivities for REG on linear model with UKCF (Bold indicates best)

| | $N \times 1\%$ | $N \times 5\%$ | $N \times 20\%$ | $N \times 50\%$ | $N \times 100\%$ |
|---|---|---|---|---|---|
| PRC(I) | $0.173 \pm 0.013$ | $0.187 \pm 0.022$ | $0.206 \pm 0.036$ | $0.309 \pm 0.094$ | $\mathbf{0.347 \pm 0.085}$ |
| PRC(C) | $0.267 \pm 0.019$ | $0.286 \pm 0.031$ | $0.307 \pm 0.038$ | $0.399 \pm 0.092$ | $\mathbf{0.433 \pm 0.083}$ |
| ROC(I) | $0.517 \pm 0.025$ | $0.551 \pm 0.041$ | $0.593 \pm 0.048$ | $0.682 \pm 0.078$ | $\mathbf{0.710 \pm 0.072}$ |
| ROC(C) | $0.520 \pm 0.026$ | $0.546 \pm 0.042$ | $0.586 \pm 0.042$ | $0.669 \pm 0.083$ | $\mathbf{0.700 \pm 0.075}$ |

The proportion of data $N$ used is randomly restricted, showing performance under various levels of data scarcity. PRC and ROC evaluations are reported separately for targets representing infections (I) and comorbidities (C). Results are grouped over the temporal axis; note that the variance between splits is an artifact of this grouping.

Table 33: Extended $N$-Sensitivities for TEA on linear model with UKCF (Bold indicates best)

| | $\tau$ | $N \times 1\%$ | $N \times 5\%$ | $N \times 20\%$ | $N \times 50\%$ | $N \times 100\%$ |
|---|---|---|---|---|---|---|
| PRC(I) | 1 | $0.199 \pm 0.016$ | $0.342 \pm 0.040$ | $0.453 \pm 0.020$ | $0.486 \pm 0.017$ | $\mathbf{0.497 \pm 0.016}$ |
| | 2 | $0.205 \pm 0.013$ | $0.336 \pm 0.031$ | $0.421 \pm 0.017$ | $0.448 \pm 0.015$ | $\mathbf{0.459 \pm 0.017}$ |
| | 3 | $0.212 \pm 0.018$ | $0.322 \pm 0.027$ | $0.398 \pm 0.015$ | $0.420 \pm 0.014$ | $\mathbf{0.432 \pm 0.015}$ |
| | 4 | $0.210 \pm 0.016$ | $0.308 \pm 0.025$ | $0.381 \pm 0.011$ | $0.402 \pm 0.011$ | $\mathbf{0.413 \pm 0.010}$ |
| PRC(C) | 1 | $0.287 \pm 0.021$ | $0.470 \pm 0.060$ | $0.610 \pm 0.017$ | $0.642 \pm 0.012$ | $\mathbf{0.653 \pm 0.009}$ |
| | 2 | $0.292 \pm 0.018$ | $0.445 \pm 0.041$ | $0.535 \pm 0.014$ | $0.557 \pm 0.012$ | $\mathbf{0.566 \pm 0.010}$ |
| | 3 | $0.302 \pm 0.014$ | $0.424 \pm 0.034$ | $0.495 \pm 0.011$ | $0.514 \pm 0.009$ | $\mathbf{0.521 \pm 0.008}$ |
| | 4 | $0.311 \pm 0.017$ | $0.417 \pm 0.030$ | $0.478 \pm 0.014$ | $0.493 \pm 0.010$ | $\mathbf{0.498 \pm 0.009}$ |
| ROC(I) | 1 | $0.570 \pm 0.022$ | $0.698 \pm 0.027$ | $0.776 \pm 0.008$ | $0.797 \pm 0.007$ | $\mathbf{0.806 \pm 0.007}$ |
| | 2 | $0.565 \pm 0.027$ | $0.684 \pm 0.019$ | $0.744 \pm 0.010$ | $0.763 \pm 0.007$ | $\mathbf{0.771 \pm 0.007}$ |
| | 3 | $0.568 \pm 0.027$ | $0.663 \pm 0.021$ | $0.723 \pm 0.011$ | $0.740 \pm 0.009$ | $\mathbf{0.750 \pm 0.007}$ |
| | 4 | $0.564 \pm 0.026$ | $0.652 \pm 0.020$ | $0.707 \pm 0.010$ | $0.729 \pm 0.010$ | $\mathbf{0.740 \pm 0.008}$ |
| ROC(C) | 1 | $0.581 \pm 0.017$ | $0.715 \pm 0.032$ | $0.795 \pm 0.009$ | $0.822 \pm 0.007$ | $\mathbf{0.829 \pm 0.006}$ |
| | 2 | $0.565 \pm 0.015$ | $0.684 \pm 0.019$ | $0.745 \pm 0.006$ | $0.768 \pm 0.007$ | $\mathbf{0.775 \pm 0.008}$ |
| | 3 | $0.570 \pm 0.015$ | $0.668 \pm 0.018$ | $0.718 \pm 0.007$ | $0.737 \pm 0.005$ | $\mathbf{0.743 \pm 0.005}$ |
| | 4 | $0.570 \pm 0.020$ | $0.653 \pm 0.015$ | $0.698 \pm 0.008$ | $0.715 \pm 0.008$ | $\mathbf{0.720 \pm 0.006}$ |

The proportion of data $N$ used is randomly restricted, showing performance under various levels of data scarcity. PRC and ROC evaluations are reported separately for targets representing infections (I) and comorbidities (C).

Table 34: Summary $N$-Sensitivities for TEA on linear model with UKCF (Bold indicates best)

| | $N \times 1\%$ | $N \times 5\%$ | $N \times 20\%$ | $N \times 50\%$ | $N \times 100\%$ |
|---|---|---|---|---|---|
| PRC(I) | $0.207 \pm 0.017$ | $0.327 \pm 0.034$ | $0.413 \pm 0.031$ | $0.439 \pm 0.035$ | $\mathbf{0.450 \pm 0.035}$ |
| PRC(C) | $0.298 \pm 0.020$ | $0.439 \pm 0.047$ | $0.530 \pm 0.053$ | $0.551 \pm 0.058$ | $\mathbf{0.559 \pm 0.060}$ |
| ROC(I) | $0.567 \pm 0.026$ | $0.674 \pm 0.029$ | $0.738 \pm 0.027$ | $0.757 \pm 0.027$ | $\mathbf{0.767 \pm 0.026}$ |
| ROC(C) | $0.572 \pm 0.018$ | $0.680 \pm 0.032$ | $0.739 \pm 0.038$ | $0.760 \pm 0.041$ | $\mathbf{0.767 \pm 0.042}$ |

The proportion of data $N$ used is randomly restricted, showing performance under various levels of data scarcity. PRC and ROC evaluations are reported separately for targets representing infections (I) and comorbidities (C). Results are grouped over the temporal axis; note that the variance between splits is an artifact of this grouping.

### E.4  Contrived Negative Example

Here with give a contrived example where the goals of prediction and reconstruction are directly at odds with each other. Specifically, we show a setup where the "*and*" part of "compact *and* predictable" representations is impossible—that is, a compact target-representation that is *more* reconstructive ends up being *less* predictable. Consider the following (true) data generating process: Latent variables $\mathbf{p}, \mathbf{u}$ each consist of 10 independent dimensions, where $p_i \sim \mathcal{N}(0,1)$ and $u_i \sim \mathcal{N}(0,1)$. Feature vectors $\mathbf{x}$ are also of length 10, and are linear in $\mathbf{p}$. Target vectors $\mathbf{y} = [\mathbf{y}_P^\top, \mathbf{y}_U^\top]^\top$ are of length 50, where $\mathbf{y}_P$ is of length 10 and linear in $\mathbf{p}$ and $\mathbf{y}_U$ is of length 40 and linear in $\mathbf{u}$. See Figure 7. Note that the input features are inadequate for predicting all targets; our choice of lettering denotes elements that are in principle predictable ("P"), and those that are in principle unpredictable ("U").

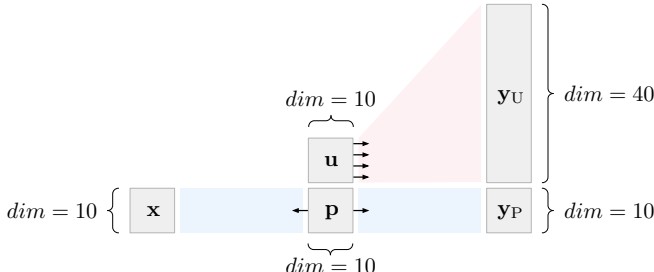

Figure 6: Data generating process for synthetic example. Latent vectors $\mathbf{p}, \mathbf{u}$ linearly generate feature vectors $\mathbf{x}$ and target vectors $\mathbf{y}$. In this situation, $\mathbf{y}_P$ is in principle predictable from $\mathbf{x}$, while $\mathbf{y}_U$ is impossible to predict.

Suppose this data generating process is unknown to us. First, consider what happens with direct prediction: A linear model would learn to predict $\mathbf{y}_P$ well, while predictions of $\mathbf{y}_U$ would be no better than random. So far, so good. Now given the feature and target dimensions, consider the (not unreasonable) choice of TEAs with a latent dimension of 10. This is an obvious problem: During reconstruction, we naturally get more bang for our buck by encoding more of the (highly compressible) $\mathbf{y}_U$ instead of $\mathbf{y}_P$; yet $\mathbf{y}_U$ is entirely *useless* to encode, as it is not predictable from inputs anyway. Reconstructing well is therefore *directly* at odds with predicting well. This is certainly an extremely contrived scenario; nevertheless, *without* sufficient domain knowledge, it serves as a caveat that—as with feature-embedding paradigms—target-embedding is only as good as its assumptions.

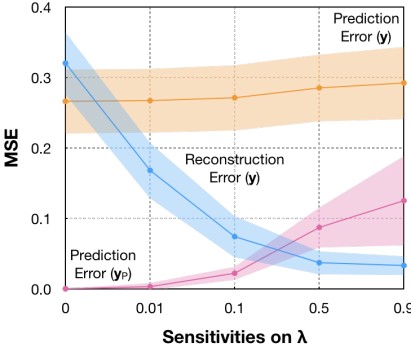

Figure 7: Synthetic scenario where prediction is directly at odds with reconstruction. The prior (that we can leverage compact and predictable representations of targets) is hugely incorrect in this case; as a result, not only is target-autoencoding not beneficial—it is positively harmful. For TEAs, we observe that as the strength-of-prior coefficient $\lambda$ increases, the overall prediction error actually increases (while the reconstruction error decreases).

Table 35: Results for TEA and comparators on linear model for negative example (Bold indicates best)

|  | Base | REG | FEA | TEA | F/TEA |
|---|---|---|---|---|---|
| MSE | **0.266 ± 0.045** | **0.266 ± 0.045** | **0.266 ± 0.045** | 0.285 ± 0.047 | 0.280 ± 0.045 |
| MSE(U) | **0.333 ± 0.056** | **0.333 ± 0.056** | **0.333 ± 0.056** | 0.334 ± 0.056 | 0.334 ± 0.056 |
| MSE(P) | **0.000 ± 0.000** | **0.000 ± 0.000** | **0.000 ± 0.000** | 0.087 ± 0.028 | 0.061 ± 0.022 |

MSE metrics are further reported separately for targets that are in principle predictable (P) and unpredictable (U).

Table 36: $\lambda$-Sensitivities for TEA on linear model for negative example (Bold indicates best)

| | $\lambda = 0$ | $\lambda = 0.01$ | $\lambda = 0.1$ | $\lambda = 0.5$ | $\lambda = 0.9$ |
|---|---|---|---|---|---|
| MSE | **0.266 ± 0.045** | 0.267 ± 0.045 | 0.271 ± 0.046 | 0.285 ± 0.047 | 0.292 ± 0.051 |
| MSE(U) | **0.333 ± 0.056** | **0.333 ± 0.056** | 0.334 ± 0.056 | 0.334 ± 0.056 | 0.334 ± 0.056 |
| MSE(P) | **0.000 ± 0.000** | 0.003 ± 0.005 | 0.022 ± 0.009 | 0.087 ± 0.028 | 0.125 ± 0.063 |

MSE metrics are further reported separately for targets that are in principle predictable (P) and unpredictable (U).

## E.5 Results from Open Discussion

One can also ask the (purely empirical) question of how much each model degrades on *out-of-distribution* data—without additional training to fine-tune the model to the new data. In this context, we actually have no reason to expect TEAs to degrade any more or less than comparators. For thoroughness, we show an additional experiment as an example for sensitivity analysis (using UKCF) as follows: Each model is trained (only) on male patients and tested (only) on female patients, and vice versa. The average results on held-out samples from in-distribution data and out-of-distribution data then allow us to compute the net *degradation* (i.e. negative difference), which is reported below. While TEAs individually perform better overall on both in-distribution and out-of-distribution samples, none of the *differences* in the amounts of degradation between models are statistically significant:

Table 37: Performance degradation for TEA and comparators on linear model with UKCF (Bold indicates best)

| | Base | REG | FEA | TEA | F/TEA |
|---|---|---|---|---|---|
| ROC(I) | 0.019 ± 0.015 | 0.020 ± 0.014 | 0.019 ± 0.015 | 0.020 ± 0.016 | **0.017 ± 0.014** |
| ROC(C) | 0.025 ± 0.015 | 0.029 ± 0.015 | 0.024 ± 0.014 | **0.013 ± 0.020** | 0.019 ± 0.018 |
| PRC(I) | 0.022 ± 0.020 | **0.018 ± 0.021** | 0.021 ± 0.022 | 0.033 ± 0.022 | 0.027 ± 0.022 |
| PRC(C) | 0.026 ± 0.021 | 0.029 ± 0.018 | 0.026 ± 0.019 | **0.018 ± 0.023** | 0.021 ± 0.019 |

PRC and ROC evaluations are reported separately for targets representing infections (I) and comorbidities (C). The two-sample $t$-test for a difference in means is conducted on the results. An asterisk next to the comparator result is used to indicate a statistically significant difference in means ($p$-value < 0.05) relative to the TEA result. Results are grouped over the temporal axis; note that the variance between splits is an artifact of this grouping.

Finally, given the staged training in Algorithm 1, it should be clear that the *order* cannot be changed. Stage 2 requires the encoder to *already* be trained to provide the requisite embeddings, so it must be preceded by Stage 1. Therefore the only relevant possibilities are: (1) Stages 1-2 by themselves, without Stage 3; this is simply the "No Joint" setting. (2) Stage 3 by itself, without Stages 1-2; this is simply the "No Staged" setting. (3) None of the stages altogether; this is simply the "Neither" setting. (4) Stages 1, 2, and 3 in order; this is simply Algorithm 1 itself. The only remaining possibility is to have Stage 3 precede Stages 1-2. This makes little sense, since when the reconstruction loss is trained by itself it is likely to "undo" the result of joint training. For thoroughness, we run an additional sensitivity experiment (using UKCF) to confirm this. The following corresponds to the left half of Table 4, with the additional column on the right (and the other columns labeled to reflect the training stages). Verifying our intuitions, the setting "3-1-2" behaves almost identically to the setting "1-2":

Table 38: Performance by training stages for TEA on linear model with UKCF (Bold indicates best); column headers indicate the sequence of training stages executed (note that "1-2-3" simply corresponds to Algorithm 1)

| | None | 1-2 | 3 | 1-2-3 | 3-1-2 |
|---|---|---|---|---|---|
| ROC(I) | 0.710 ± 0.072 | 0.747 ± 0.022 | 0.764 ± 0.022 | **0.767 ± 0.026** | 0.749 ± 0.022 |
| ROC(C) | 0.700 ± 0.075 | 0.744 ± 0.038 | 0.766 ± 0.038 | **0.767 ± 0.042** | 0.747 ± 0.037 |
| PRC(I) | 0.347 ± 0.085 | 0.402 ± 0.026 | 0.431 ± 0.031 | **0.450 ± 0.035** | 0.404 ± 0.027 |
| PRC(C) | 0.433 ± 0.083 | 0.507 ± 0.040 | 0.543 ± 0.054 | **0.559 ± 0.060** | 0.512 ± 0.042 |

PRC and ROC evaluations are reported separately for targets representing infections (I) and comorbidities (C). The two-sample $t$-test for a difference in means is conducted on the results. An asterisk next to the comparator result is used to indicate a statistically significant difference in means ($p$-value < 0.05) relative to the TEA result. Results are grouped over the temporal axis; note that the variance between splits is an artifact of this grouping.

