# OpenReview forum: "Target-Embedding Autoencoders for Supervised Representation Learning"
_ICLR.cc/2020/Conference — Accept (Talk)_

### Official Review · AnonReviewer1 · 2019-10-23
**Official Blind Review #1**

**Rating:** 8

**Review:**

This work introduces the idea of target embedding autoencoders for supervised prediction, designed to learn intermediate latent representations jointly optimized to be both predictable from features and predictive of targets. This is meant to help with generalization and has certain theoretical guarantees.

It is an interesting problem setting to consider where  Y is high dimensional instead of X. More examples of this would be useful to provide in the intro. I think this is crucial to understand where this method might be useful.

Figure 1 is super informative and very nice!

Section 2:
Why do we expect that this paradigm of autoencoder based regularization “generalizes” better?
I like the explicit and honest discussion of prior work in this section.
One question is how important is the choice of reconstruction loss function - L2, vs max likelihood gaussian, vs L1, vs cross entropy, etc for performance?
Another question: how bad is performance if the learning is done stagewise - first the Y-Z-Y^ representation is learned and then the X->Z predictor is learned.
If something is out of distribution, how easy are TEA based learners to finetune?
Overall the idea seems reasonable - if the targets have some common set of factors, just predict those instead of predicting the full target value which might be harder to get right. It’s just a question of whether this holds true in many domains and how well this reconstruction loss generalizes across problems?

Section 3:
“We havenoted that TEA components can in principle be instantiated by any architecture. Does its benefit extend beyond the commonly-studied domain of static classification?” -> not clear what this means? Does this mean this algorithm has been proposed before or is it that it can ALSO work on non static classification tasks? Not clear how to situate this claim

The theoretical section seems to follow largely from Le et al, but with important distinctions on dimensionalities of various spaces involved. I wonder if the authors can comment on how often Assumption 1 and 2 are actually satisified?

Related Work:
Is the main difference between Yu 2014 and this just in the norm based regularization? I don’t think so, can this be made more clear. This seems also fairly related to Yeh, is it just a generalization of that paradigm? Or is there more to it? In light of the contribution of Yeh, this seems like slightly more marginal of a contribution? Is the main points of contribution the theoretical analysis and the extended experiments to sequence data rather than static classification?

The results do seem to show a signficant benefit as compared to FEA or base models. It also seems like this is applicable across multiple disease datasets. Do the authors think that this could be applicable to other domains altogether? Would it be quick to run a comparison on these?

Generally seems like a well grounded and meaningful contribution with many improvements. Would be curious to see applications to other datasets and also some improvements/clarifications noted above?



**Experience Assessment:**

I have read many papers in this area.

**Review Assessment: Checking Correctness Of Derivations And Theory:**

I assessed the sensibility of the derivations and theory.

**Review Assessment: Checking Correctness Of Experiments:**

I assessed the sensibility of the experiments.

**Review Assessment: Thoroughness In Paper Reading:**

I read the paper at least twice and used my best judgement in assessing the paper.

---

> ### Author Response · Authors · 2019-11-11
> **Response for Reviewer 1 [Part 1/4]**
>
>
>
> Thank you for your insightful comments and questions. We give answers to each in turn.
>
> We believe the specific positioning and contribution of the paper may not have been the most clear. Therefore we start by emphasizing the focus of our work, in light of some of the questions. (1) First, we motivate and formalize TEA as a *general* framework, which "provide[s] a unifying perspective on recent applications of autoencoders to label-embedding" in disparate domains (p. 1). (2) This sets the stage for our theoretical contribution, which is to provide a *guarantee of generalization* for linear TEAs by demonstrating uniform stability. This allows us to distill its benefit in the simplest setting, removing any confounding factors from domain-specific architectures. (3) Our empirical novelty (in addition to verifying our claim for the linear case) is to extend validation of this approach to the *temporal* domain---for multi-variate sequence forecasting with recurrent architectures. While we make the point that certain prior works can be interpreted as specific instantiations of TEAs in the *static* setting, we are the first to do so in the recurrent, sequential setting and for both regression and classification---underscoring the further generality of this approach beyond feedforward instantiations.
>
> *** Section 1 ***
>
> (1.1) More examples in the introduction: We agree that more immediate examples of high-dimensional output would be useful. We will explicitly include (in para. 2) examples from the related work that we cite, including 3D voxel prediction [Girdhar, ECCV 2016], image segmentation [Oktay, T-MI 2018], as well as any kind of object annotation such as images, text, and music [Table 7]. (Note that for the latter, the vast majority of work takes data with bag-of-features vectors as input, although in principle this need not be the case, e.g. with more complicated underlying models). This is in addition to the (already mentioned) temporal setting of multi-variate sequence trajectory forecasting that we focus on as our empirical contribution.
>
> *** Section 2 ***
>
> (2.1) Why do we expect autoencoding to help generalization: In the case of FEAs, the intuition is that what is reconstructive of features is likely to encode what is discriminative (downstream). Specifically, [Le, NIPS 2018] quantifies the generalization benefit for the linear case, capturing the intuition that reconstruction is "more likely to prefer a more robust model amongst a set of similarly effective models". In the case of TEAs, the intuition is similar but entirely opposite---that what is reconstructive of targets is likely to encode what is predictable (upstream). Here, the inverted setting warrants moving the cross-representability assumption into latent space, and we analogously quantify the generalization benefit (Section 3).
>
> (2.2) Importance of choice of reconstruction loss: As noted in Theorem 1 (Section 3), the bound is obtained if the reconstruction loss is strongly convex. The (most commonly used) quadratic loss function is 2-strongly convex. The logistic loss function and hinge loss function are convex, but not strongly convex. However, as noted in [Liu, TPAMI 2016], in statistical learning theory we often assume that $h(x)\in[-U,U]$ for hypothesis $h$ and some positive constant $U$, in which case the loss functions may be strongly convex (e.g. the logistic loss function is then $exp(-U)/4$-strongly convex). In our experiments, the datasets are chosen to obtain a variety of binary, continuous, and mixed-target settings, and we empirically observe the benefit of TEAs in all cases. We agree that future research may empirically explore the effect of a wide variety of reconstruction losses---both convex and otherwise.
>
> (2.3) How bad is performance if learning is done stagewise: This is a very relevant question, considering the importance of joint training to obtaining uniform stability. We actually examined this as part of our source-of-gains analysis (Section 5). For every setting of our experiments, we performed the "No Joint" setting (i.e. only first two stages of Algorithm 1 are performed like you mentioned, skipping the joint training stage), as well as the "No Staged" setting (i.e. only the final joint training stage is performed, skipping the (pre)-training stages). In all cases, both linear and nonlinear (Tables 4-5, and more detail in Appendix E), we observe that both sources of benefit are important for performance: neither setting performs quite as well as when both are combined (p. 8). In other words, training stagewise is still better than having no target-autoencoding at all, but not as good as when combined with joint-training. (Finally, as expected the "Neither" setting---equivalent to vanilla prediction---performs the worst).

---

> ### Author Response · Authors · 2019-11-11
> **Response for Reviewer 1 [Part 2/4]**
>
>
>
> (2.4) Out of distribution: We assume that the question refers to how we might adapt a learned TEA model to data that the original model was not trained on. Although the focus of our analysis is on "generaliz[ing] well to new samples from the same distribution" (p. 2), we agree that this is an interesting question. First, adapting to new data (instead of retraining a model from scratch using the new data) can be done simply by treating the existing model as a pretrained set of parameters: Algorithm 1 can then proceed (on the new data) with parameters initialized with their existing values. Compared with a direct-prediction model, the only difference is (again) the staged and joint training procedure as in Algorithm 1. Of course, specialized techniques (e.g. Bayesian optimization) would have a lot more to say depending on what specifically is the expected shift in the data distribution.
>
> Second, we can also ask the (purely empirical) question of how much each model degrades on out-of-distribution data---without additional training to fine-tune the model to the new data. In this context, we have no reason to expect TEAs to degrade any more or less than the comparators we examine. To quickly test an example, we performed this additional sensitivity (using UKCF) as follows: Each model is trained (only) on male patients and tested (only) on female patients, and vice versa. The average results on held-out samples from in-distribution data and out-of-distribution data allows us to compute the net degradation (i.e. negative difference), which is reported below. While TEAs individually perform better overall on both in-distribution and out-of-distribution samples, none of the *differences* in the specific amount of degradation are statistically significant.
>
> Table A: Summary in-distribution vs. out-of-distribution performance for TEA and comparators with UKCF. PRC and ROC metrics are reported separately for variables representing infections (I) and comorbidities (C).
> ------------------------------------------------------------------------------------------------------------
>         Base                     REG                       FEA                      TEA                  F/TEA
> ------------------------------------------------------------------------------------------------------------
> ROC(I)
> 0.019+/-0.015     0.020+/-0.014      0.019+/-0.015    0.020+/-0.016    0.017+/-0.014
> ------------------------------------------------------------------------------------------------------------
> ROC(C)
> 0.025+/-0.015     0.029+/-0.015      0.024+/-0.014     0.013+/-0.020    0.019+/-0.018
> ------------------------------------------------------------------------------------------------------------
> PRC(I)
> 0.022+/-0.020     0.018+/-0.021      0.021+/-0.022     0.033+/-0.022    0.027+/-0.022
> ------------------------------------------------------------------------------------------------------------
> PRC(C)
> 0.026+/-0.021     0.029+/-0.018     0.026+/-0.019      0.018+/-0.023    0.021+/-0.019
> ------------------------------------------------------------------------------------------------------------
> * [Nov 13 Update]: These are preliminary numbers using subsets of the data for quick results as an example. We are re-running a larger-scale version in the meantime, such that the splits and samples are comparable with the rest of the existing experiments, and will post an update.
>
> (2.5) How well this works across domains: Please kindly refer to answer (3.1). There is already existing evidence (which we cite extensively) that target-embedding works well across (static) application domains where the target space is high-dimensional, such as image segmentation, voxel prediction, and object annotation; see also answer (1.1). In fact, it is precisely the empirical efficacy of these applications that motivates our theoretical investigation. (Of course, what has *not* been explored so far is the efficacy in the temporal setting, which is then part of our empirical contribution).

---

> ### Author Response · Authors · 2019-11-11
> **Response for Reviewer 1 [Part 3/4]**
>
>
>
> *** Section 3 ***
>
> (3.1) How to situate the claim: As noted throughout the paper (p. 5, 6, and 8), the application work of both [Girdhar, ECCV 2016] and [Yeh, AAAI 2017] can be interpreted as specific instantiations of TEAs in the *static* setting---the latter in the (1) multi-label classification setting (with sophisticated refinements), and the former specifically for (2) voxel prediction with convolutional architectures (under the "indirect" variant). (3) Moreover, the various works on label-space reduction [Table 7] can loosely be considered under the umbrella of target-embedding. In that sense, while we unify the essential common thread between these disparate applications under the concept of TEA (Sections 1-2), there is already empirical evidence of the benefit of target-embedding in static classification. Our point here is, what has *not* been explored at all is the utility of target-embedding in the *temporal* setting---for multivariate sequence data, especially via recurrent architectures and for both regression and classification. We are the first to do this, and we find that TEAs generously extend to this setting, thereby highlighting the further generality of the approach beyond feedforward instantiations; this is our empirical contribution.
>
> We admit that the phrasing can be clearer. The sentence in question appears in Section 3, where the primary focus is on the theoretical contribution. We completely agree that this "preview" of Section 5 may be confusing to appear so early on---esp. before Section 4, and esp. since the motivation is anyway explained in much greater detail in the beginning of Section 5. Depending on space, we will either remove this from the introduction to Section 3, or (at least) include references to [Girdhar, ECCV 2016] and [Yeh, AAAI 2017].
>
> (3.2) Reasonableness of assumptions: Indeed, Section 3 takes off from the line of work from [Bousquet, JMLR 2002], [Liu, TPAMI 2016], and [Le, NIPS 2018], which use similar tools and overall strategy. While [Le, NIPS 2018] departs from the generic multi-task analysis of [Liu, TPAMI 2016] via their Assumption 6 (identically, our Assumption 2), we in turn invert our setting from [Le, NIPS 2018] via Assumption 1. Briefly, this assumption will hold with with $\varepsilon=0$ as long as the number of independent latent vectors is at least $|\mathcal{Z}|$. This is virtually guaranteed for any compressive autoencoder, since an encoding arm that maps into the latent space from a higher-dimensional target space, we expect if some subset $\{\mathbf{b}_{1},...,\mathbf{b}_{M}\}\subset\{\mathbf{y}_{1},...,\mathbf{y}_{N}\}$ spans $\mathcal{Y}$ that $\{\mathbf{W}_{e}\mathbf{b}_{1},...,\mathbf{W}_{e}\mathbf{b}_{M}\}$ then also span $\mathcal{Z}$ in order to be maximally reconstructive. Note the central importance of the fact that the target space is *higher-dimensional* in our setting for enabling this assumption; picking vectors from feature space instead would be unreasonable (see Remark 2; see also Remark 1 for the mildness of this assumption). In addition, note that while (for simplicity) we take $\varepsilon=0$ in Appendix A, this need not even be the case (see Remark 5). Assumption 2 is identical to Assumption 6 in [Le, NIPS 2018], and we refer to their original exposition (p. 5) for details. However for further perspective, consider the simpler (but unreasonable) assumption by way of contrast: that *individually* $L^{B}_{r}(\mathbf{\Theta_{*}})-L^{B}_{r}(\kappa\mathbf{\Theta}^{\prime}_{*}+(1-\kappa)\mathbf{\Theta_{*}})
> \leq
> a\left[L^{\prime}_{r}(\mathbf{\Theta_{*}})-L^{\prime}_{r}(\kappa\mathbf{\Theta}^{\prime}_{*}+(1-\kappa)\mathbf{\Theta_{*}})\right]$ and $L^{B}_{r}(\mathbf{\Theta^{\prime}_{*}})-L^{B}_{r}(\kappa\mathbf{\Theta}_{*}+(1-\kappa)\mathbf{\Theta^{\prime}_{*}})
> \leq
> a\left[L^{\prime}_{r}(\mathbf{\Theta^{\prime}_{*}})-L^{\prime}_{r}(\kappa\mathbf{\Theta}_{*}+(1-\kappa)\mathbf{\Theta^{\prime}_{*}})\right]$.
> Although this would serve the same purpose, it is much stronger and eminently unreasonable: Consider that $L_{r}^{\prime}$ is higher at $\mathbf{\Theta}_{*}$ than $\mathbf{\Theta}^{\prime}_{*}$ but $L^{B}_{r}$ is the opposite. In the case of Assumption 2 (identically, Assumption 6 in [Le, NIPS 2018]), neither do we require that the reconstruction losses be similar, nor that their differences be similar individually---but only that the combined increase or decrease between the two points be similar. Note that both sides of the inequality are non-negative due to the loss functions being convex. Again, we refer to their original exposition and supplementary proof for context [Le, NIPS 2018].

---

> ### Author Response · Authors · 2019-11-11
> **Response for Reviewer 1 [Part 4/4]**
>
>
>
> *** Section 4 ***
>
> (4.1) Compared to [Yeh, AAAI 2017]: The entire line of work on label space reduction for multi-label classification (Table 7, which includes this) actually has a different focus than ours. Their techniques worry about *label reduction*, and about specific loss functions that aim to preserve dependencies within and among spaces; their focus is specifically on object annotation and tagging, and their starting point is *binary relevance*. Now, one such approach has successfully employed autoencoders [Yeh, AAAI 2017], which we can interpret as a specific instantiation of TEAs (although with a number of sophisticated additions to solve their problem). In contrast, we operate on a higher level of abstraction, and we focus specifically on autoencoding *in general*---and the regularizing effect of the reconstruction loss on learning the prediction model; our starting point is *direct prediction*, and the output can be of any form (classification or regression). Our contributions are therefore very different. The focus of [Yeh, AAAI 2017] is to solve a specific *applied* problem and use all the tools to do so (e.g. combining with canonical correlation analysis); the same can be said of [Girdhar, ECCV 2016], which can also be seen as an instantiation of TEAs with additional design choices (e.g. fine-tuning a pretrained AlexNet). Instead, we abstract and analyze the interesting common thread between these (seemingly disparate) static application domains: the fact that target-embedding *by itself* is very useful. You are correct; we do not attempt to rehash the static classification application setting, which existing work already does. Our main novelties are the theoretical analysis (which we also verify with experiments using the linear setting), and the empirical extension to recurrent, sequential setting and for both regression and classification (which we are also the first to do so).
>
> (4.2) Compared to [Yu, ICML 2014]: Now, [Yu, ICML 2014] is actually less directly comparable than [Yeh, AAAI 2017]: They cast label-embedding within the generic empirical risk minimization framework---as learning a linear model with a *low-rank constraint*, and specifically focus on missing labels and supporting various losses. However, we mention this work mainly for two reasons: First, this perspective captures the general intuition of a restricted number of latent factors, which is the prior that makes TEAs work, and is (loosely) analogous to the idea of using an explicit low-dimensional latent space for an intermediate mapping. Second, their analysis admits generalization bounds that are based on norm-based regularization, whereas the significance of our analysis is that we achieve our result without needing it (see Remark 3).
>
> *** Section 5 ***
>
> (5.1) Different domains: Please kindly refer to answers (2.5) and (3.1), where we mention (as we do in the paper) existing applications to static settings. So, we already have evidence that the idea works for voxel prediction, image segmentation, and object annotation and tagging. For our extension to the temporal setting, the domain of disease trajectories was carefully selected as a particularly appropriate testbed, due to the fact that medical knowledge in this domain gives us confidence that the requisite prior for TEAs is satisfied---i.e. that variations in targets are driven by a lower-dimensional set of underlying factors (p. 1, 3, and 5); see scientific papers cited (p. 6). An interesting potential future direction may be its utility for video frame prediction, although existing methods typically require much more specialized architectures per settings and datasets, and would be beyond the scope of this paper. Finally, note that we do *not* in fact expect TEAs to work just about anywhere: We highlight the central importance of the correctness of the prior in Appendix E.4, where we provide a negative example.
>
> * All citations can be found in the original bibliography.

---

### Official Review · AnonReviewer2 · 2019-10-27
**Official Blind Review #2**

**Rating:** 6

**Review:**

1. Summary: In this paper, the authors proposed a Target-Embedding Autoendocer (TEA) model for supervised representation learning. Different from the traditional feature embedding autoencoder model, TEA tries to learn a compact latent representation that can reconstruct the target vector. Hypothetically, this model should be especially useful when the target vector has a much higher dimension than the feature vector. The authors analyzed the proved some characteristics of this framework and conducted empirical experiments on three datasets to prove its effectiveness.
2. Overall assessment: The motivation of this paper is well justified. It's easy to follow and fun to read, even for a person who is not an expert in this area, like me. However, there still exist some problems in this paper. It needs more improvement to get published in a competitive conference like ICLR.
3. Comments:
3.1 Datasets used in this paper cannot fully prove the effectiveness of this framework. These datasets are all from very similar domains. The dimension of target vectors is comparable to that of feature vectors. In my view, it's necessary to test on more different types of datasets to prove the usefulness of a model, especially if it is a general framework like TEA.
3.2 Models used in this paper are relatively simple. Demonstrate the performance of TEA on more advanced models and more difficult tasks can deliver more insights to the community.
3.3 No state-of-the-art models are used in experiments. It's very likely that some existing work has already adopted the idea of target embedding. There also exist much other work on dealing with high dimensional target vector problem. How are the performances of these models? What is the advantage of the proposed framework over these existing work?
3.4 The source of gain part on page 8 should contain more explanations and analysis. This part is one of the most important parts of this paper. It can provide quite valuable insights to readers. I hope the author can expand it.
3.5 More details about training and inference are needed. The authors only use a few sentences to describe their three staged training process. I still have some questions left after reading it, such as how do you train the shared parts in TEA? Do you update its parameters in all stages? What the effect of the order of training? What will happen if I change it?

**Experience Assessment:**

I do not know much about this area.

**Review Assessment: Checking Correctness Of Derivations And Theory:**

I did not assess the derivations or theory.

**Review Assessment: Checking Correctness Of Experiments:**

I assessed the sensibility of the experiments.

**Review Assessment: Thoroughness In Paper Reading:**

I made a quick assessment of this paper.

---

> ### Author Response · Authors · 2019-11-11
> **Response for Reviewer 2 [Part 1/5]**
>
>
>
> Thank you for your thoughtful comments. We give answers to each in turn.
>
> We believe the specific positioning and contribution of the paper may not have been the most clear. Therefore we start by emphasizing our focus, in light of your comments. (1) First, we motivate and formalize TEA as a *general* framework, which "provide[s] a unifying perspective on recent applications of autoencoders to label-embedding" in disparate domains (p. 1). (2) This sets the stage for our theoretical contribution, which is to provide a *guarantee of generalization* for linear TEAs by demonstrating uniform stability. This allows us to distill its benefit in the simplest setting, removing any confounding factors from domain-specific architectures. (3) Our empirical novelty (in addition to verifying our claim for the linear case) is to extend validation of this approach to the *temporal* domain---for multi-variate sequence forecasting with recurrent architectures. While we make the point that certain prior works can be interpreted as specific instantiations of TEAs in the *static* setting, we are the first to do so in the recurrent, sequential setting and for both regression and classification---underscoring the further generality of this approach beyond feedforward instantiations.
>
> (3.1) "Datasets [...] cannot prove the effectiveness of this framework":
>
> We wish to kindly point out that it is actually *not* our objective to prove the effectiveness of this framework *from scratch*. The fact that this general idea works well in a number of (static) settings is already known (and we cite and mention them throughout the paper). In fact, the empirical efficacy of existing domain applications is precisely what motivates our main theoretical contribution: To "provide a unified perspective on recent applications" (p. 1) of this idea, which allows us to first focus on examining *why* it works (our theoretical contribution). As noted throughout the paper (p. 5, 6, and 8), the application work of e.g. [Girdhar, ECCV 2016] and [Yeh, AAAI 2017] can be interpreted as specific instantiations of TEAs in the *static* setting---the latter in the (1) multi-label classification setting (with additional refinements), and the former specifically for (2) voxel prediction with convolutional architectures (under the "indirect" variant). (3) Moreover, the various works on label-space reduction [Table 7] can loosely be considered under the umbrella of target-space embedding, as is (4) the work of [Oktay, T-MI 2018] for image segmentation. In that sense, while we unify the essential common thread between these disparate applications under the concept of TEA (Sections 1-2), there is already empirical evidence of the benefit of target-embedding in the commonly considered static setting.
>
> Now, what has *not* been empirically explored at all is the utility of target-embedding in the *temporal* setting---for multivariate sequence data, especially via recurrent architectures and for both regression and classification. We are the first to do this, and we find that TEAs generously extend to this setting. This is our empirical contribution (in addition to verifying our claims for the linear case, plus extensive sensitivities). Furthermore, the domain of disease trajectories was specifically and carefully selected as a particularly appropriate testbed, due to the fact that medical knowledge in this domain gives us confidence that the requisite prior for TEAs is satisfied---i.e. that variations in targets are driven by a lower-dimensional set of underlying factors (p. 1, 3, and 5); see scientific papers cited (p. 6). These points are all explained (with more detail) in the beginning of Section 5 (pp. 6-7), as well as a negative example to highlight the importance of the prior (Appendix E.4).

---

> ### Author Response · Authors · 2019-11-11
> **Response for Reviewer 2 [Part 2/5]**
>
>
>
> (3.2) "Models used are relatively simple":
>
> Since (3.3.a) poses a very similar question for *experiments* and similarly asks for more advanced models), we assume here that your comment is referring to our *theoretical* result.
>
> We would like to point out that a theoretical analysis of the linear setting is *not* of limited use---especially in a setting with no precedent. Our objective is to distill the essence of the target-embedding idea, such that we can isolate its theoretical benefit. To do so, a rigorous analysis deliberately and necessarily begins with a simplest incarnation of TEAs: the linear case. This is intentional and commonplace: As a matter of fact, (1) the seminal work on the generalization benefit of multi-task learning using Rademacher complexity operates in the linear setting [Maurer, JMLR 2006], and (2) the most recent landmark analysis by uniform stability is also performed in the linear setting [Liu, TPAMI 2016]. (3) For supervised feature-embedding as an auxiliary task, the first such analysis of generalization from stability is done in the linear setting [Le, NIPS 2018]. Similarly, (4) for multi-label classification, the generalization properties of label-embedding with norm-based regularization is first analyzed using the ERM framework---in the linear setting [Yu, ICML 2014].
>
> The significance our analysis is that it allows us to *unambiguously* interpret its benefit as a regularizer. Now, it is often easy to argue on an intuitive level for the "regularizing" effect of some such additional loss term. For example, [Mostajabi, CVPR 2018] also refers to the "regularizing" effect of their label autoencoder (again, not trained jointly), but this expression is used loosely and intuitively, without (1) *identifying* or (2) *quantifying* the precise mathematical mechanism. In stark contrast, in our analysis the complete loss can be summarized and rewritten as $L(\mathbf{\Theta})=L_{p}(\mathbf{\Theta})+R_{1}(\mathbf{\Theta})+R_{2}(\mathbf{\Theta})$---that is, a combination of the primary prediction loss plus additional regularization, where $R_{1}(\mathbf{\Theta})=\frac{1}{N}\sum_{m=1}^{M}\ell_{r}(\mathbf{\Theta}\mathbf{W}_{e}\mathbf{b}_{m},\mathbf{b}_{m})$ and $R_{2}(\mathbf{\Theta}) = \frac{1}{N}\sum_{n=1}^{N}\ell_{r}(\mathbf{\Theta}\mathbf{W}_{e}\mathbf{y}_{n},\mathbf{y}_{n})-\frac{M}{N}L^{B}_{r}(\mathbf{\Theta})$. In particular, the proof of Theorem 1 depends critically on $R_{1}(\mathbf{\Theta})$ to achieve the upper-bound on instability. As a result, (1) this precisely *identifies* the regularizer in question, while (2) our uniform stability result *quantifies* the generalization benefit. This fact is already implicit in the analysis of Appendix A, but we can certainly mention it explicitly at the end of Section 3 for better emphasis.
>
> This gives important theoretical insight into the empirical gains from TEAs. Significantly, we establish the fact that a tight generalization bound is obtained with absolutely nothing but the simple addition of the *joint* reconstruction loss (i.e. no additional unlabeled data, no explicit norm-based regularization, etc). Now, of course domain-specific application papers (cited previously in this response) have employed similar ideas in contexts of varying complexity, with potentially sophisticated architectures. Their *empirical* objective is to achieve SOTA in their specific domain (e.g. 2D image to 3D voxel prediction, segmentation, etc). In contrast, our *theoretical* objective as a first analysis of the TEA framework itself is---importantly---to *remove* the confounding effects of such tailored models (e.g. pretrained models, specific nonlinearities, custom losses, etc.) in order to distill the crux of the benefit in terms of generalization. (Empirically, of course, our experiments do cover both linear and nonlinear cases; see next).

---

> ### Author Response · Authors · 2019-11-11
> **Response for Reviewer 2 [Part 3/5]**
>
>
>
> (3.3.a) "No state-of-the-art models are used in experiments":
>
> First, we would like to gently reiterate that our focus is on developing a *framework-level* analysis to deepen our understanding the theoretical underpinnings of TEAs (see points (1), (2), and (3) in the first paragraph of our response). In particular, we are *not* in search of SOTA for any specific application domain. Quite to the contrary, our objective is to distill the isolated benefit of TEAs using a *minimal* setting with as few confounding factors as possible. Therefore---unlike in the variety of more application-focused papers we cite---we intentionally refrain from bolting on any additional design choices (e.g. correlation analyses, fine-tuning pretrained models, custom loss functions, and whatever else could push SOTA for each dataset and each domain). After all, we want evidence of improvement with absolutely *nothing but* the simple addition of the joint reconstruction loss.
>
> Given our theoretical findings, our empirical contribution serves two purposes (detailed in Section 5). First, we verify our claims for the linear case (corresponding to our theoretical setting). Second, we extend TEAs to the recurrent, sequential multi-variate setting for both regression and classification. Since the temporal setting has never been explored with TEAs, a standard and popular architecture such as RNNs with GRUs is appropriate. Recall our focus; we don't want other factors getting in the way of this foray. We are focusing on studying the *framework* and its potential generalizability beyond feedforward instantiations, not on excessively optimizing against *specific* architectures for SOTA applications. In this sense, our work is positioned analogously to [Le, NIPS 2018] and [Liu, TPAMI 2016], who respectively investigate---on a *framework* level---the generalization benefit of multi-task learning and supervised feature-embedding.
>
> In fact, should we instead focus on SOTA models, we would be unable to perform the methodical sensitivity analyses in Section 5. After all, in addition to (1) verifying the linear case, and (2) extending to recurrent models, we also wish to (3) pick apart the sources of gains from both joint and staged training, as well as (4) comparing the alternate variations of the TEA framework present in existing work. Moreover, we (5) examine the incremental effect of additional norm-based regularization, (6) the sensitivity of TEAs to the strength of prior, as well as (7) the comparative sample complexity of prediction with and without target-embedding. These careful analyses would all become noisy and intractable should we introduce multiple confounding factors in the form of SOTA models and their various components and configurations.
>
> (3.3.b) "It's very likely that some existing work has already adopted the idea":
>
> We completely agree, and we *already* cite all the ones we are aware of. Please kindly also refer to answer (3.1). For instance, as noted throughout the paper (p. 5, 6, and 8), the application work of e.g. [Girdhar, ECCV 2016] and [Yeh, AAAI 2017] can be interpreted as specific instantiations of TEAs (in the static setting). Please see also the works in Table 7. We also give extensive discussions (and summary tables) of how they relate and compare (see comprehensive survey in addition Appendix B.2). Having these is good, and the more empirical results there are, the more they support the relevance of our theoretical contribution. What has *not* been done by existing work is to explore the generalizability of this idea to the *temporal* setting; again, kindly refer to answer (3.1).

---

> ### Author Response · Authors · 2019-11-11
> **Response for Reviewer 2 [Part 4/5]**
>
>
>
> (3.3.c) "What is the advantage of the proposed framework over these existing work":
>
> Pardon the repetition: we would like to gently reiterate that our work serves a very different purpose than existing work. We are *not* proposing a model from scratch, to which "other" works can be viewed as "competitors". To the contrary, the empirical efficacy of existing domain applications of target-embedding is *precisely* what motivates our main theoretical contribution: To "provide a unified perspective on recent applications" (p. 1) of this idea, which allows to examine *why* it works. Again, several existing works can be viewed as specific instantiations of TEAs; see answer (3.1). Our mission is to provide *framework-level* analytical insight into why target-embedding works---not to compete against anything within a specific application domain; that would be an entirely different pursuit. There is an important distinction between studying TEAs *per se*, versus specific *architectural* instantiations present in different application domains.
>
> We are the *first* work to rigorously analyze the generalization benefit of the former (our theoretical novelty). In addition to verifying the linear case with experiments, we are also the *first* to extend the latter to recurrent, multi-variate sequence forecasting and both regression and classification (our empirical novelty)---which highlights the further generality of this approach beyond feedforward instantiations. The focus is on isolating the benefit of TEAs, not on specific architectural novelties that may boost performance for various datasets and domains. That said, we do in fact experiment with a multitude of variations (i.e. TEA, the indirect TEA(L), the hybrid TEA(LP), the "No Joint" setting, as well as the "No Staged" setting). These reflect the *framework-level* variation in the general idea of "target-embedding" present in existing work. For instance, we note (p. 8) that the TEA(L) variant corresponds to the framework-level setup in [Girdhar, ECCV 2016] and [Yeh, AAAI 2017], which are jointly learned via the reconstruction loss and a *latent* loss instead---by regressing learned embeddings during the joint training stage (Figure 4(d), in Appendix D). Then, we duly show the performance of this TEA(L) setting in comparison with all the other settings in each experiment (see Tables 4-5, as well as Appendix E). Please kindly also refer to answer (3.3.a).
>
> (3.4) "The source of gain [...] should contain more explanations and analysis":
>
> Thank you for the suggestion. Yes, we agree that a more detailed explanation will improve clarity of exposition. This was originally kept concise due to space limitation, but we can expand the explanation with more detail as follows:
>
> ---
> There are two (related) interpretations of TEAs. First, we studied the *regularization* view in Section 3; this concerns the benefit of joint training using both prediction and reconstruction losses. Ceteris paribus, we expect performance to improve purely by dint of the jointly trained TEA objective. Second, the *reduction* view says that TEAs decompose the (difficult) prediction problem into two (smaller) tasks: the autoencoder learns a compact representation $\mathbf{z}$ of $\mathbf{y}$, and the predictor learns to map $\mathbf{x}$ to $\mathbf{z}$. This suggests a simpler possibility---that of separately training the autoencoder and predictor arms one after the other in two stages. Now, our presentation of TEAs (Section 2 and Algorithm 1) is a combination of both ideas: All three components are jointly trained in a third stage following the first two, similar to [Girdhar, ECCV 2016]. Our goal is now to account for the improvement in performance due to these two sources of benefit; Table 4 does so for the linear case (on UKCF), and Table 5 for the more general nonlinear case (on all datasets). The ``"No Joint" setting isolates the benefit from staged training only. This is analogous to basic unsupervised pretraining (though using targets), and corresponds to omitting the final joint training stage in Algorithm 1. The `"`No Staged" setting isolates the benefit from joint training only (without pretraining the autoencoder or predictor), and corresponds to omitting the first two training stages in Algorithm 1. The ``"Neither" setting is equivalent to vanilla prediction without leveraging either of the advantages from target-representation learning (REG). We observe that while both sources of benefit are individually important for performance, neither setting performs quite as well as when they are combined. See Appendix E.1-2 for extended results.
> ---
>
> We agree that the source of gains is important for understanding the joint and staged training aspects of TEAs. Bear in mind that, since our goal is to thoroughly investigate TEAs overall, we also need to allow adequate space to cover all of the 7 considerations listed in the final paragraph of answer (3.3.a).

---

> ### Author Response · Authors · 2019-11-11
> **Response for Reviewer 2 [Part 5/5]**
>
>
>
> (3.5.a) "More details about training and inference are needed":
>
> We agree that expanding this subsection with more details will enhance clarity of exposition. (Bear in mind that all of these points are detailed in Algorithm 1). We can update the middle portion of the "Training and Inference" section in the main manuscript as follows:
>
> ---
> Training occurs in three stages: In the first stage, the autoencoder is trained (to learn representations); in this stage, the parameters of the encoder and decoder are trained on the reconstruction loss. In the second stage, the prediction arm is trained to regress the learned embeddings (generated by the encoder); in this stage, only the parameters of the predictor are trained (on the latent loss), and the parameters of the encoder (and decoder) are frozen. Finally in the third stage, all three components are jointly trained on both the prediction loss and reconstruction loss; in this stage, the parameters of the encoder, predictor, and shared decoder model are all trained. Note that during training, the shared forward model receives two types of latents as input: encodings of true targets (to compute the reconstruction loss), as well as encodings predicted from features (to compute the prediction loss).
> ---
>
> (3.5.b) "What is the effect of the order of training? What will happen if I change it?":
>
> Given Algorithm 1 and the more detailed explanation in answer (3.5.a), it should now become clear that the order cannot be changed. Stage 2 requires the encoder to *already* be trained to provide the requisite embeddings, so it must be preceded by Stage 1. Therefore the only relevant possibilities are: (1) Stages 1-2 by themselves, without Stage 3; this is simply the "No Joint" setting that we conduct on all experiments. (2) Stage 3 by itself, without Stages 1-2; this is simply the "No Staged" setting that we conduct on all experiments. (3) None of the stages altogether; this is simply the "Neither" setting that we conduct on all experiments. (4) Stages 1, 2, and 3 in order; this is simply Algorithm 1 itself.
>
> Finally, the only remaining possibility is to have Stage 3 precede Stages 1-2. This makes little sense, since when the reconstruction loss is trained by itself it is likely to "undo" the result of joint training. However, for thoroughness, we have run an additional sensitivity experiment (using UKCF) to confirm this. The following corresponds to the left half of Table 4, with the additional column on the right (and the other columns labeled to reflect the training stages). Verifying our intuitions, the setting "3-1-2" behaves almost identically with the setting "1-2".
>
> Table A: Summary performance by training stages for TEA on linear model with UKCF. Column headers indicate the sequence of training stages executed. Note that "1-2-3" simply corresponds to Algorithm 1. PRC and ROC metrics are reported separately for variables representing infections (I) and comorbidities (C).
> ---------------------------------------------------------------------------------------------------------
>        None                     1-2                        3                       1-2-3                   3-1-2
> ---------------------------------------------------------------------------------------------------------
> PRC(I)
> 0.347+/-0.085    0.402+/-0.026    0.431+/-0.031    0.450+/-0.035    0.404+/-0.027
> ---------------------------------------------------------------------------------------------------------
> PRC(C)
> 0.433+/-0.083    0.507+/-0.040    0.543+/-0.054    0.559+/-0.060    0.512+/-0.042
> ---------------------------------------------------------------------------------------------------------
> ROC(I)
> 0.710+/-0.072    0.747+/-0.022    0.764+/-0.022    0.767+/-0.026    0.749+/-0.022
> ---------------------------------------------------------------------------------------------------------
> ROC(C)
> 0.700+/-0.075    0.744+/-0.038    0.766+/-0.038    0.767+/-0.042    0.747+/-0.037
> ---------------------------------------------------------------------------------------------------------
>
> * Aside from the two papers that Reviewer 4 referenced, all citations can be found in the original bibliography.

---

### Official Review · AnonReviewer3 · 2019-10-31
**Official Blind Review #3**

**Rating:** 8

**Review:**

This is an extremely well-written and well-motivated paper. The idea of target-embedding autoencoders is extremely relevant for problems where the dimension of the label space is as large (or larger) than the dimension of the input features. The experiments are thorough, the theoretical guarantees are extremely well thought of and derived. The applications to modelling the progression of cystic fibrosis and Alzheimer's are extremely useful and timely.  I vote for a strong accept for this paper.

I would like to see some references to the extreme multi-label classification problems (http://manikvarma.org/downloads/XC/XMLRepository.html) and some of the other probabilistic approaches attempted in this domain (please see https://papers.nips.cc/paper/5770-large-scale-bayesian-multi-label-learning-via-topic-based-label-embeddings and the references and citations).

**Experience Assessment:**

I have published one or two papers in this area.

**Review Assessment: Checking Correctness Of Derivations And Theory:**

I assessed the sensibility of the derivations and theory.

**Review Assessment: Checking Correctness Of Experiments:**

I assessed the sensibility of the experiments.

**Review Assessment: Thoroughness In Paper Reading:**

I read the paper at least twice and used my best judgement in assessing the paper.

---

> ### Author Response · Authors · 2019-11-11
> **Response to Reviewer 3**
>
>
>
> Thank you for your thoughtful comments and suggestions.
>
> We agree that the field of *extreme* multi-label classification [3] is relevant as well, especially in the context of our discussion for Table 7. We also agree that the probabilistic methods in [1] and [2] present alternative approaches with advantages in performance and use cases; they will provide more context in the related work discussion. Finally, we also find [4] worth referencing in light of the setting for our experiments. We thank you for pointing out these works: we will reference [1], [2], [3], and [4] in our discussion of related work.
>
> [1] Piyush Rai, Changwei Hu, Ricardo Henao, and Lawrence Carin. Large-Scale Bayesian Multi-Label Learning via Topic-Based Label Embeddings. In NIPS, 2015.
>
> [2] Ashish Kapoor, Raajay Viswanathan, and Prateek Jain. Multilabel Classification using Bayesian Compressed Sensing. In NIPS, 2012.
>
> [3] Kush Bhatia, Himanshu Jain, Purushottam Kar, Manik Varma, and Prateek Jain. Sparse Local Embeddings for Extreme Multi-label Classification. In NIPS 2015.
>
> [4] Yan Yan, Glenn Fung, Jennifer G. Dy, and Romer Rosales. Medical coding classification by leveraging inter-code relationships. In KDD 2010.

---

### Official Review · AnonReviewer4 · 2019-11-04
**Official Blind Review #4**

**Rating:** 6

**Review:**

This paper examines target-embedding autoencoders (TEAs) in theory and practice. TEAs autoencode the output (rather than input) space and find a mapping from the input to the latent representation of the output. The forward pass of the decoder (for the output space) is shared by the input-to-output computation.

Target-embedding autoencoders (TEAs) have previously been proposed and used in practice (though not necessarily by the "TEA" name).  The paper's presentation is confusing on this matter, at it claims to be the first to "motivate and formalize" TEAs; I do not believe it is appropriate to claim such a contribution in light of prior work. [Girdhar et al.] clearly utilizes a target-embedding autoencoder (see [Girdhar et al.] Figure 2). In addition, more recent published work clearly utilizes TEAs (though not named as such) as the centerpiece of their approaches. See, for example:

[A] Adrian V. Dalca, John Guttag, Mert R. Sabuncu. Anatomical Priors in Convolutional Networks for Unsupervised Biomedical Segmentation. CVPR, 2018.

[B] Mohammadreza Mostajabi, Michael Maire, Gregory Shakhnarovich. Regularizing Deep Networks by Modeling and Predicting Label Structure. CVPR, 2018.

Figure 2 of [A] and Figure 1 of [B] both clearly depict applying target-embedding autoencoders on semantic image segmentation problems. [B] operates in the same supervised representation-learning setting proposed here. Notably, [B] utilizes staged training -- learning the autoencoder first -- as discussed in Section 2 of the submitted paper, and finds that to be important for achieving a regularization effect.

The real applications explored by [A] and [B] are perhaps more challenging than the datasets used in experiments here.  The concluding sentence of the paper,"Target-representation learning is potentially applicable to any high-dimensional prediction task, and exploring its utility for specific domain-architectures may be a practical direction for future research" should be changed -- prior work has already successfully utilized TEAs in the specific domain of image segmentation.

Given that the paper has missed (not cited) highly related published work that applies TEAs in practice, a rewrite of Section 4 is required. In the appendix, Table 6, Table 7 and Section B.1 also need significant updates. The proposed approach is no longer a unique entry in Table 6 or 7 -- e.g. [B] already contributed "autoencoder component as regularization for learning predictor" (Table 7). Additionally, toy experiments in Section 5 appear less significant a contribution when multiple full-scale systems already employ TEAs.

This paper's theoretical analysis does appear to set it apart from prior work. However, theorems are developed for an extremely limited context (linear TEAs) and it is unclear whether or how they might extend to practical use cases (i.e. TEAs that are nonlinear, deep neural networks).

---

The extensive author response and updated paper address many of my original concerns.  I have updated my overall rating.

**Experience Assessment:**

I have published one or two papers in this area.

**Review Assessment: Checking Correctness Of Derivations And Theory:**

I assessed the sensibility of the derivations and theory.

**Review Assessment: Checking Correctness Of Experiments:**

I assessed the sensibility of the experiments.

**Review Assessment: Thoroughness In Paper Reading:**

I read the paper thoroughly.

---

> ### Author Response · Authors · 2019-11-11
> **Response to Reviewer 4 [Part 1/5]**
>
>
>
> Thank you for your thoughtful comments, and for referring to further papers applying target-embedding to image segmentation. Mainly, you point out that (1) target-embedding has previously been proposed and used in practice. In addition, you mention that (2) the theory developed is limited in practice due to the linear setting for analysis, and that (3) the experiments are "toy", considering that imaging applications are "more challenging".
>
> We address each in turn.
>
> We believe the specific positioning and contribution of the paper may not have been the most clear. Therefore we start by emphasizing our focus, in light of your comments. (1) First, we motivate and formalize TEA as a *general* framework, which "provide[s] a unifying perspective on recent applications of autoencoders to label-embedding" in disparate domains (p. 1). (2) This sets the stage for our theoretical contribution, which is to provide a *guarantee of generalization* for linear TEAs by demonstrating uniform stability. This allows us to distill its benefit in the simplest setting, removing any confounding factors from domain-specific architectures. (3) Our empirical novelty (in addition to verifying our claim for the linear case) is to extend validation of this approach to the *temporal* domain---for multi-variate sequence forecasting with recurrent architectures. While we make the point that certain prior works can be interpreted as specific instantiations of TEAs in the *static* setting, we are the first to do so in the recurrent, sequential setting and for both regression and classification---underscoring the further generality of this approach beyond feedforward instantiations.

---

> ### Author Response · Authors · 2019-11-11
> **Response to Reviewer 4 [Part 2/5]**
>
>
>
> *** (1) Context in Existing Work ***
>
> (1.1) We are happy to mention the additional image segmentation applications you reference, since they broadly relate to the notion of label-embedding. However, their focus is *very* different than ours in formalizing and analyzing TEAs. Our clearly specified focus is on the purely *supervised* setting (see title), where the embedding component is trained *jointly* in the TEA objective (see Equation 2)---this is fundamental to our theoretical result. Unlike the existing works we already cite, both of these additional works operate squarely outside of this setting. First, [Dalca, CVPR 2018] focuses entirely on the *unsupervised*, unpaired setting. We agree that it will go nicely in Table 6 as related work on target-embedding, but under "unsupervised / unpaired" in the "setting" column, which is very different. Second, both models in [Mostajabi, CVPR 2018] are *not jointly* trained at all. In particular, for the first model (Figure 1), "the decoder parameters are frozen", and "parameters internal to the decoder are never updated" after the initial phase (p. 4). So, this corresponds to a variant of the "No Joint" sensitivity we already investigate (plus an additional direct-prediction path). Similarly, their second model (Figure 4) regresses embeddings instead, but again the "encoder parameters are ... frozen" (p. 4). The benefit they observe thus derives solely from staged training. This will go nicely in Table 7, but certainly not under "joint training". In stark contrast, the fact that all components in TEA are jointly trained is central to our argument in deriving uniform stability for the generalization. (Of course, in our source-of-gains analysis, we also empirically demonstrate that the combination of joint and staged training performs the best).
>
> (1.2) That said, we actually agree with your high-level sentiment: It is also our understanding that the general concept of target-embedding has previously been used in practical applications. This is what we desire. In fact, the empirical efficacy of existing domain applications is precisely what motivates our theoretical contribution: To "provide a unified perspective on recent applications" (p. 1) of this idea to problems in multi-label classification, e.g. [Yeh, AAAI 2017] and 3D voxel prediction, e.g. [Girdhar, ECCV 2016]. These applications that we already cite (plus more in Table 7) are jointly trained in the supervised setting. Having them is good, and the more empirical results there are, the more relevant is our theoretical contribution. After all, while the effectiveness of the general target-embedding idea has been experimentally observed in a couple of application domains, there has not been any attempt (at all) at rigorous mathematical justification such as ours (Section 3). In this sense, our work is positioned analogously to [Le, NIPS 2018] and [Liu, TPAMI 2016], who respectively (theoretically) quantify the generalization benefit of multi-task learning and supervised feature-embedding---both in light of the fact that there *is* evidence that these paradigms are of use.
>
> (1.3) Lastly, we would like to emphasize that we don't claim to be the first to apply target-embedding. Even for TEAs, we explicitly mention throughout the paper [Yeh, AAAI 2017] and [Girdhar, ECCV 2016] that can be interpreted as specific instantiations of the general framework (p. 5, 6, and 8). In fact, we specifically point out that their joint-training variant corresponds to the TEA(L) setting of the framework (p. 8). (For greater clarity, we will add an earlier citation in Section 2 in the description of staged training). There is an important distinction between TEAs *per se*, versus specific *architectural* instantiations present in different application domains. We are the first to rigorously analyze the generalization benefit of the former (our theoretical novelty), and we are also the first to extend the latter to recurrent, multi-variate sequence forecasting and for both regression and classification (our empirical novelty). We will duly amend some of our language in the introductory and concluding remarks to more clearly and accurately position our contributions. Moreover, if there exists further relevant applications in this setting, we would be glad to mention them in support of our thesis.

---

> ### Author Response · Authors · 2019-11-11
> **Response to Reviewer 4 [Part 3/5]**
>
>
>
> *** (2) Theoretical Contribution ***
>
> (2.1) We would like to point out that a theoretical analysis of the linear setting is *not* of limited use---especially in a setting with no precedent. Our objective is to distill the essence of the target-embedding idea, such that we can isolate its theoretical benefit. To do so, a rigorous analysis deliberately and necessarily begins with a simplest incarnation of TEAs: the linear case. This is intentional and commonplace: As a matter of fact, (1) the seminal work on the generalization benefit of multi-task learning using Rademacher complexity operates in the linear setting [Maurer, JMLR 2006], and (2) the most recent landmark analysis by uniform stability is also performed in the linear setting [Liu, TPAMI 2016]. (3) For supervised feature-embedding as an auxiliary task, the first such analysis of generalization from stability is done in the linear setting [Le, NIPS 2018]. Similarly, (4) for multi-label classification, the generalization properties of label-embedding with norm-based regularization is first analyzed using the ERM framework---in the linear setting [Yu, ICML 2014].
>
> (2.2) The significance our analysis is that it allows us to *unambiguously* interpret its benefit as a regularizer. Now, it is often easy to argue on an intuitive level for the "regularizing" effect of some such additional loss term. For example, [Mostajabi, CVPR 2018] also refers to the "regularizing" effect of their label autoencoder (again, not trained jointly), but this expression is used loosely and intuitively, without (1) *identifying* or (2) *quantifying* the precise mathematical mechanism. In stark contrast, in our analysis the complete loss can be summarized and rewritten as $L(\mathbf{\Theta})=L_{p}(\mathbf{\Theta})+R_{1}(\mathbf{\Theta})+R_{2}(\mathbf{\Theta})$---that is, a combination of the primary prediction loss plus additional regularization, where $R_{1}(\mathbf{\Theta})=\frac{1}{N}\sum_{m=1}^{M}\ell_{r}(\mathbf{\Theta}\mathbf{W}_{e}\mathbf{b}_{m},\mathbf{b}_{m})$ and $R_{2}(\mathbf{\Theta}) = \frac{1}{N}\sum_{n=1}^{N}\ell_{r}(\mathbf{\Theta}\mathbf{W}_{e}\mathbf{y}_{n},\mathbf{y}_{n})-\frac{M}{N}L^{B}_{r}(\mathbf{\Theta})$. In particular, the proof of Theorem 1 depends critically on $R_{1}(\mathbf{\Theta})$ to achieve the upper-bound on instability. As a result, (1) this precisely *identifies* the regularizer in question, while (2) our uniform stability result *quantifies* the generalization benefit. This fact is already implicit in the analysis of Appendix A, but we can certainly mention it explicitly at the end of Section 3 for better emphasis.
>
> (2.3) This gives important theoretical insight into the empirical gains from TEAs. Significantly, we establish the fact that a tight generalization bound is obtained with absolutely nothing but the simple addition of the *joint* reconstruction loss (i.e. no additional unlabeled data, no explicit norm-based regularization, etc). Now, of course domain-specific application papers (cited previously in this response) have employed similar ideas in contexts of varying complexity, with potentially sophisticated architectures. Their *empirical* objective is to achieve SOTA in their specific domain (e.g. 2D image to 3D voxel prediction, segmentation, etc). In contrast, our *theoretical* objective as a first analysis of the TEA framework itself is---importantly---to *remove* the confounding effects of such tailored models (e.g. pretrained models, specific nonlinearities, custom losses, etc.) in order to distill the crux of the benefit in terms of generalization. (Empirically, of course, our experiments do cover both linear and nonlinear cases; see next section).

---

> ### Author Response · Authors · 2019-11-11
> **Response to Reviewer 4 [Part 4/5]**
>
>
>
> *** (3) Empirical Contribution ***
>
> (3.1) We agree that datasets for specific domains---such as imaging applications---can be especially challenging. However, we would like to point out that the experiments in Section 5 are far from "toy"---in fact, our empirical contribution serves a very specific purpose in the context of existing applications, as well as our theoretical findings. Section 5 (p. 6) positions this very clearly: Empirical work is limited to the *static* domain, including (1) multi-label classification in e.g. [Yeh, AAAI 2017], as well as (2) specific imaging application with convolutional architectures, e.g. [Girdhar, ECCV 2016]. What has *not* been explored at all is the utility of target-embedding in the *temporal* setting---for multivariate sequence data, especially via recurrent architectures and for both regression and classification. We are the first to study this, and we focus on an important application area (i.e. forecasting disease trajectories) with multiple real-world datasets. This domain was carefully selected as a particularly appropriate testbed, due to the fact that medical knowledge in this domain gives us confidence that the requisite prior for TEAs is satisfied---i.e. that variations in targets are driven by a lower-dimensional set of underlying factors (p. 1, 3, and 5); see scientific papers cited (p. 6). These points are all explained (with more detail) in the beginning of Section 5 (pp. 6-7), as well as a negative example to highlight the importance of the prior (Appendix E.4).
>
> (3.2) We also wish to kindly point out that forecasting multi-variate disease trajectories is far from "toy"; in fact, given our emphasis on early diagnosis (i.e. with intentionally limited windows of input), the forecasting task is deliberately set up to be challenging (p. 7)---especially compared with typical time-series prediction problems. The experimental setup is also far from "toy": Given our theoretical analysis, we first (1) verify our findings for the linear models, and then (2) extend our investigation to nonlinear, recurrent models. In addition, we (3) pick apart the sources of gains from both joint and staged training, as well as (4) comparing the alternate variations of TEAs present in existing work. Moreover, we (5) examine the incremental effect of additional norm-based regularization, (6) the sensitivity of TEAs to the strength of prior, as well as (7) the comparative sample complexity of prediction with and without target-embedding. Datasets are carefully chosen to obtain a variety of binary, continuous, and mixed-target settings. Furthermore, each individual outcome is reported across 10 random train-test splits, with extended results reported by timestep in a 10-page section of the appendix---inclusive of a negative example to highlight the importance of correctness of the prior. For these reasons, we submit that our experimental approach is highly comprehensive, and achieves both the goal of verifying our theoretical result, as well as being the first to demonstrate benefit in the recurrent, multi-variate sequence setting and for both regression and classification---which highlights the further generality of TEAs beyond feedforward instantiations.

---

> ### Author Response · Authors · 2019-11-11
> **Response to Reviewer 4 [Part 5/5]**
>
>
>
> *** Miscellaneous ***
>
> (M.1) Language: As mentioned previously, we will duly amend some of our language in the introduction and conclusion to more clearly and accurately position our contributions. However, we would like to point out that we never claim to be the *first* to "motivate and formalize" the general idea of autoencoding targets per se. Now, we are specifically "the first to formalize and quantify the theoretical benefit" of TEAs (p. 1), and we stand by this claim. Separately and before that, Section 2 serves to "motivate and formalize" TEAs by way of unifying multiple multiple application papers in different domains under a single framework; these existing works are cited extensively, and we do not claim to precede them. This already includes [Girdhar, ECCV 2016], which we explicitly identify as an instantiation of TEAs (p. 5, 6, and 8). Answer (1.1) to (1.3) gives more detail.
>
> (M.2) Section 4: We would like to thank you again for pointing out the two papers. As mentioned previously, we are glad to cite them. However, in light of answers (1.1) through (1.3), we don't believe this warrants a full "rewrite". As explained, neither [Dalca, CVPR 2018] nor [Mostajabi, CVPR 2018] are directly comparable. We will add references to them in Section 4, and include the former in Table 6 (but under "unsupervised / unpaired"), as well as the latter in Table 7 (but under "not joint"). However, this does not involve *rewriting* any part of the existing (and very comprehensive related works), the full version of which already spans over 3 pages with 3 summary tables. Finally let us reiterate that, should there be further existing applications more relevant to TEAs, we are very glad to mention them to better support our thesis.
>
> (M.3) Concluding sentence: We agree that this may be misleading. We will append a clarification to the existing phrase "potentially applicable to any [...] task", so that it will become "potentially applicable to any [...] task beyond classification and image processing applications".
>
> * Aside from the two papers that Reviewer 4 referenced, all citations can be found in the original bibliography.
> * [Nov 14 Update]: Both papers are now included in the updated manuscript (text in Section 4, text in Appendix B, as well as entries for comparisons in summary Tables 6 and 7).

---

> ### Author Response · Authors · 2019-11-15
> **Dear Reviewer 4**
>
> We are sincerely grateful for your time and energy in the review process. In light of our responses (Nov 11) and revisions (Nov 13), we would appreciate if the reviewer kindly let us know of any leftover concerns in the very limited time remaining. With our responses and revisions, we humbly hope that (similar to Reviewer 2) the reviewer would kindly consider revising their rating. Thank you.

---

### Author Response · Authors · 2019-11-13
**Revised Paper**



We thank the reviewers again for their thoughtful comments.

We have updated the paper to more clearly and accurately reflect our specific positioning and contributions. Broadly, this includes (a.) fine-tuned language in the abstract, introduction, and conclusion, (b.) specific suggestions for additional clarifications, (c.) specific suggestions for additional related work, as well as (d.) additional sensitivities for thoroughness. We are grateful for all questions and suggestions, which improve the clarity and thoroughness of our arguments and presentation.

[Note on Revisions]
*Blue* indicates any additions and revisions.
*Purple* indicates purely formatting-related changes (i.e. existing words italicized for emphasis).

[Note on Related Work]
We deeply appreciate the suggestions for additional related work, and have included all such works. Given our framework-level focus, the more works there are that relate to label-embedding in general, the more strongly our thesis is motivated and supported. Since our work intersects three (very) broad areas of research, it is inevitable that any list of related works---however large---will not be 100% exhaustive. That said, while our related work section already spans 4 pages (and the bibliography already spans 5), we do provide concise and detailed summary tables that organize their distinctions and relevance in order to contextualize our work.

*** (a.) Language and Presentation ***

[Reviewers 1, 2, and 4]
The abstract, introduction, and introductory language to Section 3 (immediately before and after the section title) now most clearly reflects our positioning and three main contributions: (1) formalizing TEAs as a *general framework* that unifies several recent applications of label-embedding in disparate domains, (2) providing a first theoretical *learning guarantee* for linear TEAs by demonstrating uniform stability, and (3) first demonstrating empirically the further generality of TEAs (beyond feedforward instantiations for static classification) to the *temporal setting* for multi-variate, recurrent sequence forecasting, both regression and classification. We especially emphasize the importance of *joint* training to our analyses, as well as our focus on the purely *supervised* setting (e.g. now italicized in the abstract). The conclusion is also updated to more clearly reflect the fact that static classification applications can be regarded as existing instances of the framework. Broadly, this adds to responses (3.1) and (3.3) for Reviewer 2, and responses (1.3), (M.1), and (M.3) for Reviewer 4.

*** (b.) Additional Clarifications ***

[Reviewer 1]
The introduction now contains more examples of tasks with high-dimensional output, per our response (1.1). Since we extensively discuss the setting of disease trajectory forecasting later on, this is now replaced with the existing static classification applications to image tagging, text annotation, and image segmentation.

[Reviewer 2]
Section 2 now contains a better explanation of training and inference, per our response (3.5.a). In particular, we explicitly mention which parameters are trained in which stages, such that the reader does not need to wait until Appendix C for this general information. That said, we also include explicit mention of Algorithm 1 for quick reference, as well as pointing to the much more detailed diagrams in Figure 4 in Appendix C for step-by-step illustrations of training and inference.

[Reviewers 1, 2, and 4]
In the introduction to Section 3, as well as a separate paragraph preceding Section 3, we doubly emphasize our focus as a *framework-level* analysis. In addition to pp. 5-6 and 8, we additionally reiterate earlier here that existing static classification applications can be abstractly regarded as instantiations of this framework, such that there should not be any such confusion from the beginning to the end of the paper. In particular, this addresses response (3.1) for Reviewer 1, and adds to responses (3.1) and (3.3) for Reviewer 2, and responses (1.3), (M.1), and (M.3) for Reviewer 4.

[Reviewer 2 and 4]
To better highlight the significance of our theoretical result, we mention explicitly at the end of Section 3 how this enables us to unambiguously identify and quantify the benefit, in contrast with standard intuition-based arguments. First, we have included an additional detailed remark (Remark 4) at the end of Appendix A. In addition, a shorter version of the argument can be found at the end of Section 3, which points to Remark 4. (The original Remarks 4 and 5 are now renumbered as 5 and 6). This specifically reflects the added analysis in response (3.2) for Reviewer 2, and in responses (2.1) and (2.2) for Reviewer 4.

[Reviewer 2]
Section 5 now contains a better exposition of the source of gains, per our response (3.4). In particular, the details of each of the experimental settings are expanded with more comparisons and explanations.

---

> ### Author Response · Authors · 2019-11-13
> **Revised Paper (continued)**
>
>
>
> *** (c.) Additional Related Work ***
>
> [Reviewer 3]
> Thank you again for the suggestion to include extreme multi-label classification. These now appear at the end of Section B.2, including all four works cited in our original response, as well as a reference to the data and compilation webpage for further detail.
>
> [Reviewer 4]
> Thank you again for the suggestions of both [Dalca, CVPR 2018] and [Mostajabi, CVPR 2018]. As promised, the former fits nicely into Table 6, and the latter into Table 7. We agree that they are both broadly related to target-embedding. However, as explained in detail in response (1.1), (1.2), and (M.2), the former is highlighted under "unsupervised, unpaired", and the latter is highlighted under "not jointly trained"---both of which are important distinctions central to our theoretical and empirical analyses. These works are also mentioned in Section 4 (as well as Appendix B), with these distinctions made clear.
>
> *** (d.) Additional Sensitivities ***
>
> [Reviewer 1]
> As part of our response (2.4), we explored the idea of performance degradation due to out-of-distribution test samples. The preliminary results (along with a detailed description) are now included in Table 37 in Section E.5.
>
> [Reviewer 2]
> As part of our response to question (3.5.b), we explored the idea of rearranging the order of the training stages. The results (along with a detailed description) are now included in Table 38 in Section E.5. These are in addition to the 7 focuses of our empirical analysis in Section 5, summarized in response (3.3.a).

---

### Comment · Area_Chair1 · 2019-11-14
**Reviewers, any additional feedback following author responses?**

Dear Reviewers, thanks for your thoughtful input on this submission!  The authors have now responded to your comments.  Please be sure to go through their replies and revisions.  If you have additional feedback or questions, it would be great to get them this week while the authors still have the opportunity to respond/revise further.

Also, there is a wide range of scores for this submission.  Please consider whether the author responses and/or comments of other reviewers affect your recommendation.  Thanks!

---

### Decision · Program_Chairs · 2019-12-19

**Decision:**

Accept (Talk)

**Comment:**

The paper presents a general view of supervised learning models that are jointly trained with a model for embedding the labels (targets), which the authors dub target-embedding autoencoders (TEAs).  Similar models have been studied before, but this paper unifies the idea and studies more carefully various components of it.  It provides a proof for the specific case of linear models and a set of experiments on disease trajectory prediction tasks.  The reviewer concerns were addressed well by the authors and I believe the paper is now strong.  It would be even stronger if it included more tasks (and in particular some "typical" tasks that more of the community is focusing on), and the theoretical part is to my mind not a major contribution, or at least not as large as the paper implies, because it analyzes a much simpler model than anyone is likely to use TEAs for.